# Diameter dependence of transport through nuclear pore complex mimics studied using optical nanopores

Nils Klughammer[1†], Anders Barth[1†], Maurice Dekker[2], Alessio Fragasso[1‡], Patrick R Onck[2], Cees Dekker[1]*

[1]Department of Bionanoscience, Kavli Institute of Nanoscience, Delft University of Technology, Delft, Netherlands; [2]Zernike Institute for Advanced Materials, University of Groningen, Groningen, Netherlands

*For correspondence:
C.Dekker@tudelft.nl

[†]These authors contributed equally to this work

Present address: [‡]Department of Biology and Institute of Chemistry, Engineering and Medicine for Human Health, Stanford University, Palo Alto, United States

Competing interest: The authors declare that no competing interests exist.

**Abstract** The nuclear pore complex (NPC) regulates the selective transport of large biomolecules through the nuclear envelope. As a model system for nuclear transport, we construct NPC mimics by functionalizing the pore walls of freestanding palladium zero-mode waveguides with the FG-nucleoporin Nsp1. This approach enables the measurement of single-molecule translocations through individual pores using optical detection. We probe the selectivity of Nsp1-coated pores by quantitatively comparing the translocation rates of the nuclear transport receptor Kap95 to the inert probe BSA over a wide range of pore sizes from 35 nm to 160 nm. Pores below 55 ± 5 nm show significant selectivity that gradually decreases for larger pores. This finding is corroborated by coarse-grained molecular dynamics simulations of the Nsp1 mesh within the pore, which suggest that leakage of BSA occurs by diffusion through transient openings within the dynamic mesh. Furthermore, we experimentally observe a modulation of the BSA permeation when varying the concentration of Kap95. The results demonstrate the potential of single-molecule fluorescence measurements on biomimetic NPCs to elucidate the principles of nuclear transport.

## eLife assessment

This **important** study reports on a new method for the fabrication and the analysis of the transport through nuclear pore complexes mimic. Methods, data, and analyses are **convincing** and show a clear correlation between the size of the nuclear pore complex mimic and its transport selectivity. This work will be of high interest to biologists and biophysicists working on the mechanosensitivity of nucleocytoplasmic transport.

## Introduction

The nuclear pore complex (NPC) forms the sole connection across the nuclear envelope that regulates all transport between the cytoplasm and nucleus. The central channel of this large protein complex (52 MDa in yeast to about 120 MDa in humans, *Reichelt et al., 1990*; *Kim et al., 2018*) is lined with intrinsically disordered proteins that are rich in phenylalanine–glycine (FG) repeats, termed FG-Nups. The central transporter mesh constituted by these FG-Nups forms a selective barrier that facilitates the transport of dedicated nuclear transport receptors (NTRs) while blocking other proteins (*Kim et al., 2018*). Recently, it has been discovered that the inner diameter of the central transporter is variable and can dilate from 40 nm to 70 nm under different stress conditions (*Zimmerli et al., 2021*; *Akey et al., 2022*). How NPC dilation affects the efficiency and selectivity of nuclear transport remains an open question.

Despite extensive structural knowledge of the NPC scaffold, the mechanism of the selective barrier formed by the disordered FG-Nups remains highly debated (*Lim et al., 2015*; *Schmidt and Görlich, 2016*; *Jovanovic-Talisman and Zilman, 2017*). Remarkably, the NPC poses this selective barrier while enabling very high transport rates of ~1000 molecules per second that traverse the pore, referred to as the 'transport paradox' (*Beck and Hurt, 2017*). Whereas small molecules pass through the NPC channel without much obstruction, translocation of larger biomolecules is increasingly hindered in a continuous manner above a diameter of ~5 nm or a mass of ~30 kDa to 40 kDa, which leads to an effective blockade for large molecules unless they specifically interact with the FG-Nups mesh (*Keminer and Peters, 1999*; *Mohr et al., 2009*; *Schmidt and Görlich, 2016*; *Popken et al., 2015*; *Timney et al., 2016*). The efficient transport of large cargoes that carry a nuclear localization signal is facilitated by NTRs, which engage in multivalent interactions with the FG-repeats in the central transporter (*Aramburu and Lemke, 2017*). One of the most studied systems is the Kap95–Kap60 system in yeast (Impβ–Impα in humans), responsible for the import of proteins into the nucleus (*Görlich and Kutay, 1999*).

Over the years, many models have been proposed to explain the selective properties of the central transporter, often originating from in vitro studies of isolated FG-Nups (*Rout et al., 2003*; *Frey et al., 2006*; *Frey and Görlich, 2007*; *Yamada et al., 2010*; *Schleicher et al., 2014*; *Kapinos et al., 2014*; *Kapinos et al., 2017*; *Zilman et al., 2007*; *Zilman et al., 2010*). Following early models that explained the selectivity solely based on the properties of the FG-Nups, more recent 'Kap-centric' models suggested a central role of NTRs as an active component of the selective barrier (*Schleicher et al., 2014*; *Kapinos et al., 2014*; *Kapinos et al., 2017*; *Fragasso et al., 2022*; *Zilman et al., 2010*), supported by the presence of a large amount of strongly interacting transport receptors within the central transporter (*Kim et al., 2018*). This 'slow phase' shows reduced mobility due to the high affinity of NTRs to the FG-Nup mesh. As the FG-Nup mesh is saturated with NTRs, a mobile phase of NTRs emerges (*Schleicher et al., 2014*) that is thought to diffuse along dedicated channels. So far, however, the dependence of transport rates and selectivity on the concentration of NTRs has hardly been studied directly.

Much research on the transport mechanism is carried out using in vitro experiments since direct studies on the native NPC in its full complexity remain challenging. Using minimal biomimetic systems, it has been shown that a selective barrier with similar properties as the native NPC can be reconstituted using even a single native FG-Nup (*Frey and Görlich, 2007*; *Jovanovic-Talisman et al., 2009*; *Kowalczyk et al., 2011*; *Ananth et al., 2018*; *Celetti et al., 2020*; *Shen et al., 2021*; *Shen et al., 2023b*) or artificial mimics thereof (*Fragasso et al., 2021*; *Ng et al., 2021*). Probing translocations through the FG-Nup mesh using solid-state nanopores grafted with FG-Nups was pioneered by *Jovanovic-Talisman et al., 2009* using optical detection of fluorescently tagged molecules. However, this approach was limited to bulk measurements of transport through porous membranes that contained many such pores in parallel, thus lacking single-pore and single-molecule resolution. This limitation was subsequently addressed by measuring the electric current through individual functionalized $SiN_x$ nanopores (*Kowalczyk et al., 2011*; *Ananth et al., 2018*; *Fragasso et al., 2021*). While this approach offers single-molecule sensitivity, the current-based readout remains unspecific (i.e., cannot distinguish different proteins), requires the application of a bias voltage, which may influence the transport rate and speed of the translocations, and offers limited signal-to-noise ratio. It remains therefore restricted to pore sizes around 30–50 nm where the translocation of single molecules leads to a detectable current drop. For larger pores, the relative current blockage caused by a translocating molecule is too small compared to the fluctuations of the ion flow through the FG-Nup mesh, while the conductance of smaller coated pores is too low to detect translocation events over the inherent noise present in the experiment.

Here, we implement a fluorescence-based assay for the simultaneous detection of single-molecule translocations of different molecular species through solid-state metal nanopores made of palladium, based on our previous work (*Klughammer and Dekker, 2021*). The translocation of individual fluorescently labeled molecules is monitored by a focused laser beam at a single nanopore, which selectively excites molecules that exit from the pore (*Figure 1A*). By labeling different molecular species with different fluorophores, the species can be distinguished based on their signal. The < 200 nm nanopores act as zero-mode waveguides (ZMW, *Levene et al., 2003*) that block the propagation of the 485 nm and 640 nm wavelength excitation light through the metal membrane. Such ZMWs

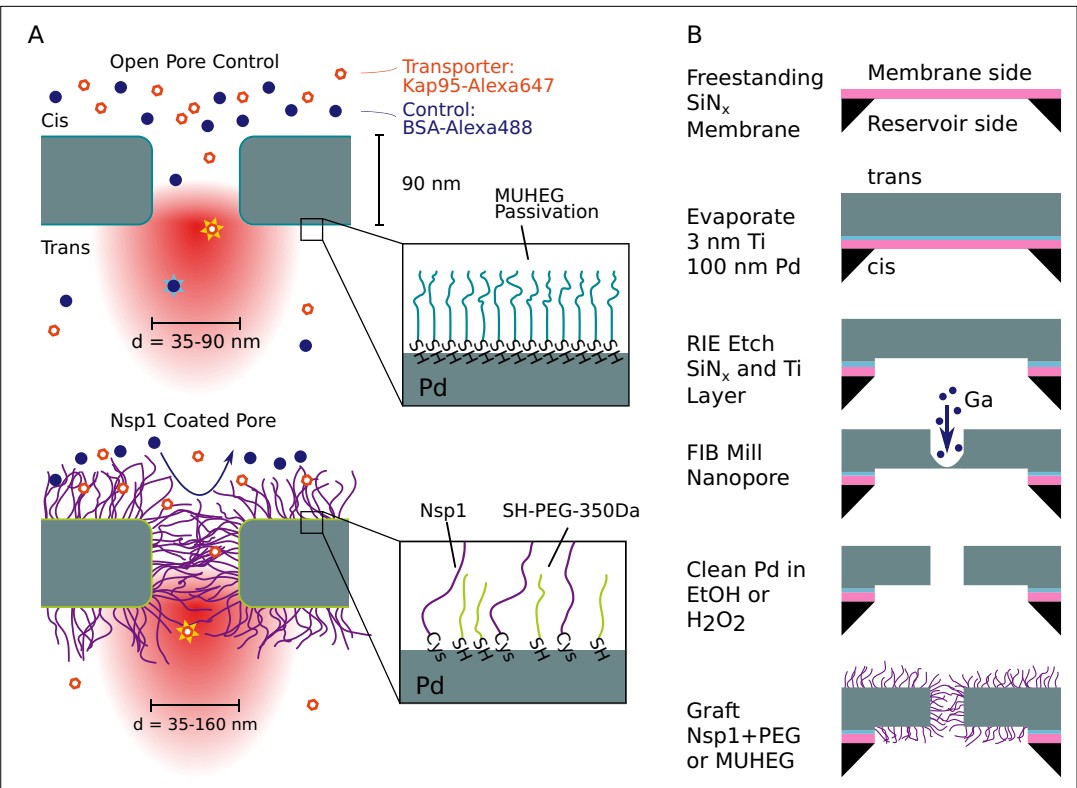

**Figure 1.** Experimental principle and nanofabrication. (**A**) Sketch of the experimental principle. Nanopores in a metal membrane block light from traversing if the pore diameter is small compared to the wavelength of light. The selectivity of Nsp1-coated metal nanopores is probed by measuring the translocation rate of fluorescently labeled proteins from the top reservoir (*cis*) to the detection (*trans*) side, where they rapidly diffuse out of the laser focus. Measurements on open pores (top) serve as a control where both the nuclear transport receptor (NTR) Kap95 and the inert protein probe BSA pass unhindered. Nsp1-coated pores are expected to block the translocation of BSA while still allowing Kap95 to translocate. Zooms at bottom right illustrate the passivation of open pores with (1-mercaptoundec-11-yl)hexa(ethylene glycol) (MUHEG) (top) and functionalization of the palladium surface with the FG-nucleoporin Nsp1 and 350 Da SH-PEG (bottom), achieved via thiol-palladium chemistry. (**B**) Fabrication of nanopores in a freestanding palladium membrane was performed by physical vapor deposition of palladium onto silicon nitride (SiN$_x$), reactive ion etching (RIE), and focused ion beam (FIB) milling of the nanopores. The palladium surface was then cleaned either with H$_2$O$_2$ or ethanol to remove contaminants before the functionalization step.

have been used on top of glass surfaces (*Levene et al., 2003*; *Samiee et al., 2005*; *Rigneault et al., 2005*) but also in a freestanding manner (*Auger et al., 2014*; *Assad et al., 2017*; *Klughammer and Dekker, 2021*) as applied here. Notably, our method does not require a bias voltage as it relies solely on the free diffusion of the fluorescent molecules, nor does the detection efficiency depend on the pore diameter. We developed robust protocols for efficient passivation of open pores and functionalization of the palladium surface with the FG-Nup Nsp1 to build a functional NPC mimic. After establishing a baseline for the translocation of fluorescently labeled proteins through open pores, we probe the selectivity of the biomimetic NPCs over a range of pore diameters from 35 nm to 160 nm by comparing the event rates of the transporter Kap95 to that of the inert probe bovine serum albumin (BSA). We find that the selectivity of Nsp1-coated pores decreases with pore diameter, with the main loss of selectivity happening at about a diameter of 55 nm. Selectivity for smaller pores arises because Kap95 proteins can efficiently cross the FG-Nup mesh in the pore while BSA transport is hindered. For larger pores, transport selectivity is gradually lost due to increasing leakage of BSA. Coarse-grained molecular dynamics (CGMD) simulations reproduce these experimental findings and show that the loss of selectivity is due to the formation of voids within the Nsp1 mesh if comparable size to BSA molecules. For small pores, these voids are transient, whereas for large pores they become persistent and a central channel forms. Upon increasing the concentration of the transporter Kap95 in the experiment, we observe a moderate increase of the transport selectivity for pores with a diameter

below 50 nm. Intriguingly, we find that the event rate of BSA translocations for large pores *increases* with Kap95 concentration, suggesting that filling the FG-Nup mesh with NTRs reduces the selective volume. These results highlight the potential of our approach to unravel the physical principles underlying nuclear transport.

## Results

### Fabrication, surface grafting, and measurement setup

Our experimental approach is based on a freestanding palladium membrane into which nanopores were drilled using focused ion beam (FIB) milling (*Figure 1B*). Fabrication of the metal membranes was achieved by evaporating a layer of Pd onto a SiN$_x$ membrane (20 nm thickness) that was subsequently removed by reactive ion etching (in a process that was slightly adapted from *Klughammer and Dekker, 2021*, see section 'Fabrication of freestanding Pd ZMWs' for details). After processing, the Pd membrane had a thickness of ≈ 90 nm. To functionalize the metal surface, we applied standard thiol chemistry, which is well established for metals such as gold or palladium (*Love et al., 2003*; *Jiang et al., 2004*). After the FIB-milling step, a cleaning step was required to remove impurities from the Pd surface prior to thiol binding to ensure efficient surface grafting. Since previous protocols relied on freshly prepared surfaces, we developed a gentle cleaning protocol that used either hydrogen peroxide or boiling ethanol, inspired by *Majid et al., 2003* (see 'Materials and methods' and Appendix 1). The cleaning step ensured that the Pd surface was competent for thiol binding, as validated by quartz crystal microbalance with dissipation monitoring (QCM-D) experiments (see *Appendix 1—figures 1–3*) and surface plasmon resonance measurements (*Andersson et al., 2022*). We additionally confirmed that the cleaning procedure does not alter the pore shape or closes the pores by transmission electron microscopy (TEM) (cf. images shown in *Klughammer et al., 2023b*).

To prevent unspecific sticking of proteins to the metal surface, the control open pores were passivated using (1-mercaptoundec-11-yl)hexa(ethylene glycol) (MUHEG), which forms a self-assembled monolayer by thiol-Pd binding enhanced with a dense packing due to the hydrophobic interactions between the alkane groups while providing a hydrophilic surface through the terminal hexaethylene glycol groups (*Prime and Whitesides, 1993*, *Figure 1A*, top). Coating of the biomimetic pores was performed with Nsp1 containing a C-terminal cysteine at a concentration of 1 µM. We expect a lower limit for the grafting distance in the pore of 6.5 nm as measured previously by surface plasmon resonance experiments on flat gold surfaces (*Fragasso et al., 2022*). Additional passivation of the remaining free area between Nsp1 molecules was achieved using short thiolated 350 Da PEG molecules (*Jovanovic-Talisman et al., 2009*; *Jovanovic-Talisman et al., 2014*, *Figure 1A*, bottom).

After functionalization, the palladium membrane was mounted in a flow cell made of polydimethylsiloxane (PDMS), which provides a reservoir on the upper side for the addition of analyte, as well as a flow channel on the lower detection side to which a constant flow was applied to avoid accumulation of fluorescently labeled molecules (*Figure 2A*). By diffusion, single proteins traveled through the pore upon which their attached fluorophore got excited by the laser. The fluorescence signal of molecules exiting the pore on the lower side was measured using a two-color confocal fluorescence microscope. A high numerical aperture objective lens with a long working distance was used to focus the picosecond pulsed lasers in a diffraction-limited spot on the pore exit. The fluorescence signal was detected on single-photon avalanche photodiodes after passing through a pinhole and bandpass filter. For accurate and unbiased detection of the signal spikes originating from single-molecule translocation, we adapted a change point detection algorithm (*Watkins and Yang, 2005*) that takes full advantage of the single-photon information, as described previously in *Klughammer and Dekker, 2021*. To avoid biases in the event detection, we ensured that experimental parameters such as the event duration and the photons per molecule did not vary between experiments, for example, due to variations of the laser intensity or setup alignment (see 'Data analysis' for more details). Note that the applied pressure induces a hydrodynamic flow that acts against the concentration gradient and results in an approximate reduction of the measured event rates by 5% (*Appendix 3—figure 2*).

The metal membrane was thick enough to prevent the incident laser light from reaching the other side, which served to suppress the background fluorescence coming from the reservoir side. Additionally, the nanopore acted as a ZMW, resulting in an evanescent wave within the pore that exponentially decays on a length scale of 10–20 nm, depending on the pore size (*Levene et al., 2003*). To obtain more

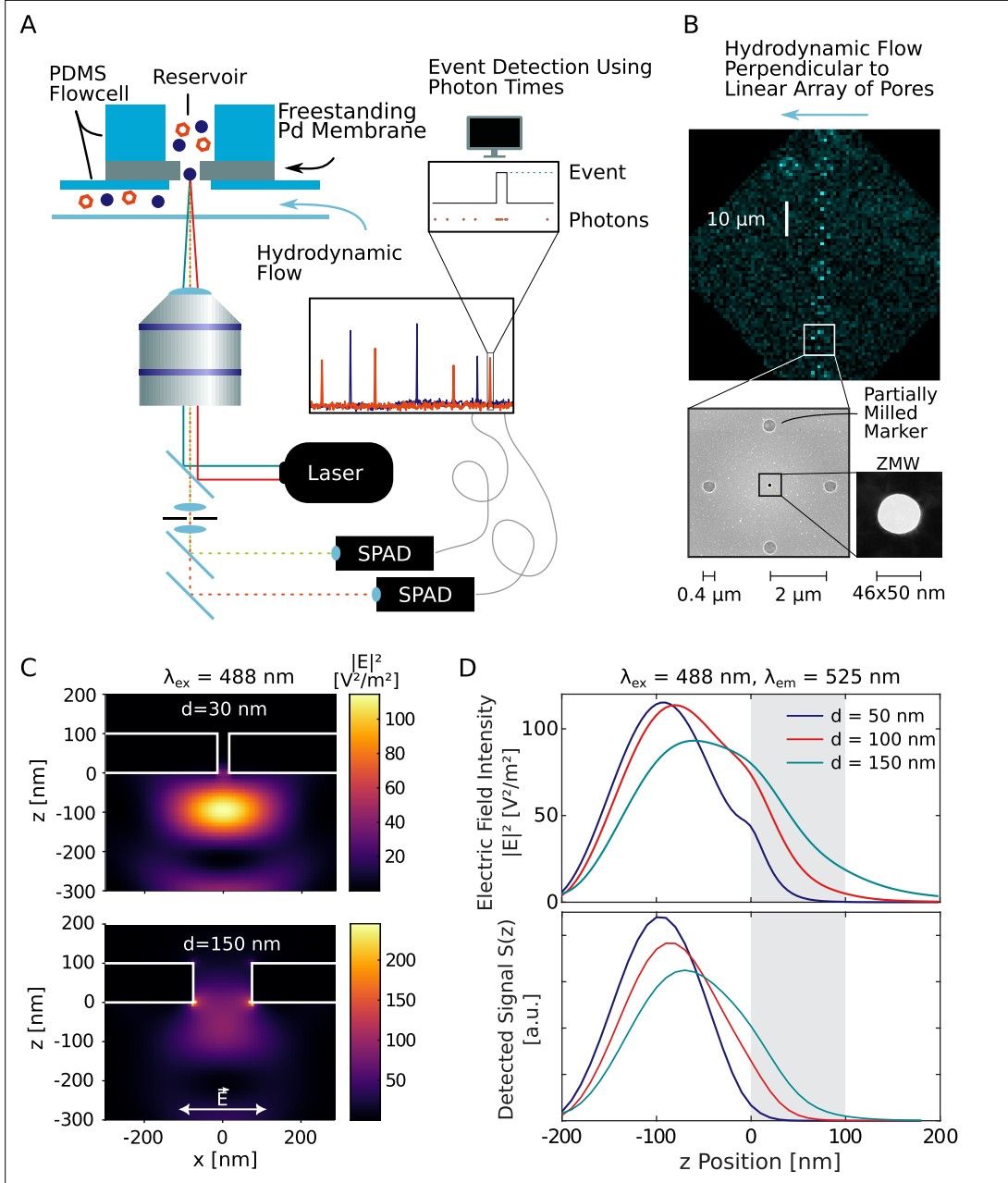

**Figure 2.** Experimental setup. (**A**) A freestanding Pd membrane containing the nanopores was mounted on a confocal microscope using a polydimethylsiloxane (PDMS) flow cell with a reservoir of ≈ 3L on the dark (*cis*) side and a flow channel on the detection (*trans*) side that faced the objective lens. A constant flow in the channel avoided the accumulation of analytes on the detection side. The lasers were focused onto the nanopore by a high numerical aperture (NA) objective lens and the fluorescence signal was detected on single-photon avalanche photodiodes. From the recorded photon arrival times, fluorescence bursts were detected using a change point detection algorithm. (**B**) A total of eight pores were milled into the Pd membrane, each surrounded by partially milled markers that facilitate the localization of the nanopores in a bright-field image (top). An additional marker was added such that the individual pores in the array could be identified. A scanning electron microscope image of a single pore with markers is shown below. The size and shape of each zero-mode waveguide (ZMW) pore used in this study was determined using transmission electron microscopy (bottom right). (**C**) Simulated electric field intensity distributions in the xz plane near a freestanding ZMW for pore diameters of 30 nm (top) and 150 nm (bottom). A Gaussian laser beam was focused on the pore at a wavelength of 488 nm, polarized in the x-direction. See *Appendix 2—figure 1* for different pore sizes and excitation at 640 nm. (**D**) Electric field intensity $|E^2|$ (top) and total detected signal $S(z)$ (bottom) as a function of the z-position along the center of the pore for an excitation wavelength of 488 nm and an emission wavelength of 525 nm, corresponding to the blue detection channel, for pores of 50 nm, 100 nm, and 150 nm diameter. The palladium membrane is indicated by the gray shaded area. See *Appendix 2—figure 4* for the corresponding plots for the red channel.

detailed insights into the optical properties in the proximity of the freestanding ZMW, we performed finite-difference time-domain simulations of the excitation electric field intensity and fluorescence emission (*Figure 2C and D*, *Appendix 2—figure 1*). As expected, the propagation of the excitation light is effectively blocked by the nanoaperture and the electric field intensity $|E|^2$ decays exponentially within the nanopore. Interestingly, the presence of the reflecting metal surface also affects the intensity distribution further away from the nanopore, leading to the formation of a standing wave pattern. This effect is most visible for plane wave excitation (see *Appendix 2—figure 3*), but is also present for focused excitation (*Figure 2C*, *Appendix 2—figure 2*). Molecules exiting the pore were hence mostly detected in the first lobe of the excitation profile and will likely diffuse away laterally before reaching intensity maxima further away from the membrane. To fully model the detected fluorescence signal $S(z)$, it is necessary to account for the modulation of the fluorescence quantum yield due to enhancement of the radiative and non-radiative rates by the metal nanostructure, as well as for the fraction of the signal that is emitted toward the upper side and thus cannot be detected (see 'Materials and methods' and *Appendix 2—figure 4*). This further improves the blocking capability of the ZMW such that background signal from the reservoir is effectively suppressed even for large pores of 150 nm diameter (see *Figure 2D* and *Yang et al., 2023* for a more detailed discussion). The detected

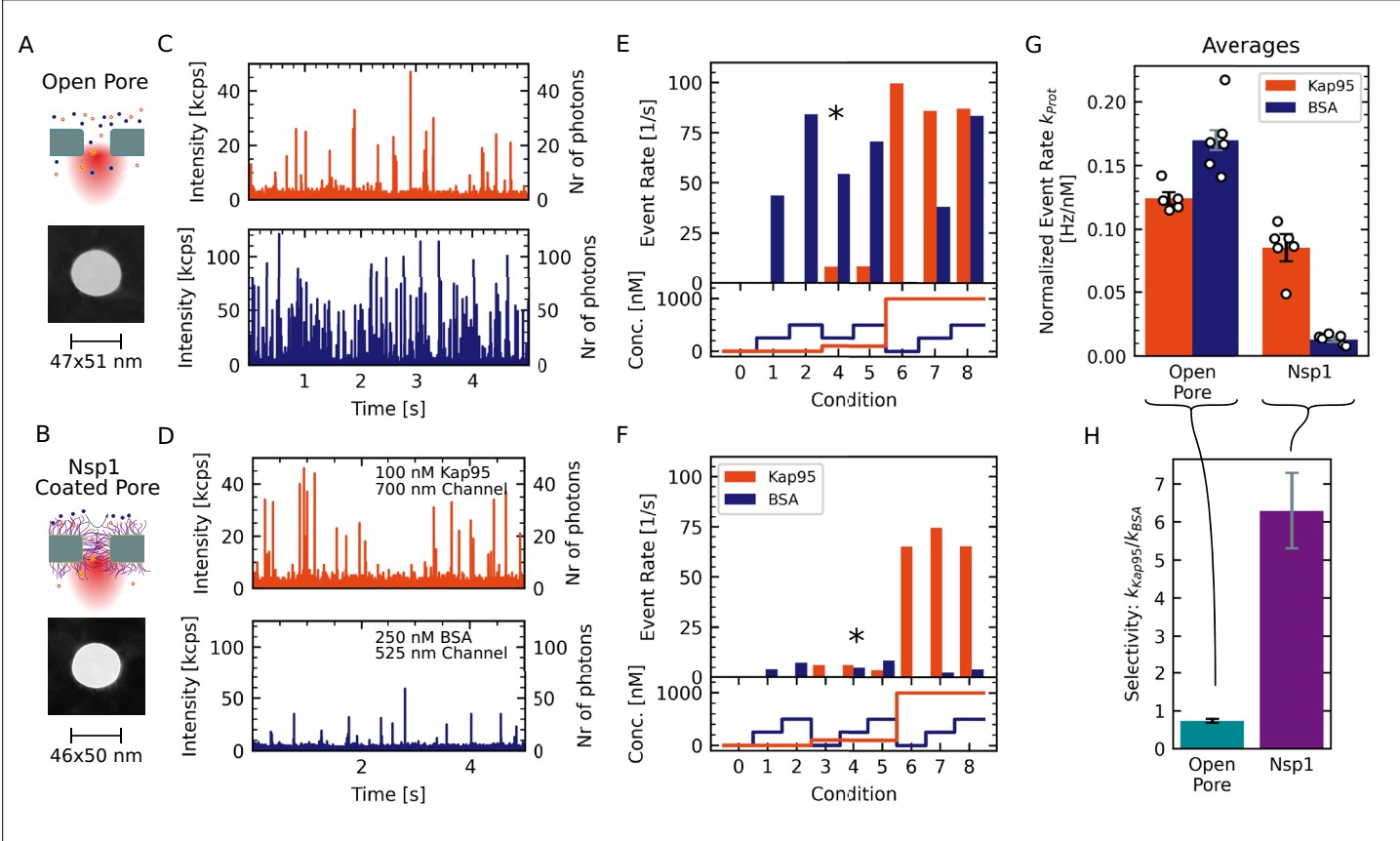

**Figure 3.** Experimental workflow for measuring the selectivity of Nsp1-coated nanopores. (**A, B**) Two pores of approximately 48 nm diameter were coated with (1-mercaptoundec-11-yl)hexa(ethylene glycol) (MUHEG) (**A**) or functionalized with Nsp1 (**B**). Pore dimensions were measured from transmission electron microscopy (TEM) micrographs. (**C, D**) Fluorescence time traces recorded for the open and Nsp1-coated pores that are shown in (**A**) and (**B**) for Kap95–Alexa647 at 100 nM (red) and BSA–Alexa488 at 250 nM (blue). Both proteins were present at the same time. Whereas the Kap95 signal is comparable between the open and Nsp1-coated pores, a clear decrease of the BSA event rate is evident for the Nsp1-coated pore. (**E, F**) Measured event rates (top panel) resulting from the analyte concentrations for the different conditions probed sequentially during the experiment (bottom panel; see Appendix 4 for details). The concentrations used in the time traces shown in (**C**) and (**D**) are indicated with an asterisk. (**G**) In order to compare the different conditions, the obtained event rates are normalized to the respective protein concentration and corrected for the labeling degree (white dots in **G**). Bars indicate the average normalized event rates $k_{Kap95}$ or $k_{BSA}$ of the pore. Error bars represent the standard error of the mean.

(**H**) The selectivity was calculated as the ratio of the average normalized event rates $\frac{k_{Kap95}}{k_{BSA}}$. Errors are propagated from the data shown in (**G**). The data show a clear selectivity of the Nsp1-coated pore compared to the open pore.

fluorescence signal thus originates predominantly from molecules that exit from the pore and diffuse into the surrounding solution, which were excited by the truncated excitation volume below the pore.

Since the nanopores block the propagation of the excitation light across the membrane, they were not visible in a bright-field optical image. To facilitate the precise localization of the nanopores, we hence added a grid of *partially* milled markers centered around each pore that were easily visible on the microscope (*Figure 2B*). To improve the throughput of the measurements, a linear array of eight pores of varying size was milled into the palladium membrane, and these pores were probed consecutively during the experiments. Crosstalk between pores was minimized by applying a flow orthogonal to the pore array, resulting in a false detection rate of less than 2 % (Appendix 3).

## NPC mimics show transport selectivity

To illustrate the workflow for estimating the transport selectivity of Nsp1-coated pores, we first describe our experiments on a pore with a diameter of 50 nm for which selectivity has previously been reported with Nsp1 in conductance measurements (*Ananth et al., 2018*) as shown in *Figure 3A and B*. The recorded time traces of Kap95 labeled with the Alexa Fluor 647 dye (Alexa647, orange) at a concentration of 100 nM and BSA labeled with the Alexa Fluor 488 dye (Alexa488, blue) at a concentration of 250 nM show efficient translocations for both proteins through the open pore (*Figure 3C*). While Kap95 still translocated through the Nsp1-coated pore at a high rate, BSA was clearly hindered as is evident from the reduction of the number of signal spikes in the time trace (*Figure 3D*).

For each pore, we probed a total of three different concentrations of BSA (0 nM, 250 nM, and 500 nM) and Kap95 (0 nM, 100 nM, and 1000 nM ), and all combinations thereof. This resulted in nine different conditions that were tested consecutively (see *Figure 3E and F*, bottom), which allowed us to assess the linearity of the measured event rates with respect to the analyte concentration. Since Kap95 interacts strongly with the Nsp1 mesh, we performed a step-wise increase of the Kap95 concentration throughout the measurement while probing the BSA response at every step (see Appendix 4). Importantly, this scheme enables us to test the influence of the Kap95 concentration on the event rates of BSA, as will be discussed below.

The measured event rates for Kap95 and BSA at the probed concentrations are shown in *Figure 3E and F*, confirming the reduction of the BSA translocation for the Nsp1-coated pore. To quantify the selectivity, we computed the concentration-normalized event rate $k_{\text{Kap95}}$ or $k_{\text{BSA}}$ in units of Hz/nM as the average over all probed conditions and corrected for the degree of the fluorescent labeling (1 for BSA and 0.7 for Kap95, *Figure 3G*). Finally, we define the selectivity of the pore as the ratio of the average normalized event rates for Kap95 and BSA, $\frac{k_{\text{Kap95}}}{k_{\text{BSA}}}$, which should be independent of the pore diameter for open pores. High selectivity values indicate that Kap95 translocates at a higher rate compared to BSA. For the open pore, we find a selectivity of 0.73±0.05. A selectivity value between 0.7 and 0.8 is expected due to the smaller size and thus faster diffusivity of BSA, leading to a higher translocation rate compared to Kap95 (see *Appendix 5—figure 1E*). By contrast, the Nsp1-coated pore shows a significantly higher value of 6.3±1.0), indicating a clear selectivity induced by the Nsp1 coating.

## Selectivity is lost at large-pore diameters

Given recent reports on the dilation of the NPC central channel under stress conditions (*Zimmerli et al., 2021*), we set out to investigate the dependence of the transport selectivity on the pore diameter. Importantly, our approach allows us to measure pore sizes well above 60 nm that were previously inaccessible in conductance-based experiments (*Kowalczyk et al., 2011*; *Ananth et al., 2018*; *Fragasso et al., 2021*). We measured the normalized event rates and apparent selectivity of a total of 46 pores with diameters ranging from 35 nm to 160 nm, which were either open or coated with Nsp1 (*Figure 4*). Note that for pores above 80 nm diameter, it was necessary to reduce the fraction of labeled proteins fivefold in order to avoid too high event rates that would lead to nonlinearity in the event detection due to overlapping events. This dilution is accounted for in the reported normalized event rates. We made sure that all datasets showed the same average molecular brightness (i.e., fluorescence signal per molecule). Datasets with lower average molecular brightness, which could occur due to suboptimal alignment or trapping of air bubbles in the flow cell, were removed from further analysis (see 'Data analysis' for details).

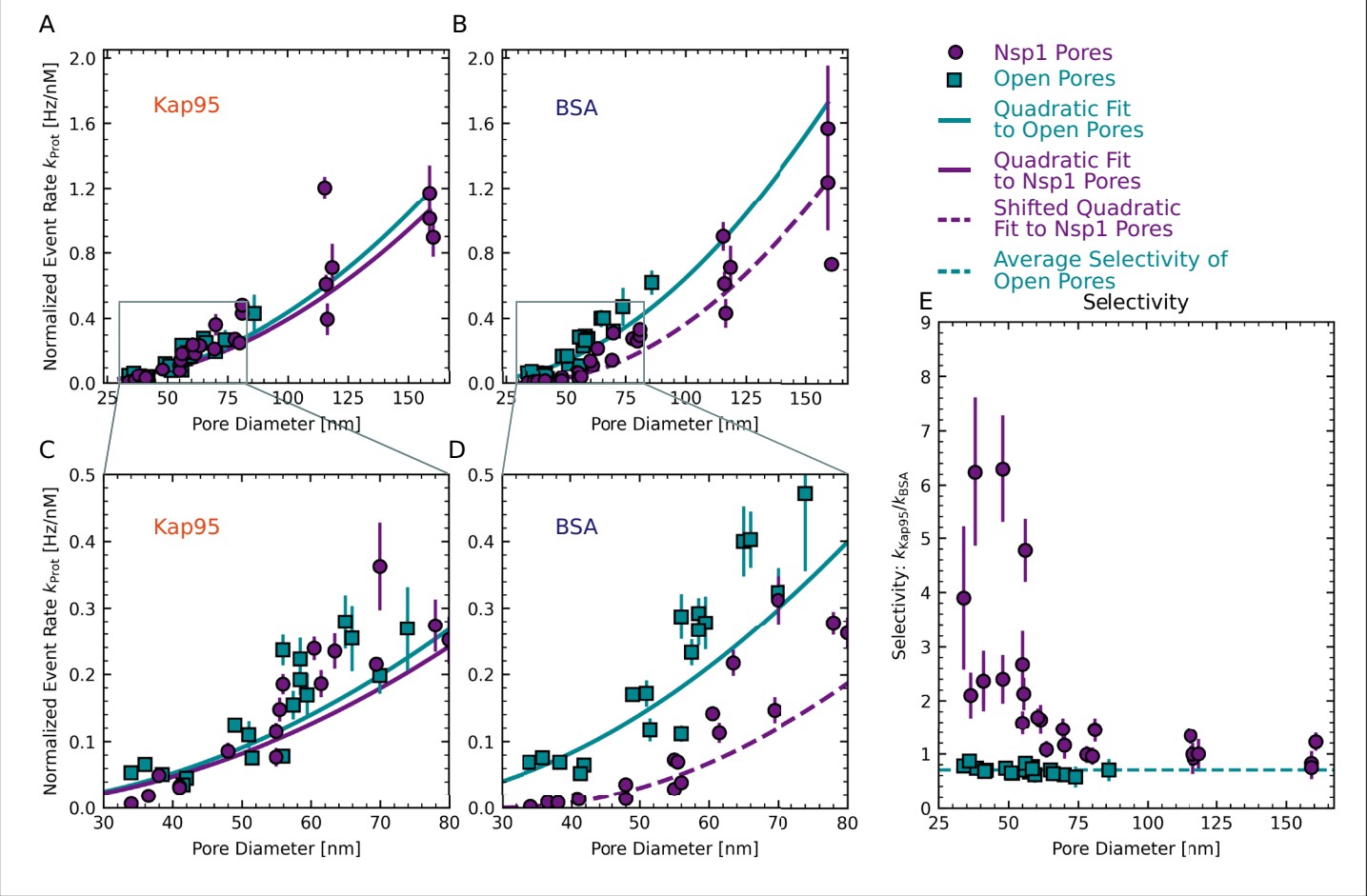

**Figure 4.** Dependence of translocation rates and selectivity on pore diameter. (**A–D**) Concentration-normalized event rate of Kap95 (**A**) and BSA (**B**) as a function of pore diameter for open pores (cyan squares) and Nsp1-coated pores (purple circles). While the normalized event rate for Kap95 did not change significantly between open and Nsp1-coated pores, a clear reduction was observed for BSA, which was most pronounced at small pore sizes. Solid lines are fits to a quadratic function given in *Equation 2*. To model the size dependence of BSA translocations through Nsp1 pores, an offset was introduced that shifts the onset of the quadratic curve to higher diameters (dashed line, *Equation 3*). (**C, D**) Zoom-ins of the indicated regions in (**A ,B**). In (**A–D**), the error bars represent the standard error of the mean of the normalized event rates obtained at different protein concentrations. (**E**) Apparent selectivity versus pore diameter. The data show that selectivity was lost for Nsp1-coated pores with increasing diameter. The average selectivity for open pores of 0.70 ± 0.02 is shown by the dashed cyan line. Error bars indicate the propagated error from the normalized event rates shown in (**A–D**).

We measured protein translocation rates versus pore diameter for transport of both Kap95 and BSA through both open pores and Nsp1-coated pores (see *Figure 4*). According to Fick's law of diffusion, the absolute translocation rate $\kappa_{\mathrm{Prot}}$ is expected to scale linearly with the concentration difference $\Delta c$ between the *cis* and the *trans* side, and with the diffusivity of the probe, $D$. Additionally, it scales linearly with the cross-sectional area of the pore, given by $A = \pi r^2$, and thus it scales quadratically with the pore radius $r$. Further on, it scales inversely with the length of the pore, $L$. This results in a protein-dependent translocation rate,

$$\kappa_{\mathrm{Prot}} = AD\frac{\Delta c}{L} = \pi r^2 D \frac{\Delta c}{L}. \tag{1}$$

As a guide to the eye and for numerical comparison, we fitted the normalized event rates versus pore radius with such a quadratic function,

$$k_{\mathrm{Prot}} = \alpha \left( r - r_{\mathrm{Prot}} \right)^2, \tag{2}$$

where $k_{\mathrm{Prot}}$ is the concentration-normalized event rate, $r$ is the pore radius, and $r_{\mathrm{Prot}}$ is the radius of the protein (Kap95 in this case). Note that this equation accounts for a reduction of the effective cross-sectional pore area due to the fact that a protein has a finite volume and hence its center cannot fully reach the rim of the pore. The only free parameter in the model is the multiplicative scaling factor $\alpha$, which combines the effects of the pore length, concentration gradient, and protein diffusivity, as well as any experimental factors arising from event detection and protein–pore interaction (see Appendix 6 for further details). Note that this simple model does not include the pore diameter-dependent reduction of the diffusivity due to confinement (*Dechadilok and Deen, 2006*), which is discussed in Appendix 5.

For Kap95, the translocation rate through Nsp1-coated pores was reduced by only about 10% compared to open pores (see *Figure 4* and *Appendix 6—figure 2E*). In other words, we observed *no* significant reduction of the normalized event rate for Nsp1-coated pores compared to open pores (*Figure 4A and C*). While this is remarkable, since one might a priori expect a reduction of the rate as an Nsp1-filled pore might obstruct protein transport, this finding signals the optimized properties of Kap95 which interacts in a highly dynamic way with the FG repeats of Nsp1 to facilitate efficient transport. Our finding is in agreement with previous results on pores smaller than 30 nm (*Jovanovic-Talisman et al., 2009*). Interestingly, we also observed no reduced diffusivity of Kap95 molecules on the pore exit side, as quantified by fluorescence correlation spectroscopy (FCS) (*Appendix 7—figure 1A and B*). However, we observed a shortening of the fluorescence lifetime of Kap95–Alexa647 for Nsp1-coated pores compared to open pores, which was not observed for BSA–Alexa488 (*Appendix 7—figure 1C and D*). As the fluorescence lifetime is shortened in the proximity of the metal nanostructure (see *Appendix 2—figure 4*), this indicates that, on the exit side, Kap95 diffuses closer to the pore walls compared to BSA due to interactions with the Nsp1 mesh.

For BSA, the normalized event rates for Nsp1-coated pores were, by contrast, significantly reduced, especially at small-pore diameters (*Figure 4B and D*). More specifically, we observed an approximately tenfold reduction of the event rate for Nsp1-coated pores with a diameter of 35 nm, whereas only a twofold reduction was observed for pores with a diameter of 100 nm, which further decreases for larger pores (*Appendix 6—figure 2E*). Normalized event rates for open pores were well described by the quadratic function *Equation 2*. However, the BSA data for Nsp1-coated pores could not be described using the quadratic dependence with a single scaling factor over the whole range of pore diameters due to the steep increase of the event rate for larger pores (*Appendix 6—figure 2B and D*). We hence introduced an additional fit parameter $b$, which shifts the onset of the curve to higher pore diameters:

$$k_{\mathrm{Prot}} = \alpha \left( r - r_{\mathrm{Prot}} - b \right)^2. \tag{3}$$

The parameter $b$ reduces the effective pore diameter accessible to BSA and can be seen as an estimate of the amount of Nsp1 inside the pore. From the fit, we obtained 11.5 ± 0.4 nm (error is SD estimated from the fit), which indeed is comparable to the height of Nsp1 brushes on flat surfaces (*Wagner et al., 2015*).

Next, we calculated the selectivity ratio $\frac{k_{\mathrm{Kap95}}}{k_{\mathrm{BSA}}}$ to facilitate a direct comparison of pores with different diameters (*Figure 4E*). For open pores, we find a selectivity ratio of 0.70±0.02, independent of the pore diameter. This agrees well with predicted selectivity values of 0.7–0.8 for open pores, based on Fick's law for the different pore diameters (see *Appendix 5—figure 1*). Note that the value of 0.70 deviates from 1 due to the different size of the two proteins, which leads to different diffusion coefficients. For Nsp1-coated pores below 50 nm, selectivity ratios of individual pores ranged between 2.1±0.4 and 6.3±1.0, which is 3–9 times higher than for open pores. This observed selectivity for smaller pores originates predominantly from a blockage of BSA translocations by the Nsp1 mesh, while the Kap translocation rates remain largely unaffected. For pores larger than 60 nm, we see a gradual decrease of the selectivity ratio with increasing pore diameter from 1.7±0.2 to a value of 0.7±0.2 for the largest pores, which approaches the selectivity ratio of open pores. The finite selectivity for large pores suggests that the Nsp1 coating on the pore walls still hindered the translocation of BSA, even after most of the selectivity was lost. The remaining selectivity decreased gradually with the pore diameter because the relative amount of Nsp1 molecules per pore cross-sectional area is reduced, as will be discussed in more detail below.

While the event rates were adequately described by the quadratic function, we observed a large variability between pores of similar size, even for open pores (*Figure 4A–D*). As the variation of the event rate of Kap95 and BSA showed a high degree of correlation (see *Appendix 8—figure 1*), the spread of the selectivity ratio was markedly reduced for open pore experiments. However, the spread remained high for Nsp1-coated pores (*Figure 4E*), which we estimate to be due to pore-to-pore variations of the grafting density. We note that the variability was not due to chip-to-chip variation as a similar spread is also seen within pores that were measured together on one chip in the same experiment (see *Appendix 9—figure 1*).

## Coarse-grained modeling reveals transition of the Nsp1 mesh

To gain a microscopic understanding of the structure and dynamics of the Nsp1 meshwork, we performed CGMD simulations of Nsp1-coated nanopores over a range of diameters and grafting densities. We used an earlier-developed residue-scale model (*Ghavami et al., 2013*; *Ghavami et al., 2014*; *Dekker et al., 2023*) that has been used to study Nsp1-functionalized nanopores (*Ketterer et al., 2018*; *Ananth et al., 2018*; *Fragasso et al., 2022*) and liquid–liquid phase separation of FG-Nups (*Dekker et al., 2023*). Within the microsecond time scale of our CGMD simulations, we found that passive diffusion of BSA through the nanopore channel was a rare event, especially at the smaller-size pores. To obtain statistically meaningful estimates of the translocation rates, we applied a void analysis method developed by *Winogradoff et al., 2022*. Rather than explicitly simulating translocation events of BSA molecules through the Nsp1 mesh, we used this theoretical approach to predict translocation rates based on the equilibrium fluctuations of the Nsp1 mesh in the absence of Kap95 or BSA. Because this method uses the entire simulation volume to characterize the energy barrier that a translocating protein such as BSA would need to overcome, the resulting event rates have much better statistical sampling than can be obtained from brute-force passive diffusion simulations. In brief, we estimated the energy barrier for the translocation of inert probes with the size of BSA from simulations of the Nsp1 mesh alone by quantifying the occurrence of openings ('voids') within the mesh that can accommodate the inert probe without steric clashes. The resulting probability distribution of the occupancy along the pore axis can then be converted into a potential of mean force (PMF) using Boltzmann inversion, which represents the energy barrier for translocation. From this, protein translocation rates were computed by relating the energy barrier to $k_\mathrm{B}T$ using the Arrhenius relation.

We performed simulations of nanopores with a length of 90 nm and diameters in the range of 40–160 nm. Nsp1 proteins were anchored at their C-terminus to the interior wall of the nanopore scaffold in a close-packed triangular distribution (*Figure 5A*).

From the equilibrium trajectories, we computed the PMF and used that to estimate the energy barrier, $\Delta E$, that BSA proteins need to overcome during translocation as the average value of the PMF at the center of the pore (*Appendix 10—figure 2*). Translocation rates were then obtained using the Arrhenius relation:

$$k_\mathrm{BSA} = k_\mathrm{0,BSA} \exp\left(-\Delta E/k_\mathrm{B}T\right), \tag{4}$$

where $k_\mathrm{0,BSA}$ is a proportionality constant to match the experimental event rates for open pores (see *Appendix 10—figure 1*). While $k_\mathrm{0,BSA}$ is the same for open and Nsp1-coated pores of any diameter, $\Delta E$ is calculated for each individual pore using the void analysis method.

As the precise grafting density of Nsp1 proteins within the pore in the experiment is not known accurately, we treated it as the only tunable parameter in the simulations to match the experimental BSA translocation rates. Using a previously estimated grafting density of 1 Nsp1 per 28 nm$^2$ for 20-nm-thick solid-state nanopores (*Ananth et al., 2018*), the calculated BSA translocation rates significantly underestimated the experimental values (not shown here), which prompted us to probe much lower grafting densities between 1 Nsp1 per 200–400 nm$^2$. The best match between the experimental and calculated BSA event rates over the whole range of probed diameters was obtained at a grafting density of 1 Nsp1 per 300 nm$^2$ (*Figure 5D* and *Appendix 10—figure 3A and B*). Note that the estimated grafting density is approximately one order of magnitude lower compared to 20-nm-thick NPC mimics (*Ananth et al., 2018*). Yet, we still obtain comparable protein densities of 50–150 mg/mL because the relatively long channel of the ZMW pores prevents the Nsp1 molecules from spilling out of the pore as in the case for 20-nm-thick SiN$_x$ nanopores. Indeed, these FG-Nup

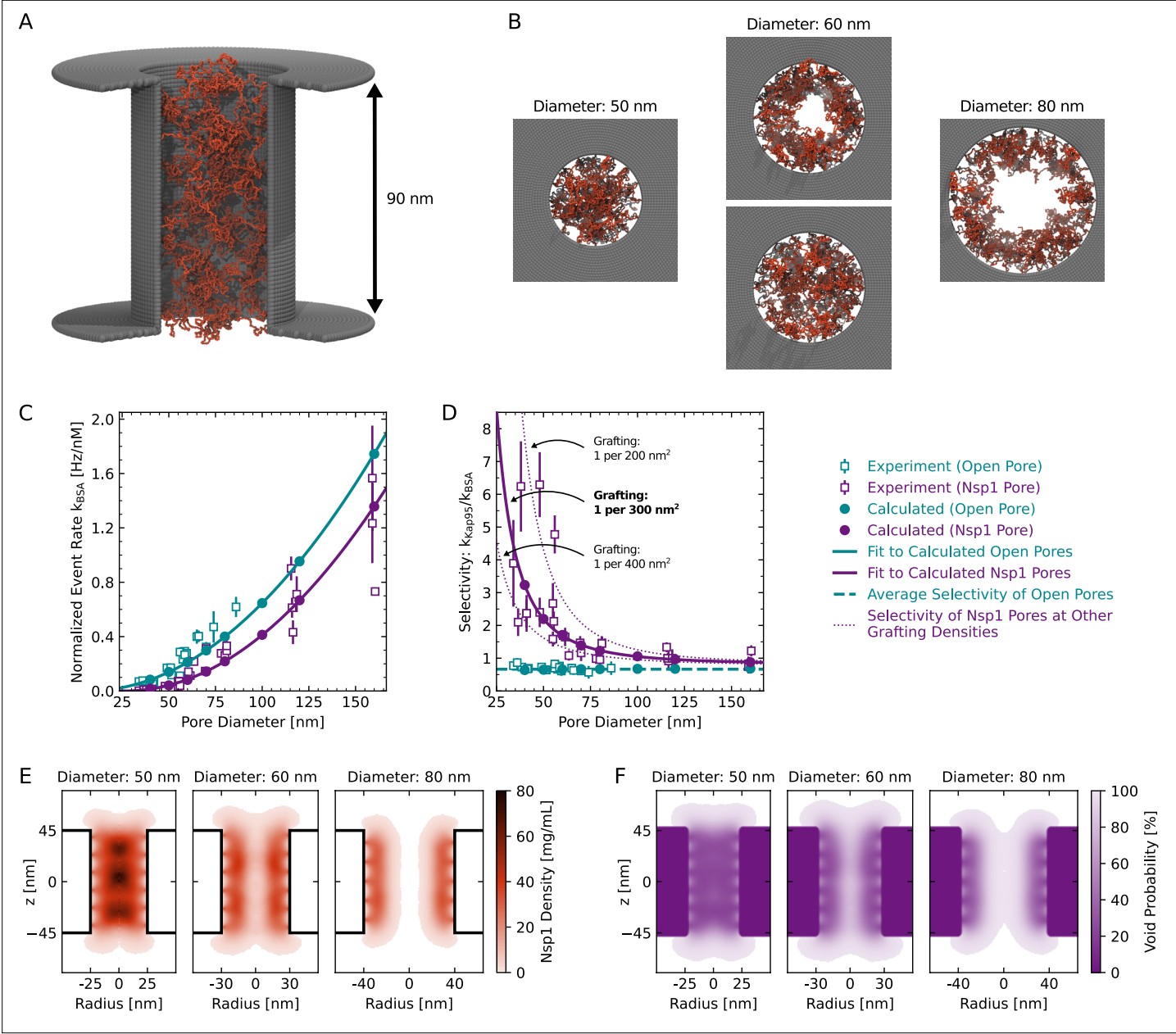

**Figure 5.** Coarse-grained modeling of Nsp1-coated pores. (**A**) One-bead-per-residue representation of an Nsp1-coated nanopore. (**B**) Top views of the Nsp1 meshwork in pores of 50 nm, 60 nm, and 80 nm diameter (for a grafting density of 1 Nsp1 per 300 nm$^2$). For pores with a diameter below 60 nm, the Nsp1 meshwork was closed throughout the entire simulation, while for pores with diameters ≈ 60 nm transient openings were observed in the center of the pore, which were persistent for diameters ≥80 nm. (**C**) Calculated event rate of BSA as a function of pore diameter for open pores (cyan filled dots) and Nsp1-coated pores (purple dots) of 1 Nsp1 per 300 nm$^2$. The calculated event rates are in good agreement with the experimental event rates (open squares). Solid lines are fits to the calculated event rates using the quadratic function given in (2) for open pores and (3) for Nsp1-coated pores. (**D**) Calculated selectivity versus pore diameter, showing that the selectivity is lost for Nsp1-coated pores with increasing diameter. The average selectivity for open pores is shown as a green dashed line. The effect of grafting density on the apparent selectivity is indicated by the purple dotted lines that depict the results for grafting densities of 1 Nsp1 per 200 nm$^2$ and 1 Nsp1 per 400 nm$^2$ (see **Appendix 10—figure 3**). (**E**) Axi-radial and time-averaged protein density distributions inside Nsp1-coated pores of 50 nm, 60 nm, and 80 nm diameter at a grafting density of 1 Nsp1 per 300 nm$^2$. For pore diameters below 60 nm, we observed the highest protein density along the pore axis, while for diameters ≥60 nm, the highest density was found near the pore walls. This observation was valid for each of the probed Nsp1 grafting densities (see **Appendix 10—figure 4**). (**F**) Axi-radial and time-averaged void distributions inside Nsp1-coated pores of 50 nm, 60 nm, and 80 nm diameter at a grafting density of 1 Nsp1 per 300 nm$^2$. The void distributions suggest that for pores of 50 nm diameter there is no preferred pathway for the BSA proteins, while for larger pores the translocations happen mostly along the central axis.

*Figure 5 continued on next page*

*Figure 5 continued*

The online version of this article includes the following video for figure 5:

**Figure 5—video 1.** Video of the top view of the pores presented in *Figure 5B* for 40 ns of the simulation.

https://elifesciences.org/articles/87174/figures#fig5video1

densities are comparable to densities of 30–300 mg/mL found in simulations of full NPCs (*Ghavami et al., 2014*; *Winogradoff et al., 2022*).

The calculated BSA translocation rates continuously increased with increasing diameter and reproduced the trend of the experimental data well (*Figure 5C*). Similar to the experimental data, we observed a delayed onset of the BSA translocation rate for Nsp1-coated pores with an offset of 9.9 ± 0.1 nm, which compares quite well to the experimental value of 11.5 ± 0.4 nm. To compute the selectivity ratio from the BSA translocation rates, we assumed that the Kap95 translocation rate through Nsp1-coated pores is equal to that for open pores, as we did not observe any significant hindrance of Kap95 translocation by the Nsp1 mesh in the experiment (*Figure 4A and C*). The computed selectivity ratios for a grafting density of 1 Nsp1 per 300 nm$^2$ are in good agreement with the experimental values (*Figure 5D*), especially given that we basically only employ a single fitting parameter.

Similar to the experiments, we observed a strong decrease of the selectivity with pore diameter that gradually approached the open pore base line. Additionally, we found that the selectivity ratio is highly sensitive to variations of the grafting density, suggesting that the considerable variation in the experimental data might originate from variations of the grafting density between 1 Nsp1 per 200 nm$^2$ to 1 Nsp1 per 400 nm$^2$ (*Figure 5D*).

The simulations provide a microscopic view of the structure of the Nsp1 mesh within the pore (*Figure 5A, B and E*, *Figure 5—video 1*). At pore diameters below 60 nm, the Nsp1 mesh remained closed over the entire duration of the simulation and the highest protein density was found along the central axis. This is facilitated by the high cohesiveness of the N-terminal domains of Nsp1, which contain a large amount of hydrophobic amino acids (*Dekker et al., 2023*). For pore diameters above 60 nm, however, the highest protein density was found at the pore walls (*Figure 5E*), and we observed the transient formation of a central channel that became persistent at pore diameters above 80 nm (*Figure 5B and E*). Despite the transient appearance of a central channel at pore diameters of 60 nm and the shift of the highest protein density away from the center, the predicted translocation rates increased continuously and followed the quadratic model over the whole range of probed diameters (*Figure 5C*). This finding is in contrast to an instantaneous onset of translocations as one might expect from a static opening of a central channel. As the pore diameter increased, the selectivity ratio became less dependent on the grafting density (*Figure 5D*) because BSA translocated mainly through the wide central channel (*Figure 5B and C*).

To estimate the pathways through which proteins can permeate the Nsp1 mesh, we determined the spatial distribution of the voids that can accommodate BSA, which represents the potential occupancy of BSA in the pore (*Figure 5F*). For diameters below 60 nm, the distribution of the voids was homogeneous across the Nsp1 mesh, suggesting the absence of a preferred pathway. Surprisingly, the potential occupancy of BSA was not markedly reduced along the pore axis compared to the periphery, despite the high protein density in this region. For diameters above 60 nm, the distribution of voids closely followed the time-averaged protein density inside the pores, confirming that most BSA translocations occur through the central channel of the Nsp1 mesh. Although this central channel was not continuously present in the 60 nm pores, the void distribution confirms that most translocations still occurred through the center at this diameter.

## Kap95 modulates the permeability for BSA

In the context of Kap-centric models of nuclear transport, it has recently been shown that depletion of Kaps from the FG-Nup mesh reduces the selectivity of the NPC in cells (*Kalita et al., 2022*). To test whether a similar effect was present in our data, we averaged the concentration-normalized event rates for the two BSA concentrations (of 250 nM and 500 nM) measured at Kap95 concentrations of either 0 nM, 100 nM, or 1000 nM, and denoted them as $k_{BSA,0}$, $k_{BSA,100}$, and $k_{BSA,1000}$ (*Figure 6A*). To minimize the dependence of the rates on the pore diameter, we assessed the relative rates with respect to the event rate in the absence of Kap95, $\frac{k_{BSA,100}}{k_{BSA,0}}$ and $\frac{k_{BSA,1000}}{k_{BSA,0}}$, which serve as a measure for Kap-induced

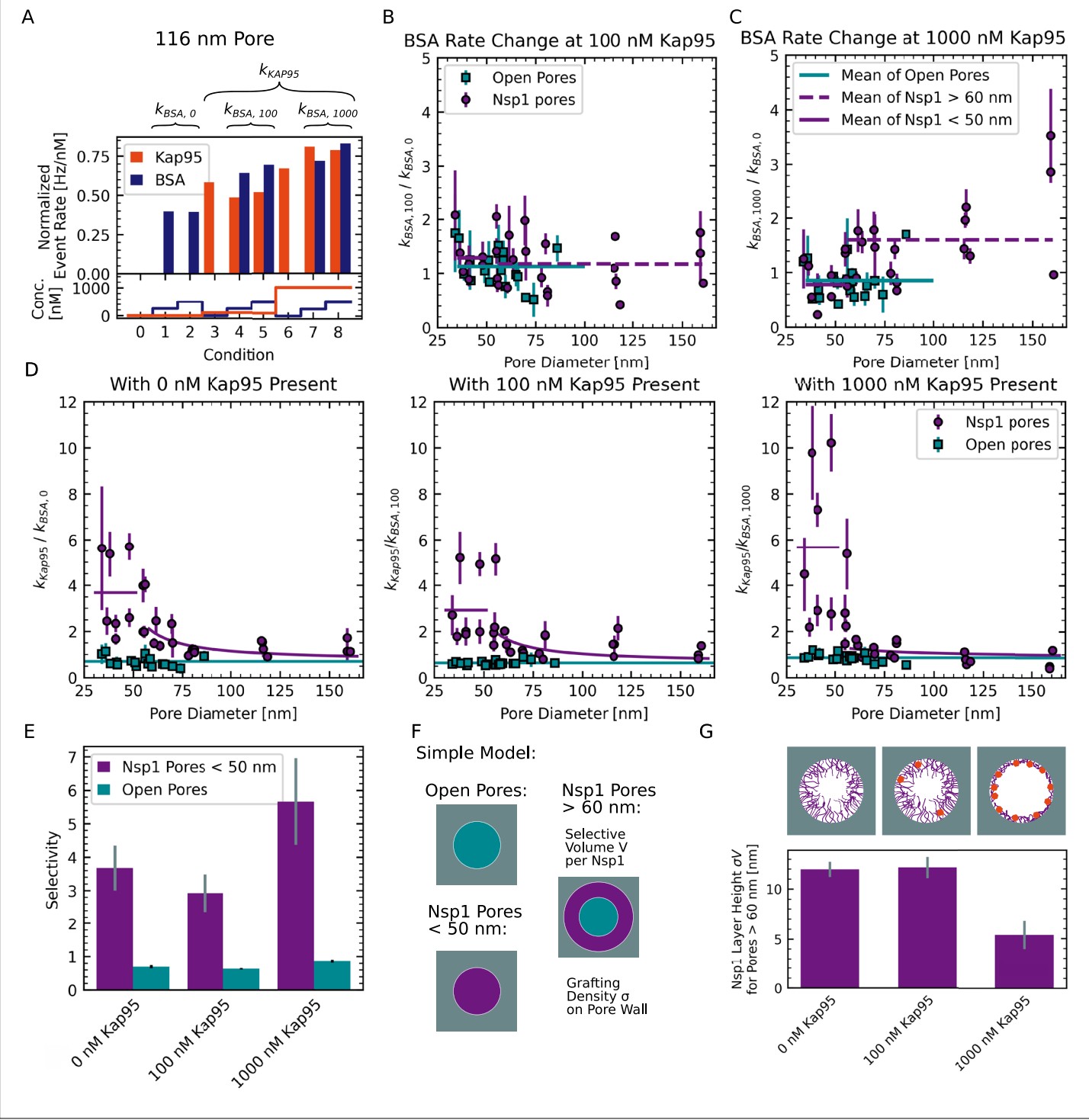

**Figure 6.** BSA permeation depends on Kap95 concentration. (**A**) BSA event rates measured at the different Kap95 concentrations of 0 nM, 100 nM, and 1000 nM were averaged to obtain the Kap95-concentration-dependent event rates $k_{\text{BSA},0}$, $k_{\text{BSA},100}$ and $k_{\text{BSA},1000}$. Data shown were obtained using an Nsp1-coated pore with a diameter of 116 nM. (**B, C**) Plots of the relative change of the BSA event rate measured at 100 nM and 1000 nM Kap95 compared to no Kap95, i.e., $\frac{k_{\text{BSA},100}}{k_{\text{BSA},0}}$ and $\frac{k_{\text{BSA},1000}}{k_{\text{BSA},0}}$, versus pore diameter. A considerable increase of the BSA event rate is observed for large pores in the presence of 1000 nM Kap95. (**D**) Plots of the selectivity ratio, defined as the ratio of the normalized BSA event rate measured at a given Kap95 concentration to the average normalized Kap95 event rate of the pore, i.e., $\frac{k_{\text{BSA},i}}{k_{\text{Kap95}}}$ for i = 0, 100, and 1000, against the pore diameter. The average selectivity ratio of open pores (cyan) and Nsp1-coated pore smaller than 50 nm (purple) are shown as a horizontal line. Data of large Nsp1-coated pores above 600 nm diameter were fitted to the selective area model function (purple lines, see *Equation 5*). (**E**) Average selectivity defined as $\frac{k_{\text{BSA},i}}{k_{\text{Kap95}}}$ for open

*Figure 6 continued on next page*

*Figure 6 continued*

pores and Nsp1-coated pores below 50 nm. Error bars represent standard deviations estimated from fitting horizontal lines. A moderate increase of the selectivity of Nsp1-coated pores is observed at 1000 nM Kap95. (**F**) Simple model for the selectivity of Nsp1-coated pores. We assume large Nsp1-coated pores ≥60 nm to separate into unselective and selective areas with selectivities equivalent to that of open pores or small Nsp1-coated pores, respectively. (**G**) The parameter $\sigma V$ quantifies the thickness of the Nsp1 layer, which decreases in the presence of 1000 nM Kap95. Error bars represent standard deviations estimated from the fits.

changes of the permeability of the pore for BSA (*Figure 6B and C*). In other words, a factor of 1 for this ratio would indicate that BSA transport does not depend on the Kap95 concentration.

We found no significant change of the BSA event rate at 100 nM Kap95 compared to the absence of Kap95 $\left(\frac{k_{\mathrm{BSA},100}}{k_{\mathrm{BSA},0}} = 1.2 \pm 0.1\right)$ (*Figure 6B*), which is similar to the value measured for open pores $\left(\frac{k_{\mathrm{BSA},100}}{k_{\mathrm{BSA},0}} = 1.3 \pm 0.1\right)$. However, at a Kap95 concentration of 1000 nM, we observed a clear increase of the BSA leakage for pores larger than 60 nm $\left(\frac{k_{\mathrm{BSA},1000}}{k_{\mathrm{BSA},0}} = 1.6 \pm 0.2\right)$, and a small reduction of the BSA translocation rates for smaller Nsp1-coated pores $\left(\frac{k_{\mathrm{BSA},1000}}{k_{\mathrm{BSA},0}} = 0.8 \pm 0.1\right)$ similar to the data for open pores $\left(\frac{k_{\mathrm{BSA},1000}}{k_{\mathrm{BSA},0}} = 0.9 \pm 0.1\right)$ (see *Figure 6C*). The increase of the BSA translocation rate at a high concentration of Kap95 seems counterintuitive given that the accumulation of Kap95 in the Nsp1 mesh is expected to obstruct BSA translocations by steric hindrance (*Zilman et al., 2010*). However, we thus observed the opposite effect, where a high concentration of Kap95 appeared to increase the event rates of BSA. Since this effect was not observed for open pores, it must have been induced by a rearrangement of the Nsp1 mesh.

Next, we computed the selectivity ratio based on the normalized event rates of BSA measured at the different Kap95 concentrations (*Figure 6D*). Note that for the calculation of the selectivity ratio, we assumed that the normalized event rate of Kap95 is independent of the Kap95 concentration by using the average normalized event rate over all concentrations, $k_{\mathrm{Kap95}}$, which is an approximation for the more complex behavior discussed in *Appendix 11—figure 1*. For small pores (< 50 nm), we observed a moderate increase of the selectivity ratio at 1000 nM Kap95 compared to lower concentrations of Kap95 (*Figure 6E*). Interestingly, we found an almost twofold increase of the Kap95 translocation rate for small pores at 1000 nM compared to 100 nM, which was absent for large pores (see *Appendix 11—figure 1*). Combined, the reduction of the translocation rate for BSA and the increase for Kap95 resulted in an increase of the apparent selectivity for small pores to 7.1 ±1.6 at a Kap95 concentration of 1000 nM. No effect of the BSA concentration on the translocation rates of Kap95 was observed (*Appendix 11—figure 1*). This suggests that the binding of Kap95 to the FG-Nup mesh increased the selectivity of Nsp1-coated pores, leading to a reduction of the BSA translocation rate, whereas no such effect was seen for open pores. In contrast, larger pores above 60 nm showed a lower selectivity at all Kap95 concentration compared to small Nsp1 pores (*Figure 6D*). While this difference was small for Kap95 concentrations of 0 nM and 100 nM, a considerable reduction to almost the bare pore selectivity was observed for 1000 nM Kap95. After this initial step, we found a gradual decrease of the selectivity with the pore diameter, which is expected because the amount of Nsp1 deposited on the pore wall scales only linearly with the pore diameter while the pores cross-sectional area scales quadratically.

As suggested by the MD simulations, the loss of selectivity for pores above 60 nm may be explained by the opening of a central channel that allows BSA to pass unhindered. To obtain an estimate for the size of such a channel, we devised a simple model to estimate the fraction of the cross-sectional pore area that is occupied by the selective FG-Nup phase (*Figure 6F*). The selective area fraction was obtained from the measured selectivity ratio under the assumption that large Nsp1-coated pores are divided into selective and non-selective areas, where the selective area was assumed to have a selectivity equivalent to the value measured for small pores. We related the selective area fraction to the structural change of the Nsp1 brush by assuming that each Nsp1 molecule renders a certain volume $V$ selective. At a given grafting density on the pore wall of diameter $r$, the selective area fraction $\frac{A_{\mathrm{Nsp1}}}{A}$ is then given by:

$$\frac{A_{\mathrm{Nsp1}}}{A} = \frac{2}{r}\sigma V, \tag{5}$$

where the factor $\sigma V$ was the only fit parameter. The unit of the product $\sigma V$ is a length that can be interpreted as an effective Nsp1 layer thickness at the pore wall under the simplifying assumption of an open channel at large-pore diameters (see Appendix 11 for the details of the model). From the selective area fraction, we computed the apparent selectivity of a pore and estimated the factor $\sigma V$ by fitting it to the experimentally obtained selectivities (*Figure 6D and G*). The estimated effective layer thickness in the absence of Kap95 of ~12 nm agrees well with measurements on flat surfaces (*Wagner et al., 2015*). While no significant change was detected at 100 nM Kap95, we observed a drastic decrease of the estimated layer thickness by a factor of ≈ 2 at 1000 nM Kap95 compared to the absence of Kap95 (*Figure 6G*), suggesting that the Nsp1 layer collapsed as it was occupied by Kap95.

## Discussion

In this study, we present an assay to measure the selectivity of nanopore-based NPC mimics at single-molecule resolution using optical detection. Our assay is based on nanopores drilled into freestanding palladium membranes that act as ZMWs to block the propagation of the excitation light. This allows the localized and selective detection of fluorescently labeled molecules that exit from the pore. We build on our previous work, which established palladium nanopores as a viable tool for single-molecule studies (*Klughammer and Dekker, 2021*) by functionalizing the nanopores with the FG-Nup Nsp1 to build a minimal biomimetic system of the NPC. To this end, we developed a gentle cleaning protocol that leaves the pore shape unaltered while rendering the metal surface susceptible for efficient thiol binding.

Our approach offers several advantages compared to conductance-based nanopore measurements (*Kowalczyk et al., 2011*; *Ananth et al., 2018*; *Fragasso et al., 2021*; *Fragasso et al., 2022*), namely a superior signal-to-noise ratio (see Appendix 3), an excellent specificity for the labeled analyte that enables simultaneous detection of different components, increased throughput by measuring multiple pores in a single experiment, no limitations on pore size, and the potential to add unlabeled components while still investigating their effects. Importantly, our approach also allows us to measure translocations by free diffusion in the absence of an external electric field, which may potentially bias the experiment. Compared to previous optical approaches (*Jovanovic-Talisman et al., 2009*), our assay offers the capability to follow single molecules through individual pores. On the technological side, one could increase the throughput of our approach by moving toward a camera-based readout, which would enable the simultaneous reading of hundreds of pores.

Despite the many benefits of our approach, some differences remain when comparing our biomimetic ZMW pores to the NPC, as, for example, the 90-nm-long channel used here is approximately three times longer (*Kim et al., 2018*), and FG-Nups are grafted to the entire metal surface rather than being limited to the pore walls. In spite of these differences, the in vitro biomimetic approaches remain useful to study key elements of nuclear transport.

To evaluate the performance and sensitivity of our approach, it is relevant to compare the absolute event rates through open pores to theoretical expectations. Compared to Fick's law of diffusion, the absolute event rates for BSA and Kap95 through open pores were underestimated by approximately a factor of 3, as estimated from the scaling factors (*Appendix 5—figure 1*). Additionally, we had previously found that repeated detection of the same molecule leads to an overestimation of the event rate compared to the actual translocation rate by approximately a factor of 2 (*Klughammer and Dekker, 2021*). Combined, this resulted in an ≈ 6 times lower translocation rate compared to what is predicted from Fick's law, potentially due to the hydrodynamic pressure applied in our experiments (*Auger et al., 2014*), protein–pore interactions, and missing of low-signal events by the detection algorithm.

Compared to the study of *Jovanovic-Talisman et al., 2009*, who report values of 0.003 Hz/nM obtained from bulk studies on 30 nm pores, we find an order of magnitude higher event rates of approximately 0.04 Hz/nM for pores of similar size. This discrepancy can be explained by the fact that the functionalized pores used by *Jovanovic-Talisman et al., 2009* were placed at the ends of 6-μm-long channels, which is expected to considerably reduce the translocation rates. When comparing with conductance-based nanopore measurements in 20-nm-thick silicon nitride membranes, previous studies have reported rates of 0.0003 Hz/nM for 30 nm pores (*Fragasso et al., 2021*) and 0.00005Hz/nM for 42 nm to 46 nm pores (*Kowalczyk et al., 2011*). Both these values are many orders of magnitude lower compared to what is predicted by Fick's law, indicating that a large fraction of potential translocation events was previously not detected (*Plesa et al., 2013*). On the other hand,

*Ananth et al., 2018* measured event rates of Kap95 through open 48 nm pores of 0.017 Hz/nM, which is much closer to our measurements of 0.08 Hz/nM compared to other conductance-based studies. The system in conductance-based experiments is more complex, however, with an interplay of diffusion and electro-osmotic and electro-phoretic forces. This illustrates that our optical approach offers a higher detection efficiency of translocation events and provides a better mimic of the diffusion-driven transport through the real NPC, compared to conductance-based nanopore experiments.

We implemented an experimental scheme to assess the selectivity of single Nsp1-coated pores by measuring the concentration-dependent translocation rates of the inert probe BSA and the NTR Kap95. To elucidate the size dependence of the selectivity, we scanned a wide range of pore diameters from 35 nm to 160 nm, which was previously not possible in conductance-based approaches. We found a steep decrease of the selectivity with the pore diameter, where the BSA event rate approaches its open pore value for large-pore diameters. For small diameters, the BSA event rate was decreased tenfold compared to open pores, similar to a fivefold decrease reported for pores smaller than 30 nm by *Jovanovic-Talisman et al., 2009*. For large pores, BSA translocation was only hindered by a factor of 1.4 (*Appendix 6—figure 2E*). On the other hand, translocation rates for Kap95 were unaffected by the Nsp1 coating (reduction factor of 1.1), in agreement with previous reports (*Jovanovic-Talisman et al., 2009*; *Kowalczyk et al., 2011*; *Ananth et al., 2018*; *Fragasso et al., 2021*). Notably, our data reveals a gradual decrease of the selectivity rather than a threshold-like behavior, in line with the results of *Jovanovic-Talisman et al., 2009*, who found a gradual decrease of the selectivity for three distinct pore sizes of 30 nm, 50 nm, and 100 nm.

CGMD simulations allowed us to reproduce the experimental selectivities for a grafting density of $1/(300 \text{ nm}^2)$. While this value is lower compared to previous reports on silicon nitride nanopores (*Fragasso et al., 2021*), we still achieve similar protein densities at the center of the 90-nm-long channel. Similar to the experimental data, the steepest decrease of the selectivity occurred at small diameters. In addition, simulations at varying grafting density revealed that stochastic variation of the surface grafting efficiency is a likely cause for the pore-to-pore variation of the event rates observed in the experiment. The simulations reveal the opening of a central channel in the Nsp1 mesh for pore diameters above 60 nm. However, the event rates predicted by the void analysis showed a continuous decrease of the selectivity with pore diameter even before a central channel was observed in the Nsp1 mesh. This suggests that the formation of a stable transport conduit is not required for the permeation of BSA through the Nsp1 mesh, but may also occur efficiently through transient openings. The absence of a discrete step in the experimental selectivity at a specific pore diameter is thus at odds with a static picture of either an extended or collapsed Nsp1-mesh but rather supports a dynamic transition between both states for intermediate pore diameters. For larger pore diameters, a central channel opened and only a peripheral ring was occupied by Nsp1. This occurred due to the following factors. First, the increased entropic cost of extension for the Nsp1 molecules to interact across the pore. Second, the amount of Nsp1 molecules per pore volume decreases with pore diameter as the volume increases quadratically while the number of molecules increases linearly. Last, while increasing the diameter, also the curvature decreases, which reduces the lateral constraint of neighboring Nsp1.

We found experimentally that the main loss of selectivity falls in a size range between 40 nm and 60 nm. Intriguingly, it has recently been reported that the inner ring diameter of the NPC can be significantly larger in situ at 60 nm compared to 40 nm for isolated NPCs (*Akey et al., 2022*). Furthermore, NPC dilation is modulated in cellulo in response to stress conditions such as energy depletion or osmotic shock (*Zimmerli et al., 2021*). This suggests that the dilation of the NPC might be a way for the cell to tune the permeability of the NPC under stress to increase the selectivity at the cost of lower transport rate.

In light of Kap-centric models of nuclear transport (*Lim et al., 2015*), we also tested the influence of the Kap95 concentration on the selectivity and permeability of Nsp1-coated pores. Small pores below 50 nm showed a moderate increase in selectivity at a Kap95 concentration of 1 µM compared to the absence of Kap95, caused by a slight reduction of the permeability of the pores for BSA. A comparable effect was described by *Jovanovic-Talisman et al., 2009*, who found that the BSA flux through 30 nm pores halved when 2µM of the nuclear transport factor NTF2 was present due to increased competition for the unoccupied space within the pore in the presence of transporters (*Zilman et al., 2010*). In contrast, we observed an almost twofold increase of the BSA translocation rate for large pores above 60 nm by increasing the Kap95 concentration from 0 µM to 1 µM. This

result seems counterintuitive considering that Kap95 occupancy within the pore should pose an additional hindrance for the translocation of BSA. Quantification of this effect, however, showed that the selective area fraction within the pore was significantly reduced at 1μM Kap95, which we attribute to a compaction of the Nsp1 layer. From our data, we found a two fold reduction of the cross-sectional area of the Nsp1 layer inside the pore. A much smaller Kap-induced compaction of 16 nm high Nsp1 brushes on flat surfaces of <10% was observed in SPR measurements at a Kap95 concentration of 100 nM by *Wagner et al., 2015*, where, however, only a fragment of the extended domain of Nsp1 was used. The stark change of the Nsp1 layer thickness seen in our experiments could be a consequence of the pore geometry, which resembles the actual NPC much better than flat surfaces. Here, our measurements on large pores above 100 nm diameter provide an effective bridge between nanopore studies and surface techniques such as QCM-D and SPR. Note that the reported height change of the Nsp1 brushes serves as an approximate quantification due to the high pore-to-pore variability in the dataset. Additionally, we acknowledge that 1 μM of Kap95 is still considerably below physiological Kap95 concentrations of around 4 μM (*Kalita et al., 2022*), so it is hard to relate the effect we observed to the physiological NPC.

We have limited this study to a case study with single combination of FG-Nup (Nsp1) and NTR (Kap95), but our approach could easily be expanded to other FG-Nups, transporters, or control proteins. The NPC contains two main classes of FG-Nups that differ in their cohesiveness, amino acid composition and localization within the central channel. For instance, FxFG-type FG-Nups, such as Nsp1, contain mainly FxFG repeats and have a high content of charged amino acids in its extended domain. Consequently, they are more extended and do not phase separate (*Yamada et al., 2010*; *Dekker et al., 2023*) but instead form percolating hydrogels at high concentration (*Frey et al., 2006*). These FxFG-Nups are predominantly anchored on the nuclear and cytosolic side of the NPC, with Nsp1 being an exception that is also located in the center (*Kim et al., 2018*). On the other hand, GLFG-type FG-Nups contain a low amount of charged amino acids and, as a result, are more cohesive and prone to phase separation (*Schmidt and Görlich, 2015*; *Dekker et al., 2023*). They are localized mainly at the central channel of the NPC, where they might be necessary to form the selective barrier (*Strawn et al., 2004*; *Adams et al., 2015*). While we observed moderate selectivity ratios of 2–6 for Nsp1, we expect that more cohesive GLFG-type FG-Nups, such as Nup100 in yeast or Nup98 in humans, would form a tighter, more selective barrier with lower permeability for BSA. While we did not observe a significant obstruction of the diffusion of Kap95 through the Nsp1 mesh, the dense FG-Nup phase formed by GLFG-type FG-Nups could pose a tighter barrier for Kap95 diffusion. Recent efforts have also focused on designing FG-Nups with desired properties from the bottom-up, where our assay could provide important information on the relation between protein properties and transport selectivity and kinetics (*Fragasso et al., 2021*; *Ng et al., 2021*; *Ng et al., 2022*).

Finally, our approach could be used to study the full systems for protein import or export with all required cofactors, including, for example, the Ran system that provides directionality to molecular transport across the nuclear envelope (*Görlich and Kutay, 1999*). In particular, by using specific labeling coupled with multicolor detection, it will be possible to simultaneously follow different components of the transport machinery, providing direct mechanistic insights into important steps of the transport cycle, such as cargo release or transport factor recycling. An open question in the field also regards how large cargoes such as mRNA (*De Magistris, 2021*) or even viral particles (*Burdick et al., 2020*; *Yang et al., 2023*; *Shen et al., 2023a*) can pass through the NPC, which could be readily tested with our assay. We envision that NPC mimics based on metal nanopores will continue to provide important answers to key questions on the mechanism of nucleocytoplasmic transport.

## Materials and methods
### Fabrication of freestanding Pd ZMWs
Fabrication of nanopores in freestanding palladium membranes was performed as shown in *Figure 1* based on the procedures described in *Klughammer and Dekker, 2021* with minor modifications. Freestanding 20 nm SiN$_x$ membranes were manufactured as described in *Janssen et al., 2012* and cleaned using oxygen plasma for 2 min at 100 W at an oxygen flow rate of 200 mL/min using a PVA Tepla 300 plasma cleaner. As an adhesion layer, 3 nm titanium was deposited onto the SiN$_x$ membrane at 0.05 nm/s under a base pressure of $3 \times 10^{-6}$ Torr in a Temescal FC2000 e-gun evaporator, immediately

followed by a 100 nm layer of Pd at 0.1 nm/s to 0.2 nm/s with a base pressure below $2 \times 10^{-6}$ Torr without venting the chamber in between.

The SiN$_x$ and Ti layers were removed by dry etching using CHF$_3$ at a flow of 50 SCCM and O$_2$ at a flow of 2.5 SCCM for 10 min at 52 W power in a Sentech Etchlab 200 plasma etcher, resulting in an effective chamber pressure of 1.2 Pa. To ensure that the SiN$_x$ and Ti layers are completely removed, the etch time was chosen to partly etch into the Pd layer, resulting in a palladium membrane thickness of 90 nm after etching, estimated by cutting a slit into the palladium membrane using FIB milling and measuring the resulting wall thickness using an SEM on a FEI Helios G4 CX microscope. On the same FIB/SEM, we developed a protocol to reproducibly mill circular nanopores into Pd membranes: After adjusting the eucentric height, the ion column for a 1.1 pA Ga beam at 30 kV acceleration voltage was automatically aligned on a reference sample of gold sputtered on carbon.

A test pore was then milled and inspected for circularity at high magnification using the immersion mode of the SEM. If the test pore was not circular due to astigmatism of the Ga beam, the ion column alignment was repeated. Linear pore arrays with surrounding markers were then milled on the membrane using an automatic script written in Nanobuilder. Individual pores were made at a distance of at least 9 µm in order to avoid later cross-talk between the individual pores during the experiment. An additional marker pattern without pore was added for the identification of individual pores on the membrane. Subsequently, each membrane was examined on a JEOL JEM-1400 TEM for integrity and the minimum and maximum diameters were determined for each pore from the TEM images. The two diameters typically differed by less than 10 %. Pore sizes stated in this study are the arithmetic means of these two values.

## Chemicals, protein purification, and protein labeling

MUHEG with 90% purity (Sigma-Aldrich) was dissolved at a concentration of 200 mM in degassed ethanol. Aliquots were prepared under nitrogen atmosphere and stored at –20 °C until use. 350 Da thiol-PEG (polyethylene glycol) with more than 95% purity (PG1-TH-350, Nanocs Inc, New York) was aliquoted under nitrogen atmosphere and stored at –20 °C until use. Nsp1 protein (Appendix 12) was kindly provided by the Görlich lab (Göttingen, Germany) in lyophilized form and resuspended in denaturing buffer at 10 µM. For long-term storage, samples were snap-frozen in liquid nitrogen and stored at –80°C. Alexa488-labeled BSA was purchased from Invitrogen, Thermo Fisher. On average, one BSA molecule was labeled with six Alexa488 molecules. It was diluted in PBS to a final concentration of 72.6 µM. The diluted sample was dialyzed on ice using 10K Slide-A-Lyzer Dialysis Cassettes for 3 mL for 24 hr, exchanging the 250 mL PBS dialysis buffer four times until no free fluorophores were detectable in the dialysis buffer in an FCS experiment. The protein solution was snap-frozen in aliquots and stored at –20 °C until use in the experiment. Unlabeled BSA was purchased from Thermo Fisher (Ultra Pure BSA [50 mg/mL], Invitrogen), diluted to 5 mg/mL split into aliquots, snap-frozen, and stored at –20 °C until use in the experiment. Kap95 (Appendix 12) was purified as described previously (*Fragasso et al., 2021*) and C-terminally labeled with AZDye647, which is structurally identical to AlexaFluor647, using sortase-mediated ligation. Sortase labeling was performed following published protocols at 50-fold excess of the dye-triglycine conjugate (Click Chemistry Tools, USA) for 1 hr at room temperature in Tris buffer (50 mM Tris, 150 mM NaCl, pH 7.4) (*Guimaraes et al., 2013*). Unreacted dyes were removed by size-exclusion chromatography on a Superdex S200 column pre-equilibrated with PBS buffer. To fully remove free fluorophores, labeled Kap95 was further dialyzed as described above. We used two separate preparations of the labeled Kap95 with 70 and 62.5% degree of labeling and 16.8 µM and 5.9µM stock concentrations. The stock solutions were split into aliquots and stored at –80°C after snap-freezing. The purity of Nsp1 and Kap95 samples was assessed using sodium dodecyl sulfate-polyacrylamide gel electrophoresis (SDS-PAGE).

## Cleaning and surface grafting of Pd

### Grafting of thiols to Pd surface

In order to make Pd accessible for thiol binding, a cleaning procedure was performed. Two different cleaning methods were used depending on the grafting solution. For ethanol-based grafting solutions, we performed cleaning in hydrogen peroxide; and for PBS-based grafting solutions, we performed cleaning in boiling ethanol.

For MUHEG grafting, chips were mounted in a custom-built Teflon holder, rinsed with DI water, and submersed in >99% isopropylalcohol (Riedel-de Haën, Honeywell Research Chemicals) to remove bubbles. Then, 30% hydrogen peroxide (Honeywell Research Chemicals) was brought to 45 °C in a water bath. The chip was rinsed in DI water and then submersed in the hot $H_2O_2$ for 15 min. MUHEG solution was prepared by diluting stock solutions in absolute ethanol (Riedel-de Haën, Honeywell Research Chemicals) to a final concentration of 250 µM. The solution was sonicated for 5–15 min at 20°C. The chip was taken out of the Teflon holder, washed in DI water, and submersed fully in the MUHEG solution for grafting. During the grafting, the chips were gently shaken overnight at 45 rpm for 11–22 hr at 20 °C. Before mounting, the chip was washed in ethanol for 15 min, dried under a stream of nitrogen, and mounted within minutes after drying.

For Nsp1 grafting, chips were mounted in a Teflon holder and rinsed with DI water. Pure ethanol was heated in a water bath until boiling to 78 °C. The chip was submersed in boiling ethanol for 15 min and rinsed with DI water. Boiling ethanol was proposed to efficiently remove organic residues from silver surfaces (*Majid et al., 2003*). Nsp1 aliquots were diluted to 1 µM in PBS buffer. Tris-(2-carboxyethyl)phosphine, hydrochloride (TCEP, 646547-10X1ML, Supelco, Sigma-Aldrich) was added until a final concentration of 1 mM to reduce potential disulfide bonds. The solution was briefly vortexed and then incubated for 15 min at 20°C. Chips were incubated in the Nsp1 solution for 1 hr while shaking at 450 rpm. The chips were then transferred to a solution containing 2 mM 350 Da PEG in PBS buffer with 5 mM TCEP for 1 hr. Before mounting, the chip was washed in PBS for 15 min by shaking at 450 rpm, and subsequently rinsed with DI water and dried under a flow of nitrogen. The chip was reimmersed in buffer within minutes after drying.

## QCM-D experiments

QSense Analyzer gold-coated quartz QCM-D chips were purchased from Biolin Scientific, Västra Frölunda, Sweden. Similar to the freestanding Pd membranes, a 3 nm titanium layer was deposited onto the Au surface at 0.05 nm/s and a 100 nm layer of Pd was evaporated at 0.1 nm/s with a base pressure below $2 \times 10^{-6}$ Torr in a Temescal FC2000 e-gun evaporator. The cleaned chips were mounted in the flow cell in dried state. The flow cell was filled with buffer or ethanol until a stable base line was detected, before the respective grafting solution was applied. The experiments were conducted at 21 °C. See Appendix 1 for details.

## Experimental setup and measurement

Freestanding Pd membranes were mounted in a modified flow cell as described in *Keyser et al., 2006* and are very comparable to what was used in *Klughammer and Dekker, 2021*. The reservoir was made from a PDMS ring that was pressed onto the membrane chip. The reservoir volume was approximately 3 µL. To avoid cross-contamination, the reservoir ring was discarded after each experiment, while the flow channel on the detection side was reused several times.

After drying, the chips were immediately mounted within minutes such that the Pd membrane faced toward the flow channel and the microscope objective. Nsp1-coated chips were immediately immersed in PBS buffer and contact with ethanol was avoided. MUHEG-coated chips were flushed with 1:1 ethanol:water mixtures to remove air bubbles. The conductance of the chip was measured using Ag/AgCl electrodes and an Axopatch 200B amplifier (Molecular Devices) to check that they were properly wetted. The flow cell was subsequently mounted onto the stage of the confocal microscope. We applied a flow to the channel on the detection side using a Fluigent Microfluidic Flow Control System to prevent the accumulation of fluorophores. We applied 50 hPa of pressure to a vessel directly connected via 1 m of tubing (Masterflex Microbore Transfer Tubing, Tygon ND-100-80) to the flow cell. The outlet of the flow cell was connected to another tubing of the same length. Due to symmetry, this results in an estimated pressure of 25 hPa at the location of the membrane. The flow of buffer was measured to be 0.7 mL hr$^{-1}$. The applied pressure induces a hydrodynamic back flow through the pores against the concentration gradient, which results in an approximate reduction of the detected event rates of 5% compared to when no pressure is applied (*Appendix 3—figure 2*). We ensured that the flow was constant between different experiments such that the relative event rates remain unchanged. Experiments were performed at 21 ± 1 °C in a temperature- and vibration-controlled room.

After positioning the membrane in the field of view, markers were localized using transmitted light and the laser focus was centered on the nanopore between the markers. Data was acquired on a Picoquant Microtime 200 microscope using the Symphotime software. We used a 60× Olympus LUMFLN60XW water immersion objective (NA 1.1), which provides a working distance of 1.5 mm to enable imaging of the mounted chip. Excitation lasers at wavelengths of 640 nm and 485 nm were operated in pulsed interleaved excitation at a repetition frequency of 40 MHz and 10 μW power as measured at the imaging plane. Before each experiment, the collar ring of the objective was aligned by optimizing the molecular brightness of a solution of Alexa488 fluorophores. The emission light was passed through a 50 μm pinhole, split by a dichroic mirror, and filtered by either a 600/75 or 525/50 optical band pass filters (Chroma, USA) before being focused on single-photon avalanche-diode detector (PD5CTC and PD1CTC, Micro Photon Devices, Italy).

For each experiment, the same measurement scheme was followed as shown in *Figure 3* and described in detail in Appendix 4. In this scheme, we continuously increased the Kap95 concentration during the experiment to avoid accumulation of Kap95 in the Nsp1 brushes. Before decreasing the BSA concentration, a wash with 5 % hexane-1-6-diol was performed. For experiments involving pores of diameter larger than 70 nm, the labeled protein was mixed with unlabeled protein at a ratio of 1:4 to avoid that the event rate exceeded the detection limit. During the translocation experiment, the respective dilutions of proteins were prepared with PBS and kept on ice before pipetting them into the reservoir. After the experiment, translocations of free Alexa647 and Alexa488 were measured to exclude that pores were clogged (see *Appendix 3—figure 1*).

## Data analysis

Event rate detection was performed using a custom-written Python script based on several packages (*Hunter, 2007*; *Virtanen et al., 2020*; *McKinney, 2010*; *van der Walt et al., 2011*; *Perez and Granger, 2007*). The analysis pipeline was based on previously published work (*Klughammer and Dekker, 2021*; *Klughammer, 2020*) and is deposited in an open repository (*Klughammer et al., 2023a*). In brief, photon bursts were detected using a change point detection algorithm (*Watkins and Yang, 2005*) that detects discrete changes of the photon statistics from the single-photon arrival times. Background events were discarded based on an empirical criterion. In previous work, we had interpreted subsequent events on the millisecond time scale as reentry events of the same molecule (*Klughammer and Dekker, 2021*). To avoid potential biases, no combination of closely spaced events was performed here. However, we could not exclude that molecules may re-enter the laser focus after translocation. Normalized event rates were calculated by dividing the measured event rate by the degree of labeling and the concentration of the respective protein. A purely statistical uncertainty of the event rate was estimated from the assumption that translocation events follow Poisson statistics.

Over the course of the study, several datasets had to be discarded based on the following criteria. First, we discarded a dataset for which a lower excitation power was used (eight pores). Next, we discarded the data of four pores that showed negligible protein translocations due to clogging. For three full datasets totaling 24 pores, we found a significant reduction in the amount of photons detected per molecule, which biased the event detection (see *Appendix 13—figures 1 and 2* for details). Further, in cases where the normalized Kap95 event rate differed significantly for one out of the three measurements for the same concentration, the condition was removed (17 conditions). When there was doubt about perfect focusing of the lasers for certain pores, these were discarded from further analysis (two pores). Finally, if there was any indication of sticking of proteins to the pore surface, as visible from the FCS curve, these pores were removed (two pores). In total, 46 pores (27 with Nsp1 coating, 19 open pores) with 400 time traces (248 for Nsp1-coated pore, 152 for open pores) from seven individual measurement days and chips were used for the final analysis.

Normalized event rates in *Figures 4 and 5* were fitted using *Equations 2* and *3* by optimizing the functions to the individual translocation datasets, taking into account their statistical errors using the least-squares method (see *Appendix 6—figure 1*).

FCS and lifetime analyses were performed using the PAM software package (*Schrimpf et al., 2018*). For fitting of the FCS curves, the size of the confocal volume was determined from measurements of the free dyes Alexa647 and Alexa488 by fitting a single-component diffusion model with triplet state. The diffusion coefficients at 21 °C were set to 297 μm²/s for Alexa647 and 372 μm²/s for Alexa488, based on the values provided in *Kapusta, 2020*. The axial and lateral sizes of the confocal volume, $\omega_z$

and $\omega_r$, were fixed for further analysis. FCS amplitudes and diffusion coefficients were subsequently fitted for each dataset separately. For Kap95–Alexa647 containing samples, the triplet state was fitted individually for each FCS curve. Fluorescence lifetimes were determined by a single-exponential tail fit of the fluorescence decays, ignoring the first 1160 ps of the decay in order to reduce variations introduced by the instrument response function.

## CGMD simulations

CGMD simulations were performed with the implicit-solvent 1BPA-1.1 model for intrinsically disordered proteins (*Ghavami et al., 2013*; *Ghavami et al., 2014*; *Jafarinia et al., 2020*; *Jafarinia et al., 2022*; *Dekker et al., 2023*). This residue-scale model discriminates between all 20 amino acids through residue-specific hydrophobic, charge, and cation–pi interactions, and accounts for the sequence-specific backbone stiffness (see *Dekker et al., 2023* for a detailed description of the 1BPA-1.1 model). Simulations were performed with the GROMACS (*Van Der Spoel et al., 2005*) molecular dynamics software (version 2019.4) using a time step of 0.02 ps and inverse friction coefficient $\gamma^{-1}$ = 50 ps for the Langevin dynamics integrator. All nanopores were simulated at 294 K and a salt concentration of 150 mM KCl by setting the Debye screening constant κ = 1.27 nm$^{-1}$. Nanopore scaffolds were generated from partly overlapping sterically inert beads with a diameter of 3.0 nm with their centers placed 1.5 nm apart. Nsp1 proteins were then grafted to the scaffold wall at their C-terminal Cys-residue, with the N-terminus of the Nsp1 proteins pointing out of the nanopore occlusion. Nanopore systems were equilibrated for $1.0 \times 10^8$ steps (2 µs), followed by a production run of $2.5 \times 10^8$ steps (5 µs) to generate the equilibrium trajectories.

Axi-radial density maps were obtained from the equilibrium trajectories using the *gmx densmap* utility of GROMACS, where a sample was taken every 5000 steps (100 ps). The 2D number-density maps created using GROMACS (in nm$^{-3}$) were converted to mass densities (in mg/mL) using the average mass of an Nsp1 residue (~100 Da). We note that the obtained densities are slightly lower than observed previously for 20 nm pores (*Ananth et al., 2018*) as there a simplified average residue mass of 120 Da was used.

## Void analysis method and calculation of translocation rates

Protein translocations both for BSA and Kap95 were not explicitly simulated. BSA and Kap95 translocation rates were calculated as described in the following paragraph. As experimentally only a negligible hindrance of Kap95 by Nsp1 was observed, the same calculated translocation rates of Kap95 through open pores were assumed for Nsp1-coated pores as well.

PMF curves for protein translocation across the Nsp1 pores were obtained using the void analysis method of *Winogradoff et al., 2022*. The simulation volume was converted into a 3D grid where each voxel has a side length of 6 Å. For each instantaneous configuration of the Nsp1 mesh, we probed for each voxel whether a spherical probe with the size of the translocating protein (BSA or Kap95, see Appendix 10) could be placed at its center without sterically overlapping with an Nsp1 bead or the pore scaffold. The resulting 3D void map was then converted into a 1D potential occupancy map by calculating the percentage of available voxels for each slice along the pore axis. The potential occupancy function was calculated for every $5 \times 10^4$ steps (1 ns) of the equilibrium trajectory. The trajectory average of the potential occupancy function was converted into an effective PMF curve through Boltzmann inversion, as shown in *Appendix 10—figures 1 and 2* (see *Winogradoff et al., 2022* for a more detailed description of the procedure). The analysis was performed with the codes provided by the paper, where a custom constraint was used for each nanopore to exclude the Pd layer volume from the void analysis. Protein translocation rates were obtained from the PMF barriers, calculated by averaging the PMF over a specified range (*Appendix 10—figures 1 and 2*), using an Arrhenius relation:

$$k = k_0 \exp\left(-\Delta E / k_\mathrm{B} T\right), \tag{6}$$

in which $\Delta E$ is the energy barrier that the translocating protein has to overcome and $k_0$ is a proportionality constant that is obtained by fitting the calculated rates to the experimental event rates for open pores (see Appendix 10). This resulted in two independent scaling factors, $k_{0,\mathrm{BSA}}$ and $k_{0,\mathrm{Kap95}}$, for BSA and Kap95, respectively. We note that the use of protein-specific scaling factors follows from the observation that the void analysis method does not take into account any diffusion properties of

the translocating protein. Nevertheless, the same scaling factor was used for both open and Nsp1-coated pores.

To assess the path that BSA proteins take through the Nsp1 mesh, we determined the time-averaged distribution of the 'voids' in the Nsp1 mesh. This was done by computing the 3D void map for each instantaneous configuration of the Nsp1 meshwork and calculating the simulation average. This time-averaged void map represents the probability for each voxel to accommodate a BSA protein. The 3D void map was then circumferentially averaged around the pore's axis to obtain the 2D $(r, z)$ void map shown in *Figure 5F*.

## Finite-difference time-domain (FDTD) simulations

Three-dimensional FDTD simulations were performed using Lumerical FDTD (ANSYS, Inc, USA). The surrounding medium was modeled as water with a refractive index of 1.33. The refractive index of the 100-nm-thick palladium membrane was modeled according to *Palik, 1998*. For the simulation of the excitation field, the ZMW was illuminated by a pulse from a total-field scattered-field source polarized in the x-direction, set as a plane wave source for widefield excitation and a Gaussian source with a numerical aperture of 1.1 for focused excitation. The simulation box size was $1 \times 1 \times 0.8$ μm³ for widefield excitation. A larger box of $4 \times 4 \times 0.8$ μm³ was required for the Gaussian source to correctly model the focused beam. The electromagnetic field intensity distribution was monitored in the xz and yz planes passing through the center of the pore and in the xy plane at the pore entry with a grid resolution of 5 nm (Appendix 2). To model the dipole emission in the proximity of the ZMW, a dipole emitter was placed at varying z-positions at the center of the pore. For the estimation of the quantum yield, the radiated power was integrated over all sides of the box (see below). For the estimation of the detection efficiency, the emitted power was integrated only on the detection side of the ZMW. To isolate the effect of the ZMW on the distribution of the signal on the two sides of the metal membrane, we did not account for the numerical aperture of the objective lens in the computation of the detection efficiency, which represents an additional loss factor in the experimental system. To model isotropic emission, all reported quantities were averaged over horizontal and vertical orientations of the dipole. The power was only weakly affected by the lateral position of the emitter with respect to the center of the pore (data not shown) (*Levene et al., 2003*). The simulated electric field intensities $|E|^2$ are shown in *Appendix 2—figures 1–3*.

## Estimation of detected signal and fluorescence lifetimes

In the absence of the ZMW, the decay rate of the excited molecule is given by $\gamma^0 = \gamma_r^0 + \gamma_{nr}^0$, where $\gamma_r^0$ and $\gamma_{nr}^0$ are the radiative and non-radiative decay rates. Note that $\gamma_{nr}^0$ accounts only for internal processes that lead to non-radiative relaxation to the ground state and was assumed to be unchanged in the presence of the ZMW. The intrinsic quantum yield is defined as $\Phi_0 = \gamma_r^0/(\gamma_r^0 + \gamma_{nr}^0)$ and was estimated from the measured fluorescence lifetimes $\tau_0$ for BSA–Alexa488 and Kap95–Alexa647 of 2.30 ns and 1.37 ns, respectively, as

$$\Phi_0 = \frac{\tau_0}{\tau_{\text{lit}}} \Phi_{\text{lit}}, \tag{7}$$

where $\tau_{\text{lit}}$ and $\Phi_{\text{lit}}$ are reference values for the free dyes ($\tau_{\text{lit}}$ = 4.0 ns and $\Phi_{\text{lit}}$ = 0.80 for Alexa488, $\tau_{\text{lit}}$ = 1.37 ns and $\Phi_{\text{lit}}$ = 0.33 for Alexa647) (*Sanabria et al., 2020*; *Hellenkamp et al., 2018*). This led to quantum yields of $\Phi_0$ = 0.46 and 0.39 for BSA–Alexa488 and Kap95–Alexa647, respectively. Note that the quantum yield of Alexa647 increased slightly due to steric restriction when attached to the protein, an effect known as protein-induced fluorescence enhancement (*Stennett et al., 2015*). The lower quantum yield for BSA–Alexa488 compared to the literature value is most likely a consequence of dye–dye interactions due to the high degree of labeling of ≈ 6 dye molecules per protein, as specified by the manufacturer.

In the presence of the nanostructure, the radiative decay rate $\gamma_r$ is modified and an additional non-radiative rate $\gamma_{\text{loss}}$ is introduced because part of the power emitted by the dipole is absorbed by the metal nanostructure. The quantum yield $\Phi$ in the presence of the ZMW was given by *Bharadwaj and Novotny, 2007*:

$$\Phi = \frac{\gamma_r/\gamma_r^0}{\gamma_r/\gamma_r^0 + \gamma_{\text{loss}}/\gamma_r^0 + (1 - \Phi_0)/\Phi_0}, \tag{8}$$

where $\gamma_r^0$ and $\gamma_r$ are the radiative rates in the absence and the presence of the ZMW, respectively. The absolute decay rates $\gamma_r$, $\gamma_{\text{loss}}$, and $\gamma_r^0$ cannot be obtained from FDTD simulations. However, relative rates with respect to the radiative rate in the absence of the ZMW, $\gamma_r^0$, can be estimated from the power $P$ radiated by the dipole as (*Kaminski et al., 2007*)

$$\frac{\gamma_r}{\gamma_r^0} = \frac{P_{\text{ff}}}{P_r^0} \text{ and } \frac{\gamma_{\text{loss}}}{\gamma_r^0} = \frac{P_r}{P_r^0} - \frac{P_{\text{ff}}}{P_r^0}, \tag{9}$$

where $P_r$ and $P_r^0$ are the powers radiated by the dipole in the presence and absence of the ZMW, and $P_{\text{ff}}$ is the power that is radiated into the far-field in the presence of the ZMW. The fluorescence lifetime $\tau$ is given by the inverse of the sum of all de-excitation rates and can be obtained from *Equation 8* using the relation $\tau = \Phi/\gamma_r$ as

$$\tau = \frac{1}{\gamma_r + \gamma_{\text{loss}} + \gamma_{nr}^0} = \frac{1/\gamma_r^0}{\gamma_r/\gamma_r^0 + \gamma_{\text{loss}}/\gamma_r^0 + (1 - \Phi_0)/\Phi_0}. \tag{10}$$

Here, the intrinsic radiative rate $\gamma_r^0$ in the numerator was estimated as $\gamma_r^0 = \Phi_{\text{lit}}/\tau_{\text{lit}}$. The detection efficiency $\eta$ was estimated from the ratio of the power radiated toward the lower (detection) side of the ZMW, $P_{\text{ff}}^{z-}$, to the total radiated power:

$$\eta = \frac{P_{\text{ff}}^{z-}}{P_{\text{ff}}}. \tag{11}$$

Finally, the total detected signal as a function of the z-position of the emitter with respect to the ZMW was computed as the product of the excitation intensity $I_{\text{ex}}(z)$, detection efficiency $\eta(z)$, and quantum yield $\Phi(z)$ as

$$S(z) \propto I_{\text{ex}}(z)\eta(z)\Phi(z) \tag{12}$$

and normalized to unity. The radiative and loss rates obtained from the FDTD simulations ($\gamma_r/\gamma_r^0$ and $\gamma_{\text{loss}}/\gamma_r^0$), which are used to compute the quantities $\Phi$, $\tau$, and $\eta$, are given in *Appendix 2—figure 5* as a function of the z-position within the ZMW. Z-profiles of the computed detection efficiency $\eta$, quantum yield $\Phi$, detected signal $S(z)$, and lifetime $\tau$ are shown in *Appendix 2—figure 4*.

Using the signal profile $S(z)$, we compute the signal-averaged fluorescence lifetime $\langle\tau\rangle_S$ as

$$\langle\tau\rangle_S = \frac{\int S(z)\tau(z)dz}{\int S(z)dz}, \tag{13}$$

which agrees well with the experimental fluorescence lifetimes measured in the translocation experiments (*Appendix 2—figure 4*).

## Acknowledgements

We thank Eli van der Sluis, Ashmiani van den Berg, and Angeliki Goutou for support with protein expression and purification. We thank Paola de Magistris, Xin Shi, Sonja Schmid, and Biswajit Pradhan for fruitful discussions. The Nsp1 protein used in this study was a kind gift from Dirk Görlich. Financial support was provided by the NWO program OCENW.GROOT.2019.068, ERC Advanced grant no. 883684, and the NanoFront and BaSyC programs of NWO/OCW. AB acknowledges funding from the European Union's Horizon 2020 research and innovation program under the Marie Skłodowska-Curie Grant agreement no. 101029907.

# Additional information

## Funding

| Funder | Grant reference number | Author |
| --- | --- | --- |
| H2020 European Research Council | 101029907 | Anders Barth |
| Nederlandse Organisatie voor Wetenschappelijk Onderzoek | OCENW.GROOT.2019.068 | Nils Klughammer |
| European Research Council | 883684 | Nils Klughammer |
| Ministerie van Onderwijs, Cultuur en Wetenschap | BaSyC | Nils Klughammer |
| Nederlandse Organisatie voor Wetenschappelijk Onderzoek | BaSyC | Nils Klughammer |

The funders had no role in study design, data collection and interpretation, or the decision to submit the work for publication.

## Author contributions

Nils Klughammer, Conceptualization, Data curation, Software, Formal analysis, Validation, Investigation, Visualization, Methodology, Writing – original draft, Writing – review and editing, Performed nanofabrication, translocation experiments and the data analysis, Developed and optimized the experimental protocol, Designed figures, Wrote the initial draft; Anders Barth, Resources, Software, Formal analysis, Funding acquisition, Validation, Investigation, Visualization, Methodology, Writing – original draft, Writing – review and editing, Developed and optimized the experimental protocol, Performed protein labeling and purification, Designed figures, Wrote the initial draft; Maurice Dekker, Software, Formal analysis, Investigation, Visualization, Methodology, Writing – original draft, Writing – review and editing, Performed coarse-grained molecular dynamics simulations, Designed figures; Alessio Fragasso, Conceptualization, Resources, Methodology, Writing – review and editing, Performed protein labeling and purification; Patrick R Onck, Supervision, Funding acquisition, Writing – review and editing; Cees Dekker, Conceptualization, Supervision, Funding acquisition, Writing – review and editing

## Author ORCIDs

Nils Klughammer (ID) https://orcid.org/0000-0002-8792-2459
Anders Barth (ID) https://orcid.org/0000-0003-3671-3072
Maurice Dekker (ID) https://orcid.org/0000-0001-7096-8576
Patrick R Onck (ID) http://orcid.org/0000-0001-5632-9727
Cees Dekker (ID) https://orcid.org/0000-0001-6273-071X

Reviewer #1 (Public Review): https://doi.org/10.7554/eLife.87174.3.sa1
Reviewer #2 (Public Review): https://doi.org/10.7554/eLife.87174.3.sa2
Author Response https://doi.org/10.7554/eLife.87174.3.sa3

# Additional files

## Supplementary files
• MDAR checklist

## Data availability

All single photon counting data is deposited in the open Photon-HDF5 file format (*Ingargiola et al., 2016*) together with the unprocessed TEM images of nanopores in a repository at https://doi.org/10.4121/22059227. The code for data analysis can be found at: https://doi.org/10.4121/21027850.

The following dataset was generated:

| Author(s) | Year | Dataset title | Dataset URL | Database and Identifier |
|---|---|---|---|---|
| Klughammer N, Barth A, Dekker M | 2023 | Data from paper Diameter Dependence of Transport through 1140 Nuclear Pore Complex Mimics Studied Using Optical Nanopores | https://doi.org/10.4121/22059227 | 4TU.ResearchData, 10.4121/22059227 |

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

## Appendix 1

### Different cleaning methods tested

We encountered that after fabrication the Pd surface inside the pores needed thorough cleaning to be accessible for thiol binding. During the study, we tested cleaning with 100 W oxygen plasma, 25 and 5 % nitric acid, 1 hr and 15 min RCA1, 15 min sulfuric acid piranha solution, 1 M potassium hydroxide solution, 10 min fuming sulfuric acid, 1 min sulfuric acid, 15 min hydrogen peroxide solution at 45 °C and 20 °C, 3 and 3 % hydrochloric acid, 1 min gold etch (KI and $I_2$), 1:20 and 1:40 diluted commercial copper etch (Sigma-Aldrich, comparable composition as commercial Pd etch), 30 % ammonia, 2 min sodium hydroxide solution, and boiling ethanol. Cleaning methods were evaluated based on two criteria. First, nanopores should remain intact, that is, neither closed nor grew, and surfaces needed to be competent for thiol binding as monitored by QCM-D. Only hydrogen peroxide and ethanol boiling fulfilled both these conditions.

### QCM-D experiments

In order to test how well MUHEG passivates a hydrogenperoxide-cleaned Pd surface, we coated two QCM-D chips with MUHEG and monitored their frequency and dissipation shifts. We found a resulting frequency shift of about 15 Hz when switching back to ethanol after incubating the chip with 250 μM of MUHEG in ethanol, indicating that mass had attached to the QCM-D chip and that the cleaning procedure thus allowed the thiols to bind to the surface (*Appendix 1—figure 1*).

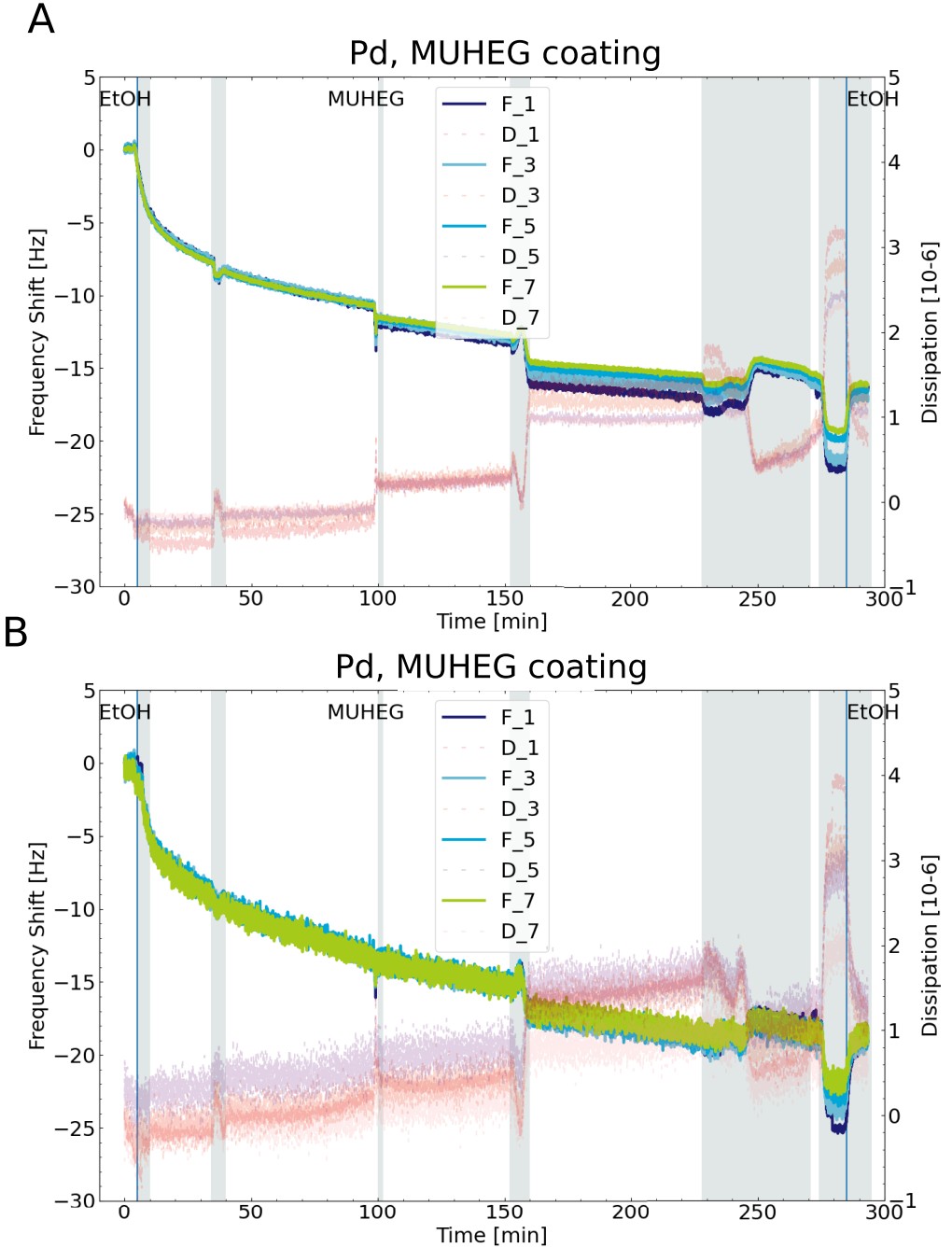

**Appendix 1—figure 1.** (1-mercaptoundec-11-yl)hexa(ethylene glycol) (MUHEG) grafting established on quartz crystal microbalance with dissipation monitoring (QCM-D). (**A, B**) The frequency (F1–F7) and dissipation (D1–D7) response of the different harmonics (numbers) for two QCM-D sensors versus time upon grafting of 250 µM MUHEG in ethanol (see 'Cleaning and surface grafting of Pd'). The blue vertical lines show when the solution was switched in the flow cell. The gray shaded regions show the time when the solution was flowed through the flow cell. Upon MUHEG binding, a decreasing frequency can be observed, which shows that mass attaches to the sensor's surfaces.

Next, we switched the buffers to 150 mM of KCl in 1×TE and flushed 500 nM of BSA followed by TE buffer and 500 nM of Kap95. After switching back to TE buffer, we found a frequency shift of less than 5 Hz, which was much less than expected for untreated surfaces (*Appendix 1—figure 2*). This result indicates that the surface had been passivated against proteins adhering to the surface. While

this was a promising result, we knew from previous experiments that even if a surface is passivated well enough on QCM-D, we might still see considerable effects of protein sticking in translocation experiments. Thus we determined a real proof of good passivation to be a linear relationship of concentration versus event rate in open pore experiments.

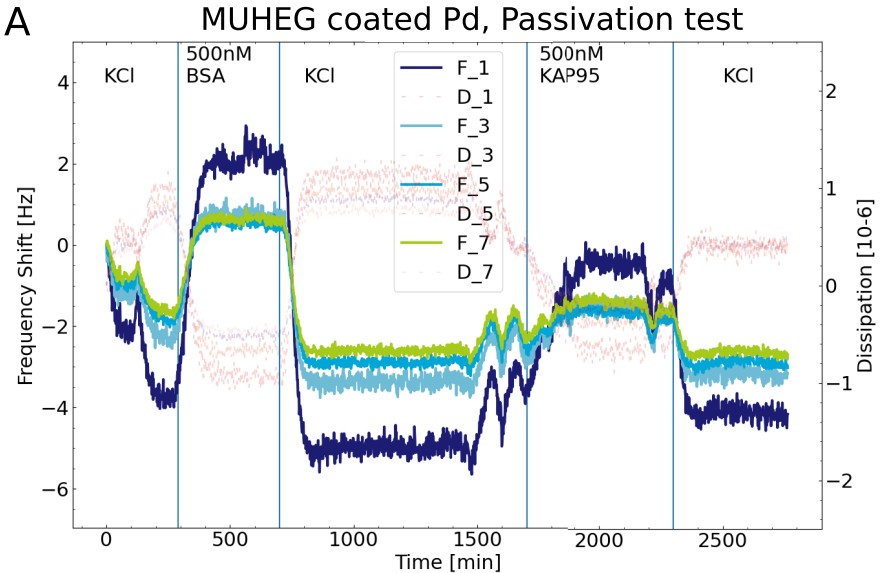

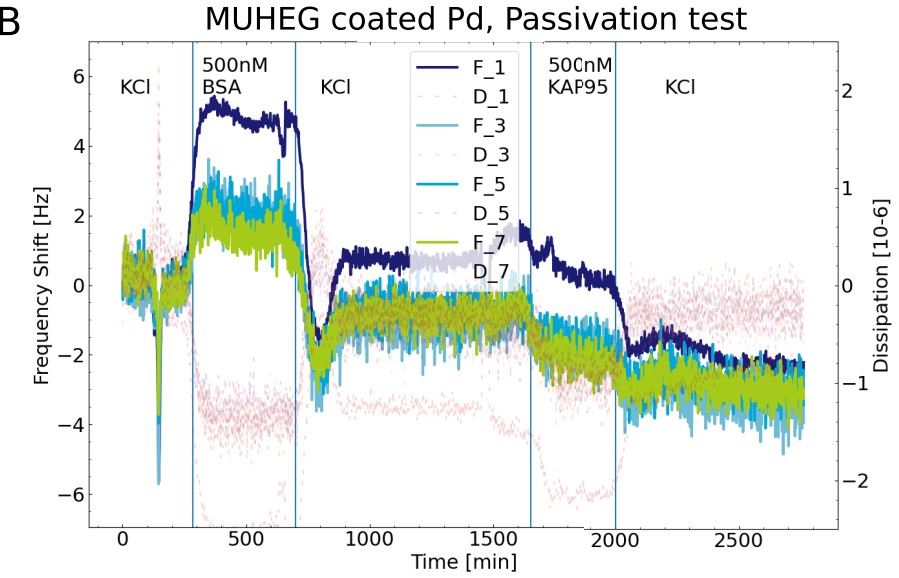

**Appendix 1—figure 2.** (1-mercaptoundec-11-yl)hexa(ethylene glycol) (MUHEG) passivation established on quartz crystal microbalance with dissipation monitoring (QCM-D). (**A, B**) Same chips and setup as in *Appendix 1—figure 1* The frequency response of flushing 500 nM of BSA and 500 nM of Kap95 over the MUHEG passivated surface of the QCM-D chips only shows a minor frequency shift of less than 5 Hz (KCl level before and after flushing the proteins). This suggests that the Pd surface can be effectively passivated against adhering proteins by a MUHEG coating.

During our study, we noted that Pd nanopores could close upon incubation in PBS after peroxide cleaning. This effect was not observed for MUHEG-coated pores that were incubated for extended time in ethanol after the peroxide cleaning. Thus we developed another cleaning strategy for Pd surfaces based on ethanol boiling, inspired by *Majid et al., 2003*. In order to test the capability to

bind Nsp1 after this treatment, we performed a QCM-D experiment. The cleaned chip was flushed with 150 mM KCl + 1×TE buffer, then 1.14 µM of Nsp1 and then TE buffer again. The resulting frequency shift due to Nsp1 attachment was determined to be approximately 60 Hz (*Appendix 1—figure 3*). This was comparable to previous experiments on piranha cleaned gold QCM-D chips. Since deducing a grafting density from QCM-D experiments was difficult, this serves more as a qualitative result and the actual test of sufficient grafting needed to be made in the nanopore.

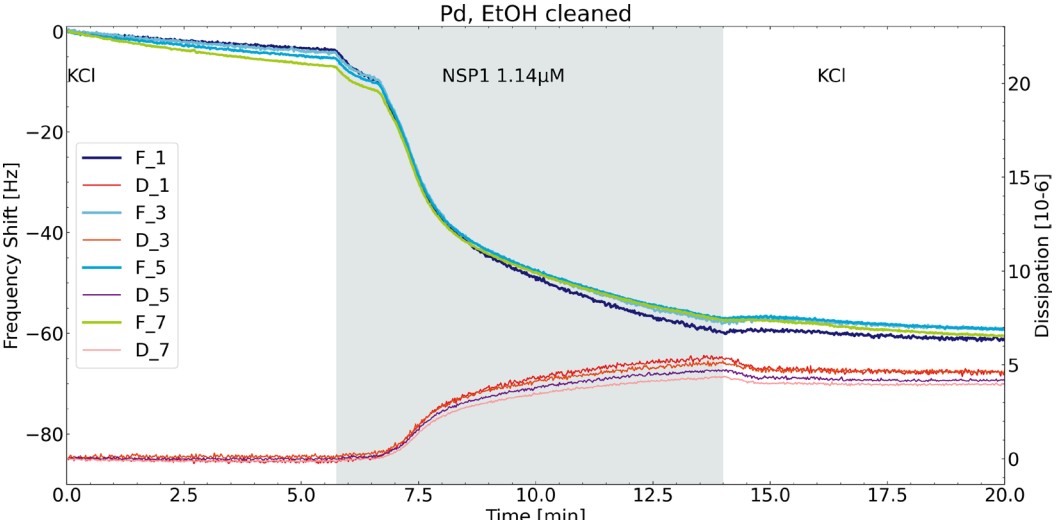

**Appendix 1—figure 3.** Nsp1 binding on quartz crystal microbalance with dissipation monitoring (QCM-D). The frequency (F1–F7) and dissipation (D1–D7) response of the different harmonics (numbers) for a Pd-coated QCM-D sensors versus time upon grafting of 1.14 µM Nsp1 in PBS. The QCM-D chip was cleaned by boiling ethanol (see 'Cleaning and surface grafting of Pd'). The frequency shift from before the Nsp1 coating to after was approximately 60 Hz, which shows an acceptable coating efficiency. Nsp1 was flushed in the gray shaded area.

## Appendix 2

## FDTD simulations

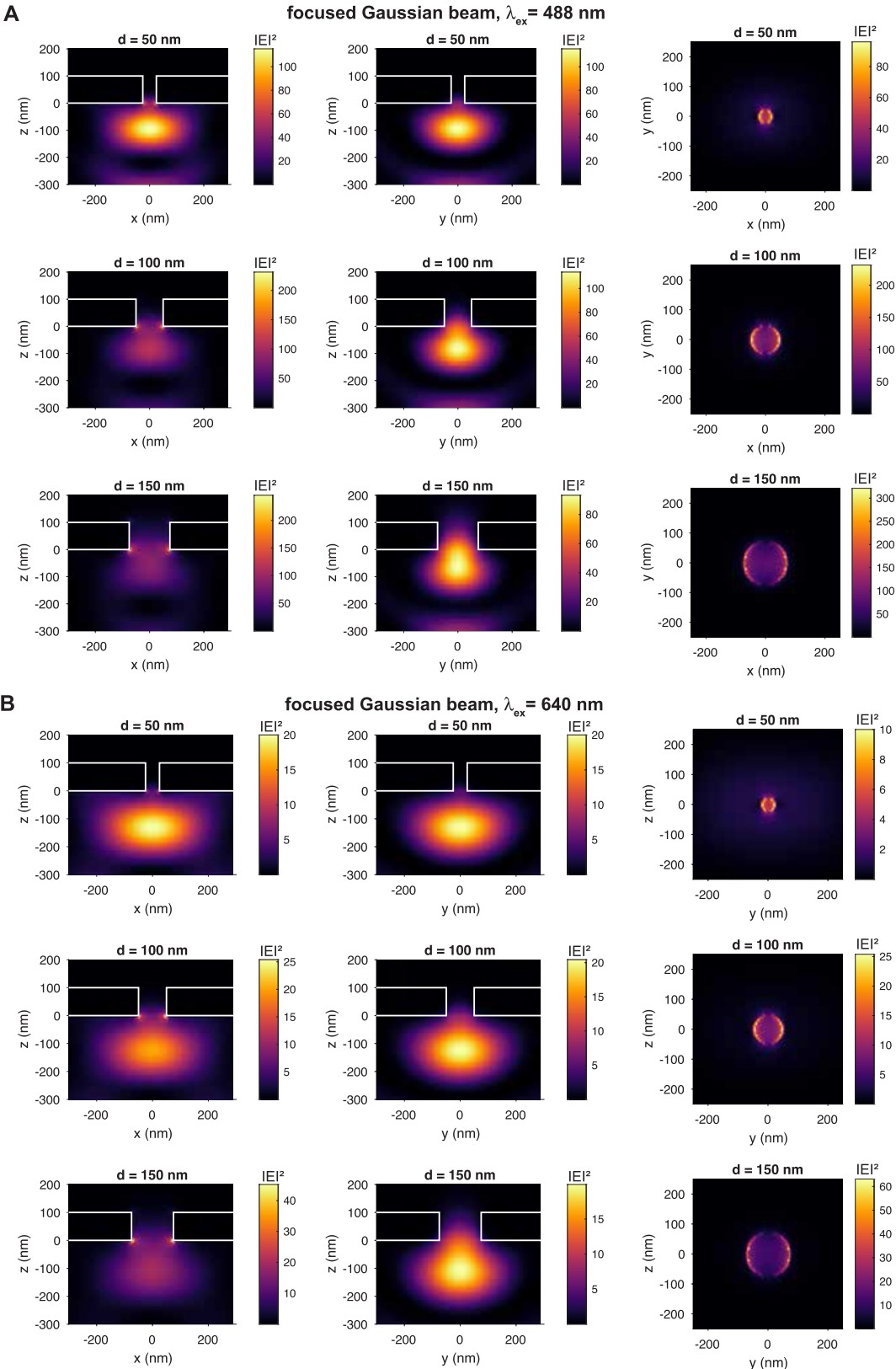

**Appendix 2—figure 1.** Three-dimensional finite-difference time-domain simulations of the electric field intensity distribution |E|$^2$ in units of $V^2/m^2$ in the proximity of the zero-mode waveguides (ZMW) for excitation by a diffraction-limited focused Gaussian beam with wavelengths of 488 nm (**A**) and 640 nm (**B**). The lower side of the 100-nm-thick palladium membrane is placed at $z$ = 0 nm. The source is located at the bottom and the electric field is polarized in the x-direction. The electric field intensity distributions are shown for pores with a diameter of 50 nm, 100 nm, and 200 nm (from top to bottom) in the xz (left) and yz (middle) planes passing through the center of the pore, and the xy (right) plane at the entrance to the pore at $z$ = 0 nm. See **Appendix 2—figure 2** for a zoomed-out representation of the intensity distribution.

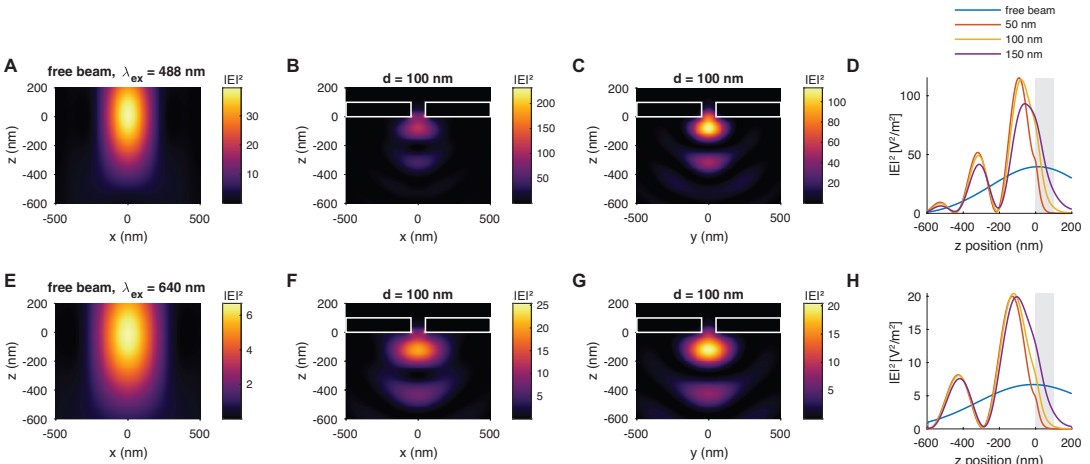

**Appendix 2—figure 2.** Three-dimensional finite-difference time-domain simulations of the electric field intensity distribution of the excitation spot in the presence of a freestanding zero-mode waveguide (ZMW) (compare **Appendix 2—figure 1**), at excitation wavelengths of 488 nm (**A–D**) and 640 nm (**E–H**). Shown are the intensity distributions of the focused Gaussian beam in the absence (**A, E**) and presence (**B–C, F–G**) of the freestanding palladium ZMW. (**D, H**) The z-profiles of the intensity distribution along the center of the pore. The position of the palladium membrane is indicated as a gray shaded area.

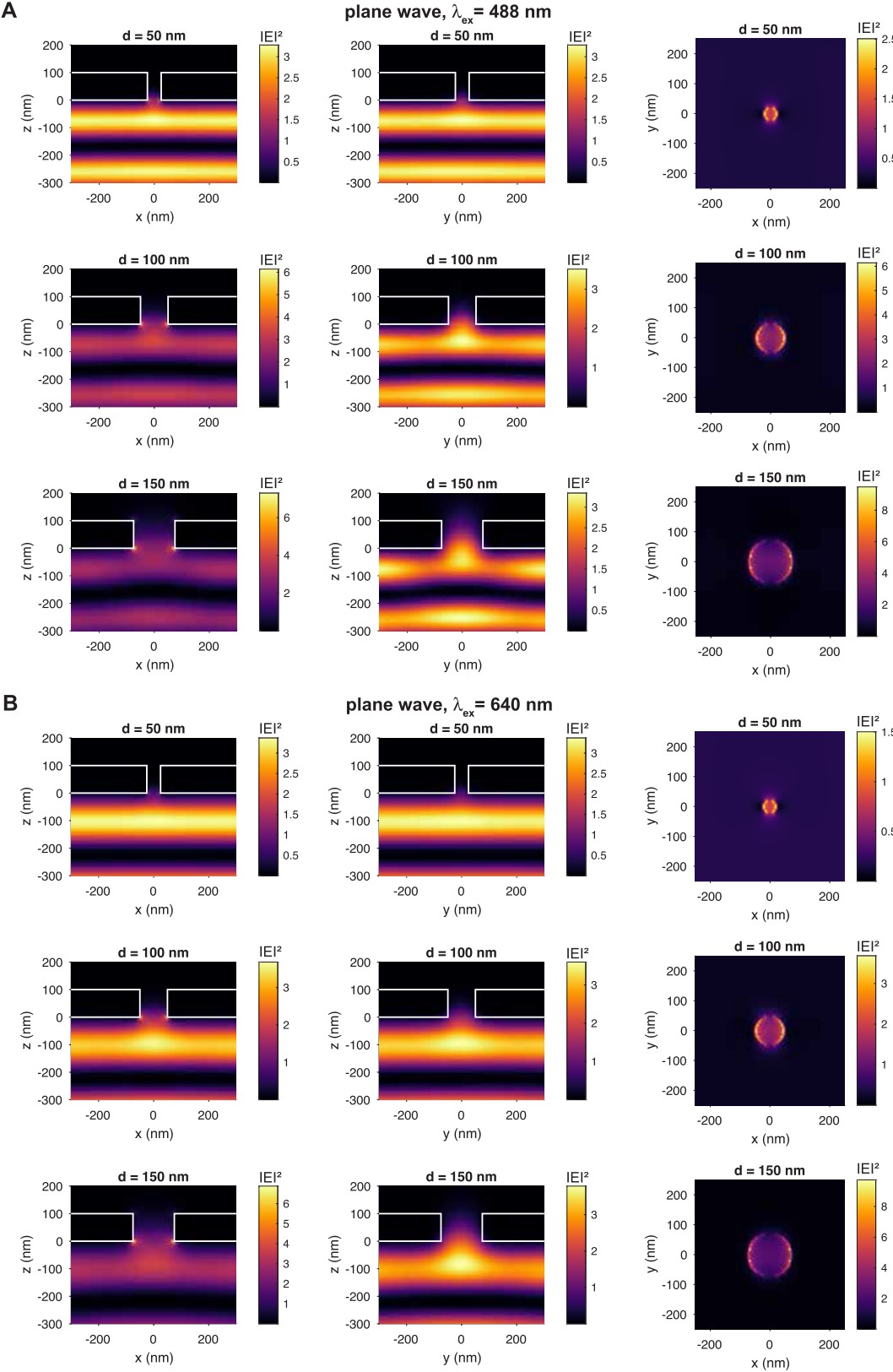

**Appendix 2—figure 3.** Three-dimensional finite-difference time-domain simulations of the electric field intensity distribution |E|$^2$ in units of V²/m² in the proximity of the zero-mode waveguide (ZMW) for excitation by a plane wave with wavelengths of 488 nm (**A**) and 640 nm (**B**). The lower side of the 100-nm-thick palladium membrane
*Appendix 2—figure 3 continued on next page*

*Appendix 2—figure 3 continued*
is place at $z = 0$ nm. The source is located at the bottom and the electric field is polarized in the x-direction. The electric field intensity distributions are shown for pores with a diameter of 50, 100, and 200 nm (from top to bottom) in the xz (left) and yz (middle) planes passing through the center of the pore, and the xy (right) plane at the entrance to the pore at $z = 0$ nm.

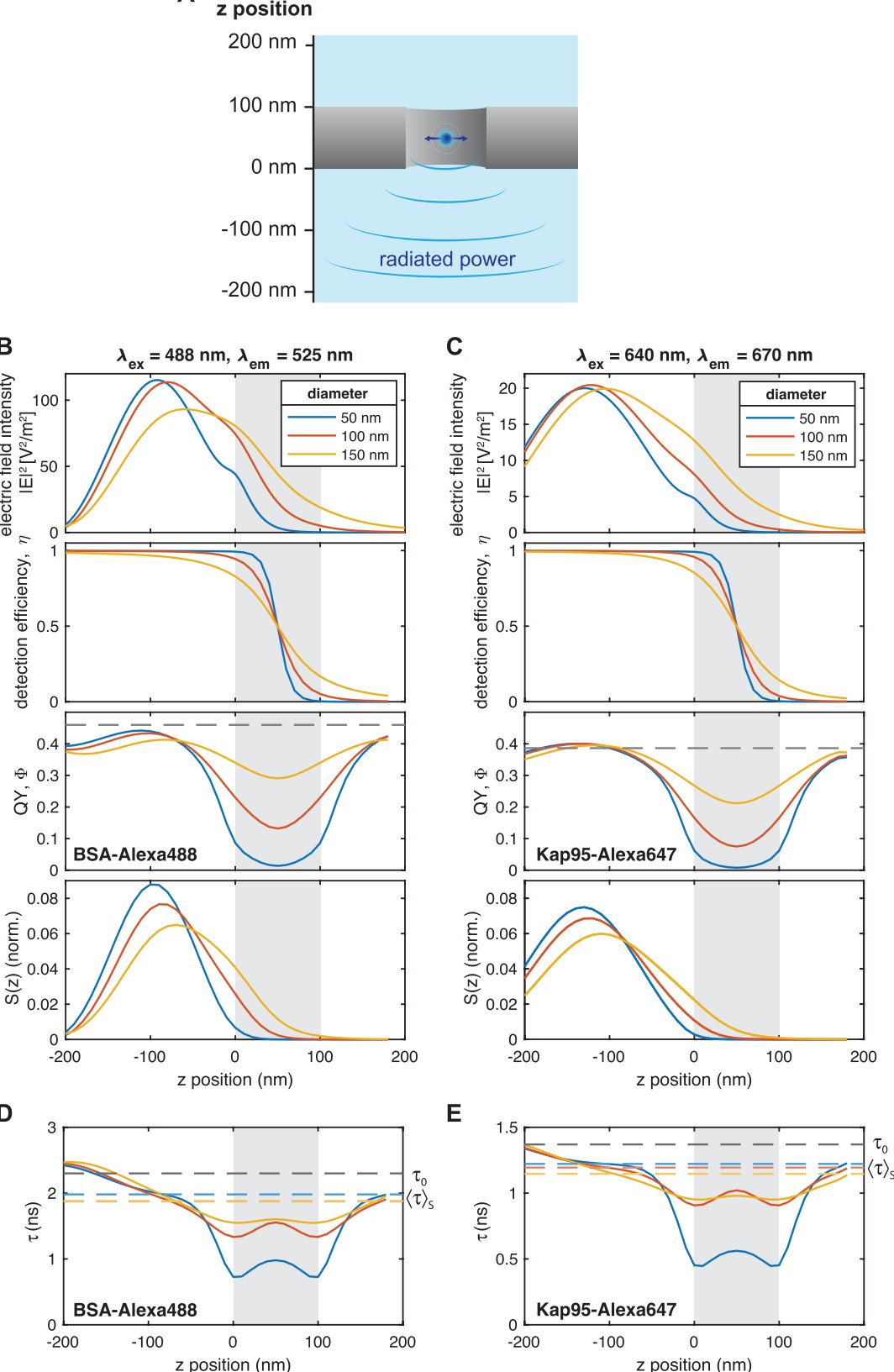

**Appendix 2—figure 4.** Finite-difference time-domain simulations of the dipole emission in the proximity of the freestanding zero-mode waveguide (ZMW). (**A**) A scheme of the simulation setup. The dipole is placed in the
*Appendix 2—figure 4 continued on next page*

*Appendix 2—figure 4 continued*

center of the pore in the xy plane at varying z-positions. The detected signal is monitored on the detection (i.e., lower) side. (**B, C**) From top to bottom: the z-profiles of the excitation probability, the detection efficiency $\eta$, the emitter quantum yield $\Phi$, and the total detected signal along the center of the nanopore are shown for the blue (**B**, $\lambda_{ex}$ = 488 nm, $\lambda_{em}$ = 525 nm) and red (**C**, $\lambda_{ex}$ = 640 nm, $\lambda_{em}$ = 670 nm) channels. The total detected signal $S(z)$ is defined as the product of the excitation intensity, detection efficiency, and quantum yield. (**D, E**) Predicted fluorescence lifetimes $\tau$ of BSA–Alexa488 and Kap95–Alexa647. The position of the palladium membrane is indicated as a gray shaded area. The weighted averages of the fluorescence lifetime based on the detected signal $S(z)$, $\langle \tau \rangle_S$ are shown as colored horizontal dashed lines. The gray dashed line indicates the measured fluorescence lifetime $\tau_0$ in the absence of the ZMW. The predicted signal-averaged lifetimes $\langle \tau \rangle_S$ are 1.98 ns, 1.88 ns, and 1.88 ns for BSA–Alexa488, and 1.22 ns, 1.95 ns, and 1.15 ns for Kap95–Alexa647, for pore diameters of 50 nm, 100 nm, and 150 nm, respectively (see *Equation 13*). The quantum yields and fluorescence lifetimes were estimated based on a literature values of $\Phi_{lit}$ = 0.8 and $\tau_{lit}$ = 4.0 ns for Alexa488 (*Sanabria et al., 2020*), and $\Phi_{lit}$ = 0.33 and $\tau_{lit}$ = 1.17 ns for Alexa647 (*Hellenkamp et al., 2018*), and measured lifetimes in the absence of the ZMW of $\tau_0$ =2.3 ns for BSA–Alexa488 and $\tau_0$ = 1.37 ns for Kap95–Alexa647 (compare *Appendix 7—figure 1*).

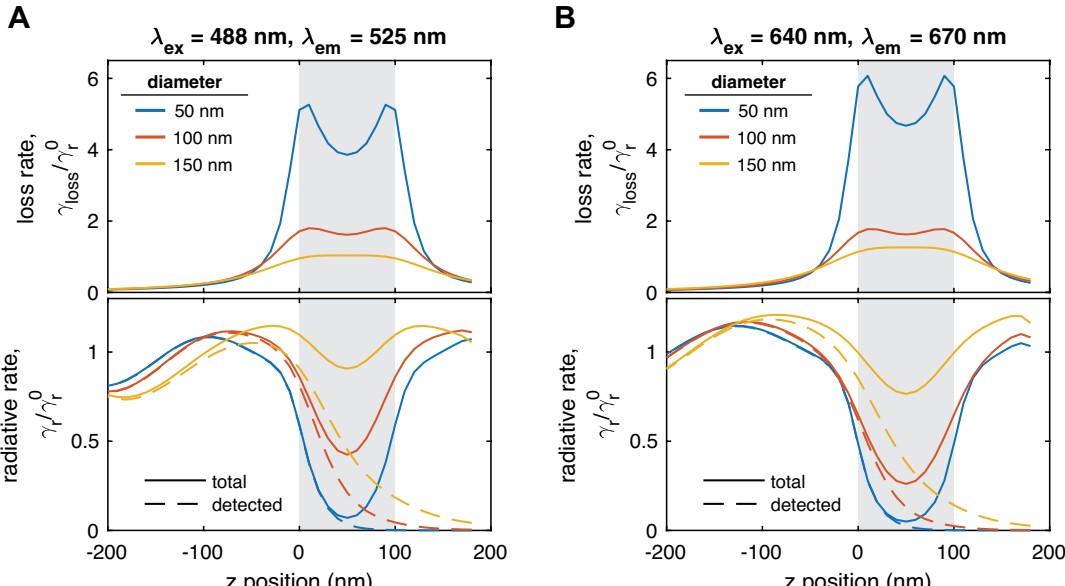

**Appendix 2—figure 5.** Z-profiles of the radiative (emission) and non-radiative (loss) rates in the proximity of the freestanding zero-mode waveguide (ZMW) obtained from finite-difference time-domain (FDTD) simulations for the blue (**A**, $\lambda_{ex}$ = 488 nm, $\lambda_{em}$ = 525 nm) and red (**B**, $\lambda_{ex}$ = 640 nm, $\lambda_{em}$ = 670 nm) channels. The position of the palladium membrane is indicated as a gray shaded area. The z-axis is defined as in *Appendix 2—figure 4A*. For the radiative rate (bottom), the rate of emission directed toward the objective lens is displayed in addition as a dashed line. The normalized loss rate $\gamma_{loss}/\gamma_r^0$ and radiative emission rate $\gamma_r/\gamma_r^0$ are obtained by measuring the total power emitted by the dipole and comparing it to the power that is emitted into the far field, that is, not absorbed by the metal. From these rates, the quantum yield $\Phi$ and fluorescence lifetime $\tau$ are computed according to *Equations 8 and 10*. The ratio of the total emission rate and the rate of the detected emission (solid and dashes lines) is used to compute the detection efficiency $\eta$ as given in *Equation 11*. See 'Materials and methods' for details.

# Appendix 3

## Free Alexa fluorophore translocation

At the end of a full experiment with BSA and KAP95, mixtures of 50 nM or 100 nM Alexa 647 and Alexa 488 were flushed into the reservoir to detect clogged pores from the absence of events for the small fluorophores. As expected, small fluorophores were barely hindered in their diffusion from the Nsp1 molecules in the pore. Notably, this contrasts the findings of *Ananth et al., 2018*, who reported only a residual conductance of (charged) ions through Nsp1-coated pores smaller than 40 nm, whereas we still found unhindered diffusion of free fluorescent dyes.

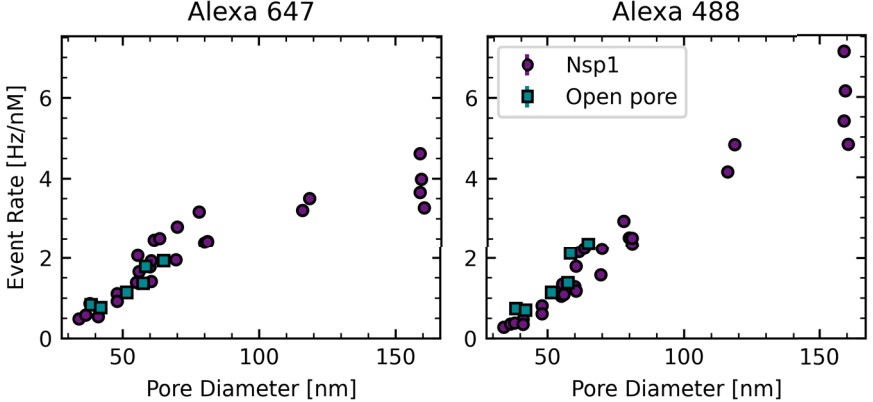

**Appendix 3—figure 1.** Free fluorophore translocations. (Left) Event rate versus pore diameter for Alexa 647. (Right) Event rate versus pore diameter for Alexa 488. The event rate of both fluorophores does barely change between Nsp1-coated pores and open pores.

## Influence of neighboring pores

In order to estimate an upper limit on the event rate that is found at a pore location due to analytes that diffused into the detection region from adjacent pores, we looked for an experiment where a closed pore was neighbored by two unblocked pores. In an open pore experiment, we encountered such a setting, where the closed pore was identified by its very low event rate. Specifically, in this case, pore 6 had a diameter of 56 nm, pore 7 was closed, and pore 8 had a diameter of 70 nm. The respective BSA translocation rates at 250 nM were 36 Hz, 0.8 Hz, and 93 Hz. Assuming that all events found on pore 7 were due to diffusion from their neighboring pores, this gives that on any pore less than $2\% \approx \frac{0.8\,\mathrm{Hz}}{36\,\mathrm{Hz}}$ of the events are not due to translocations through the pore itself but are due to diffusion from neighboring pores.

## Effect of pressure-induced hydrodynamic flow

In the experiments, we applied a constant flow to the microfluidic channel below the nanopore to continuously remove translocated analytes from solution by applying a pressure of 50 mbar. As a side effect, this will also induce a back flow through the pore from the detection side to the reservoir side, with a theoretical pressure difference over the membrane of 25 mbar. Because all experiments were performed under identical pressure, this effect should result in a constant reduction of the event rate and not affect the conclusions. To test the magnitude of the pressure-induced reduction of translocation rates, we performed control experiments on an open pore with a diameter of 76 nm using free fluorophores (Alexa488, Alexa647) at concentrations of 100 nM. At these concentrations, it was not possible to detect single events owing to the high event rates, hence we quantified the total count rate detected at varying pressures. The normalized signal count rates as function of the applied pressure are shown in *Appendix 3—figure 2A*, showing the expected linear dependence as predicted by the Hagen–Poiseuille equation. At the applied pressure of 50 mbar, we observe an approximately 5% reduction of the detected count rate, suggesting only a minor effect of the pressure gradient on the observed translocation rates. This is confirmed by direct measurements of the Kap95 event rates for open pores as a function of the applied pressure (*Appendix 3—figure 2B*), which show only a minor effect of the pressure in the range below 100 mbar. However, at 200 mbar,

event rates are approximately reduced by half, which exceeds the count rate reduction observed for the dye solution in *Appendix 3—figure 2A*, most likely due to increased viscous drag acting on the larger protein compared to the small organic dyes. Note that the absolute event rates obtained for these experiments are not directly comparable to the results presented in the main text as the experiments have been performed on gold pores using a different cleaning protocol using acidic piranha etching.

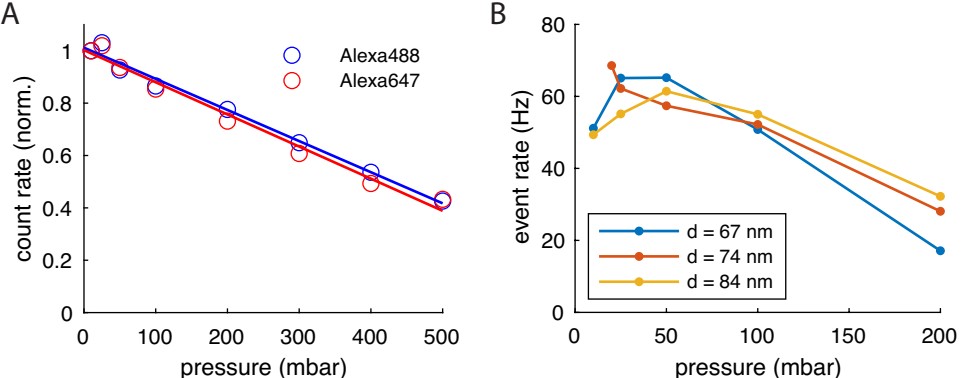

**Appendix 3—figure 2.** Pressure-dependent reduction of translocation rates. (**A**) A solution of the free fluorophores Alexa488 and Alexa647 at a concentration of 100 nM was placed in the reservoir. The signal count rate was monitored at the exit of a pore with a diameter of 74 nm as a function of the applied pressure to the flow channel. Count rates were normalized to the values obtained in the absence of a pressure difference. Due to the pressure-induced hydrodynamic flow against the concentration gradient, the translocation rates decrease linearly with the applied pressure. (**B**) Event rates for Kap95 at a concentration of 1 μM acquired for three different open pores with diameters in the range of 67–84 nm. The event rates decrease markedly at a high pressure of 200 mbar, but remain approximately constant in the range below 100 mbar.

## Signal-to-background ratio

The single-molecule signal obtained in this study offers a significantly higher signal-to-background ratio compared to previous approaches. To illustrate this point, we compare the signal-to-background ratio in our experiments to the average signal-to-noise ratio for a comparable conductance-based system (*Fragasso et al., 2022*).

By quantifying the event-wise signal-to-background ratios for the measurement shown in *Figure 3*, we estimate an average signal-to-background ratio of 56 ± 41 for BSA and 67 ± 53 for Kap95 (*Figure 3*). Using a representative segment of a 1-min-long current trace of a Nsp1-coated pore with a diameter of 55 nm acquired under experimental similar conditions (TE buffer, pH = 7.4, 150 mM KCl, 21, 100 mV bias voltage), we estimate a current noise of 0.014 nA with a mean current of 3.139 nA (compare Figure 1d in *Fragasso et al., 2022*). The average current blockade of single Kap95 translocations was reported to be 0.08 nA at the applied voltage, which results in a signal-to-noise ratio of approximately 5.6 for conductance-based experiments. Compared to a value of 67 reported here, the ZMW approach thus offers a more than 10 times better separation of spikes originating from single translocations events.

In *Fragasso et al., 2022*, individual Kap95 translocations could only be resolved at low concentrations of 119 nM, above which single events were not visible due to an insufficient signal-to-noise ratio, as stated by the authors. Such a limitation does not exist for the ZMW approach, where we could detect single-protein translocations also at high occupancy of Kap95 in Nsp1-coated pores. Moreover, as pointed out in the 'Discussion' section, the discrepancy between theoretically predicted translocation rates and experimentally measured translocation rates is orders of magnitude better for the ZMW approach compared to the conductance-based readout. We hence conclude that the capability to resolve single translocations is markedly improved for the ZMW-based fluorescence readout compared to the conductance-based approach.

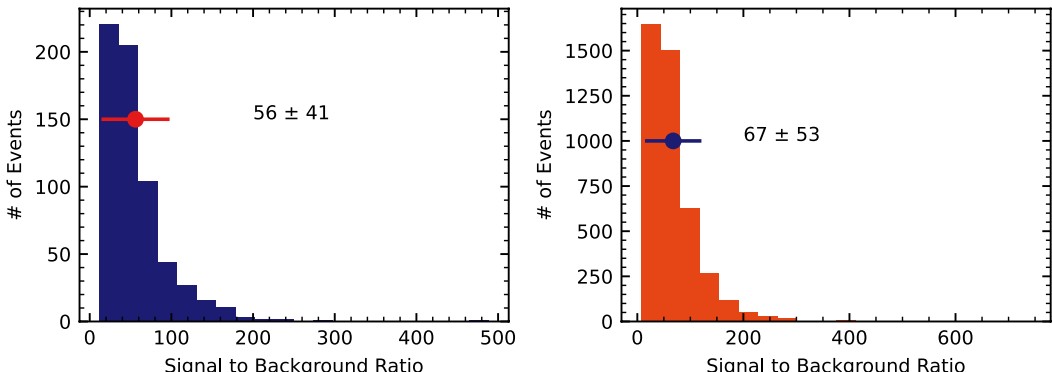

**Appendix 3—figure 3.** Signal-to-background ratios of single-molecule events. Distributions of the signal-to-background ratio for BSA (left) and Kap95, defined as the ratio of the event signal to the average background signal. The background level was estimated as the average photon rate in the absence of fluorescently labeled analytes. The dot represents the mean and the bar the standard deviation of the distribution.

## Appendix 4

### Measurement scheme

The measurement scheme was the following:

- No Kap95

  Measure PBS as a background reference
  Pre-incubate with 500 nM of BSA, wash it out with 250 nM of BSA
  Measure 250 nM of BSA
  Measure 500 nM of BSA

- Wash
  ○ Wash with PBS
  ○ Wash with 5 % hexane-1-6-diol
  ○ Wash with PBS
- 100 nM Kap95
  ○ Measure 100 nM of Kap95
  ○ Measure 100 nM of Kap95 and 250 nM of BSA
  ○ Measure 100 nM of Kap95 and 250 nM of BSA
- Wash
  ○ Wash with PBS
  ○ Wash with 5% hexane-1-6-diol
  ○ Wash with PBS
- For some experiments, an additional 200nM of Kap95 measurement was performed at this point.
- 1000 nM Kap95
  ○ Measure 1000 nM of Kap95
  ○ Measure 1000 nM of Kap95 and 250 nM of BSA
  ○ Measure 1000 nM of Kap95 and 500 nM of BSA
- Wash with PBS
- Measure 50 nM of Alexa 488 and 50 nM of Alexa 647 in PBS.

Changing the analyte in the reservoir was done by replacing the volume at least three times. After changing the contents of the reservoir, a new transmission light scan of all pores was performed to obtain the accurate locations. Subsequently, each pore was measured one after another.

Hexane-1-6-diol (Sigma-Aldrich) was diluted to 5% in 150 nM KCl with 1×TE buffer.

## Appendix 5

### Detailed diffusion model

We explored how well the translocation rates can be accurately described by a diffusion model. In theory, the event rates should follow a diffusion model based on Fick's law. For reference, such a model was fit to the individual data points underlying the averages shown in *Figure 4A–D* based on Fick's law of diffusion:

$$k_{\text{Prot}} = \alpha \pi r^2 D_p(r) \frac{\Delta c}{L}, \tag{14}$$

where $k_{\text{Prot}}$ is the translocation rate, $r$ the pore radius, $\Delta c$ the concentration difference between *cis* and *trans*, and $L$ the length of the pore. The scaling factor $\alpha$ accounts for deviation from the ideal behavior due to protein–pore interactions and events missed by the detection algorithm. $D_p(r)$ is the reduced diffusion coefficient due to confinement in the pore and is calculated as given by *Dechadilok and Deen, 2006*:

$$\frac{D_p(r)}{D} = \quad 1 + \frac{9}{8}\left(\frac{R_g}{r}\right)\ln\left(\frac{R_g}{r}\right) - 1.56034\left(\frac{R_g}{r}\right) + 0.528155\left(\frac{R_g}{r}\right)^2 + 1.91521\left(\frac{R_g}{r}\right)^3$$

$$-2.81903\left(\frac{R_g}{r}\right)^4 + 0.270788\left(\frac{R_g}{r}\right)^5 + 1.10115\left(\frac{R_g}{r}\right)^6 - 0.435933\left(\frac{R_g}{r}\right)^7, \tag{15}$$

where $R_g$ is the radius of gyration of the protein ($R_g$ = 3.15 nm for Kap95 and $R_g$ = 2.68 nm for BSA as determined from their crystal structures) and $D$ is the diffusion coefficient that was obtained from FCS measurements as $D$ = 46 µm²/s for Kap95 and $D$ = 54 µm²/s for BSA at 21 °C. As an alternative analysis, this model was fitted using the least-squares method to the individual data points underlying the averages from *Figure 4* in the main text, as shown in *Appendix 6—figure 1*. The open pore data as well as the Nsp1 pore data for Kap95 can be well fitted by the diffusion model. The BSA rates through Nsp1 pores, however, cannot be described with a single diffusion model. Inspired by the opening of the Nsp1 mesh seen in the CGMD simulations, we introduced a threshold, where the translocation rates would follow a different diffusion regime. We note, however, that in this transition region between 48 nm and 55 nm the data is better described by the shifted quadratic function.

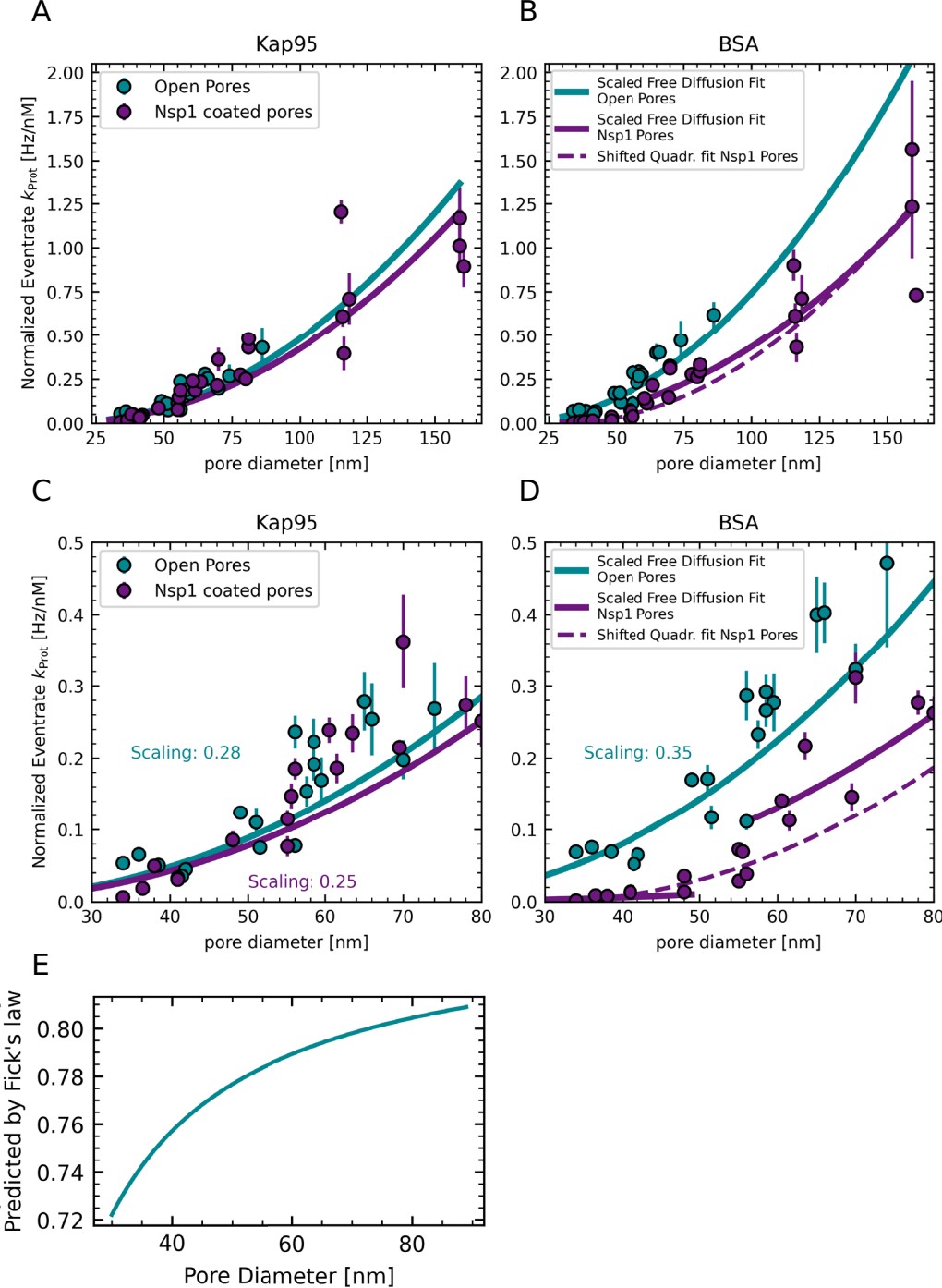

**Appendix 5—figure 1.** Free diffusion fits to experimental data. (**A, B**) Normalized and averaged event rate vs pore diameter as in *Figure 4*. Solid lines are fits of a diffusion model as described in this section. A single fit for BSA through Nsp1 would deviate both for small- and large-pore diameters. Therefore, the dataset was split into two regimes: below 50 nm and above 60 nm in order to fit the data. This threshold was inspired by the results of the coarse-grained molecular dynamics (CGMD) simulations. The dashed line shows a fit of the shifted quadratic function to the BSA data through Nsp1 pores as described in the main text. (**C, D**) Zoom-ins of (**A, B**). (**E**) Selectivity of open pores, predicted by the diffusion model. The value is pore size dependent because of the different sizes of Kap95 and BSA.

## Appendix 6

### Fitting of translocation rates

For congruence with the computational data, we fit a simplified diffusion model to the individual data points underlying the averages presented in *Figure 4* using a quadratic function:

$$k_{\text{Prot}} = \alpha \left( r - r_{\text{Prot}} \right)^2, \tag{16}$$

where $k_{\text{Prot}}$ is the concentration-normalized translocation rate, $r$ is the pore radius, $r_{\text{Prot}}$ the protein's equivalent radius as proposed by *Winogradoff et al., 2022* (see Appendix 10 for details), and $\alpha$ a multiplicative scaling factor incorporating all factors such as concentration gradient and pore length. All fits were performed using a least-squares fitting, taking a statistical error on the individual event rates into account. This error was estimated from Poisson statistics, where the number $N$ of events within a timetrace that leads to $\Delta N = \sqrt{N}$ was fitted individually on the event rates of Kap95 through open and Nsp1-coated pores. Additionally, the BSA event rates through open pores were fitted separately. The parameter $\alpha$ resulting from the BSA translocations for open pores was used as a fixed parameter when fitting the BSA event rates through Nsp1-coated pores with a shifted and reduced quadratic function:

$$k_{\text{Prot}} = \alpha \left( r - r_{\text{Prot}} - b \right)^2, \tag{17}$$

where $b$ is a shift parameter introduced to take into account a further reduction of the BSA event rate at small-pore diameters.

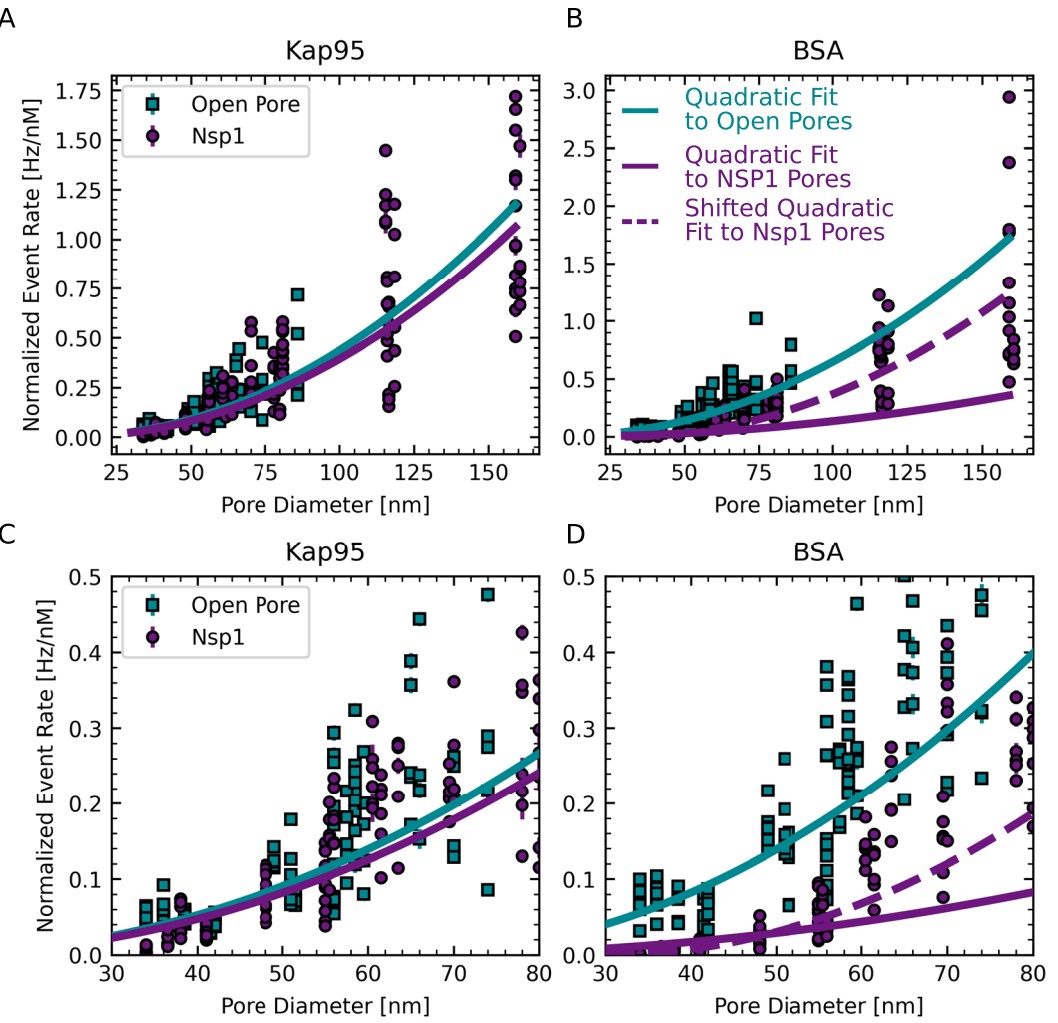

**Appendix 6—figure 1.** Fits to individual datapoints. (**A, B**) Normalized event rate vs. pore diameter of the individual conditions underlying the points shown in *Figure 4*. Solid lines are fits of a quadratic function as given in *Equation 16*. The BSA event rates through Nsp1-coated pores were not well described by the quadratic function. Therefore, a shift parameter was introduced as given in *Equation 17* (dashed line). (**C, D**) Zoom-ins of (**A, B**).

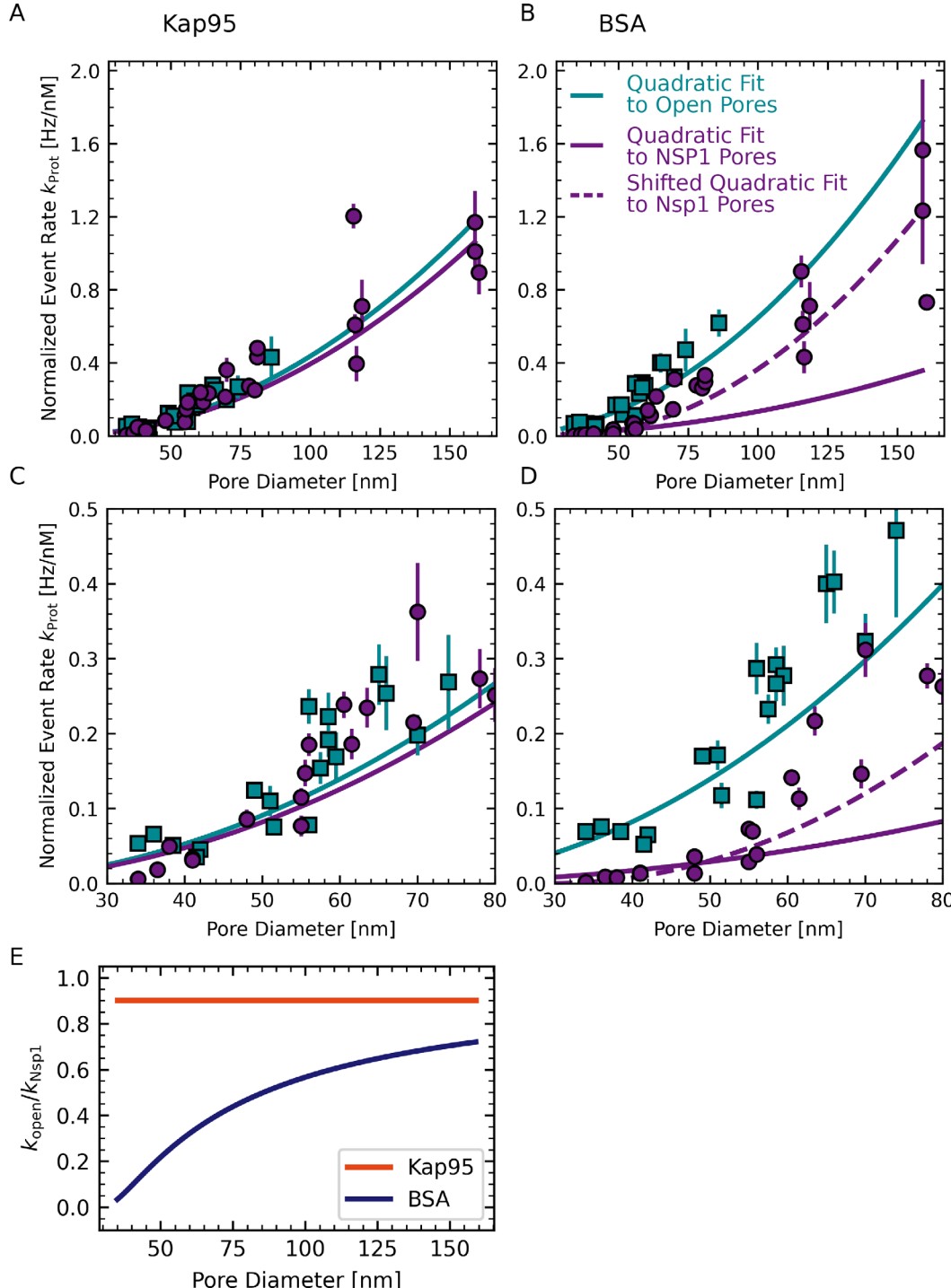

**Appendix 6—figure 2.** Simple quadratic fit to experimental data. (**A, B**) Normalized and averaged event rate vs. pore diameter as in *Figure 4*. Solid lines are fits of a quadratic function as given in *Equation 16*. The fit for BSA through Nsp1 deviates both for small- and large-pore diameters. Therefore, a shift parameter was introduced as given in *Equation 17* to fit the data (dashed line).(**C, D**) Zoom-ins of (**A, B**). (**E**) Open pore event rates obtained from the quadratic fit divided by Nsp1 pore event rates obtained from the quadratic fit vs. pore diameter. While there is barely any decrease of the event rate for Kap95 when Nsp1 is present, BSA experiences an approximately tenfold decrease for small pores of 35 nm diameter. This ratio increases with pore diameter and approaches a value of 1 in the limit to very large pores.

## Appendix 7

### Diffusion coefficients and fluorescence lifetimes

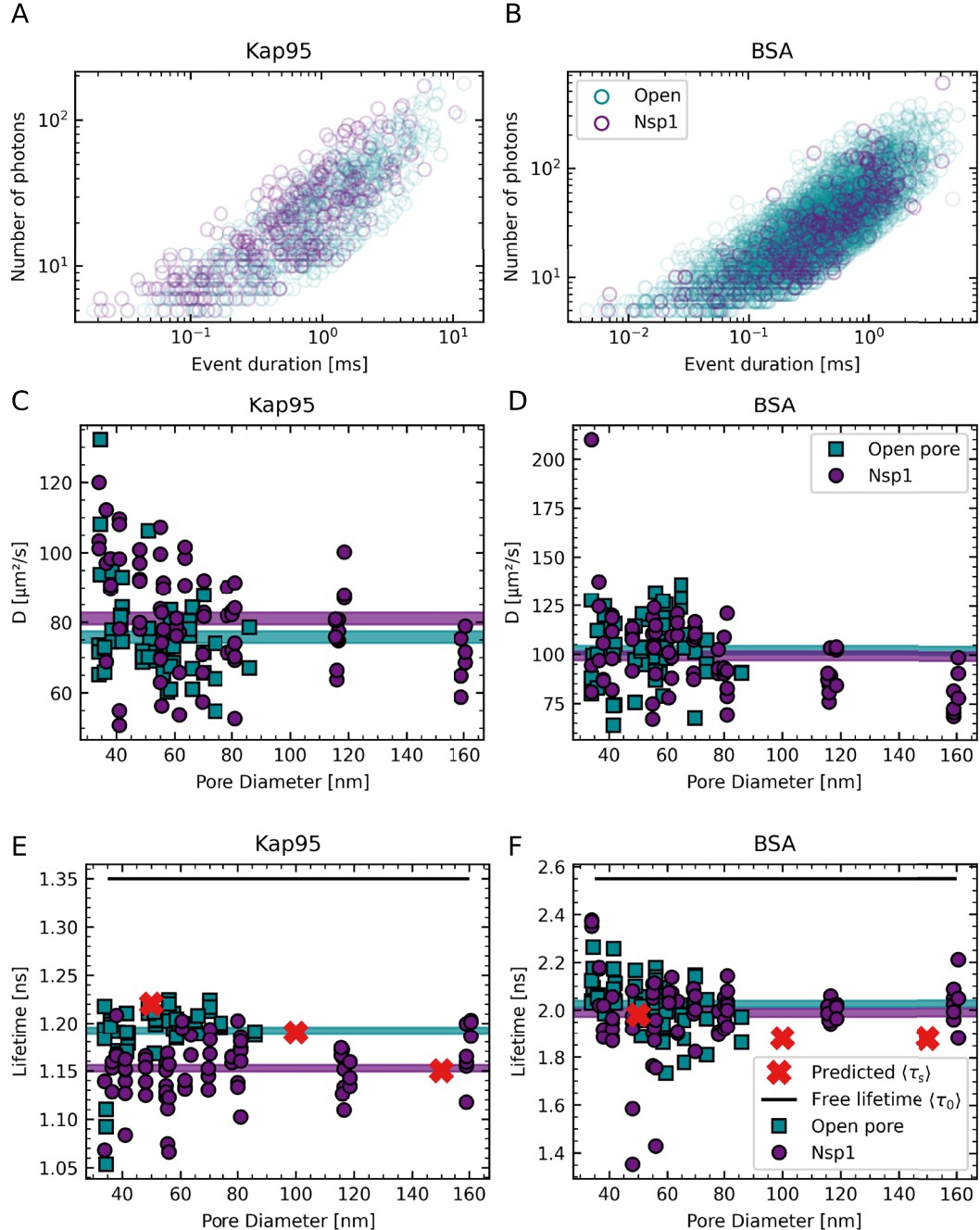

**Appendix 7—figure 1.** Physical characteristics of timetraces. (**A, B**) Scatter plots of the events detected for Kap95 (**A**) and BSA (**B**) indicating the distributions of event duration and the amount of photons within an event for the data shown in *Figure 3C and D*. Both distributions overlap, showing that the spike detection works equally for coated and open pores. (**C, D**) Plots of the diffusion coefficient vs. pore diameter for Kap95 and BSA, respectively. The diffusion coefficient is estimated by fluorescence correlation spectroscopy (FCS) analysis of the time traces obtained in the nanopore experiments at the highest protein concentration. The horizontal lines indicates the average diffusion coefficient and the width corresponds to twice the standard error of the mean. The average diffusion coefficient of both Kap95 and BSA shows no significant difference between open pores and Nsp1-

*Appendix 7—figure 1 continued on next page*

*Appendix 7—figure 1 continued*

coated pores. This indicates that interactions of the proteins with the Nsp1 mesh do not obstruct the diffusion. Alternatively, it is possible that the bound fraction is not detected in our experiments if it is close to the metal surface due to metal-induced quenching of the fluorescence signal. (**E, F**) Plots of the fluorescence lifetime vs. pore diameter for Kap95 and BSA, respectively. The fluorescence lifetime is calculated based on the individual time traces of the highest protein concentration. The mean (horizontal lines) with twice the standard error of the mean (width of the lines) gives an estimate of the spread. For both Kap95 and BSA, the average fluorescence lifetime is significantly lower than what is measured in open solution (black lines). This can be attributed to the influence of the nearby metal nanostructure on the radiative and non-radiative rates. The predicted lifetimes based on finite-difference time-domain (FDTD) simulations are shown as red crosses (compare *Appendix 2—figure 4*). Whereas for BSA the lifetime in Nsp1-coated pores and open pores does not differ significantly, there is a significant decrease of the fluorescence lifetime of Kap95 in Nsp1-coated pores compared to open pores. This suggests that Kap95 remains within the proximity of the pore for a longer time when Nsp1 is present.

Our measurements provide additional information on the diffusivity of the proteins and the fluorescence lifetime of the fluorophores (*Appendix 7—figure 1*). From this, we can gain additional insight on the interaction of the proteins with the Nsp1 mesh within the pore. Interestingly, we do not observe a significant hindrance of the diffusivity for either BSA or Kap95, despite the known interaction of Kap95 with the FG repeats of the Nsp1 mesh. This indicates that the dynamic and multivalent interactions do not markedly slow down the diffusion of Kap95. Alternatively, it is possible that the fluorescence signal of the interacting species is quenched due to the close proximity to the metal surface due to metal-induced electron transfer (*Gregor et al., 2019*). This would render it impossible to detect the interactions because the detected signal would originate predominantly from freely diffusing molecules that have left the Nsp1 mesh.

Additional information on the photophysics is obtained from the fluorescence lifetime. We find a significant decrease of the fluorescence lifetime in the presence of the metal nanostructure (compared to open focus measurements) both for the dyes Alexa488 on BSA and Alexa647 on Kap95. Note that this lifetime reduction can originate both from metal-induced quenching (*Gregor et al., 2019*) or a radiative rate enhancement within the ZMW (*Levene et al., 2003*). FDTD simulations of the dipole emission confirm that the fluorescence lifetime is shortened in the proximity of the metal nanostructure (*Appendix 2—figure 4D and E*) and provide good qualitative agreement with the measured fluorescence lifetimes of translocating molecules (see red crosses in *Appendix 7—figure 1E and F*).

Whereas for BSA there was no difference between the fluorescence lifetimes obtained for Nsp1-coated pores and open pores, we found a small but significant reduction of the fluorescence lifetime for Kap95 for Nsp1-coated pores. This indicates that, in the presence of Nsp1, Kap95 molecules diffuse closer to or spend more time in proximity of the metal nanoaperture on the exit side. Intriguingly, it has recently been reported that Kap95 predominantly translocates along the periphery of the NPC (*Chowdhury et al., 2022*), which falls in line with our observation of a stronger coupling to the metal nanoaperture.

## Appendix 8

A

B

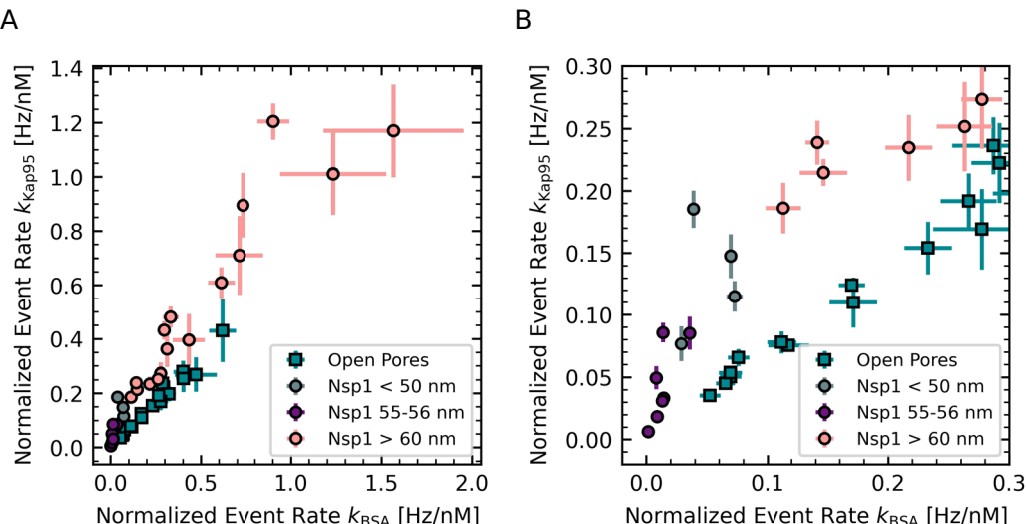

**Appendix 8—figure 1.** Correlation between event rates for Kap95 and BSA. (**A**) Normalized event rates of Kap95 vs. BSA for small (purple) and large (pink) Nsp1-coated pores and open pores (cyan). The event rates show a high degree of correlation with correlation coefficients of 0.72, 0.93, and 0.98, respectively. Additionally, pores with diameters of 55–56 nm are shown in gray. (**B**) Zoom-in of (**A**).

# Appendix 9

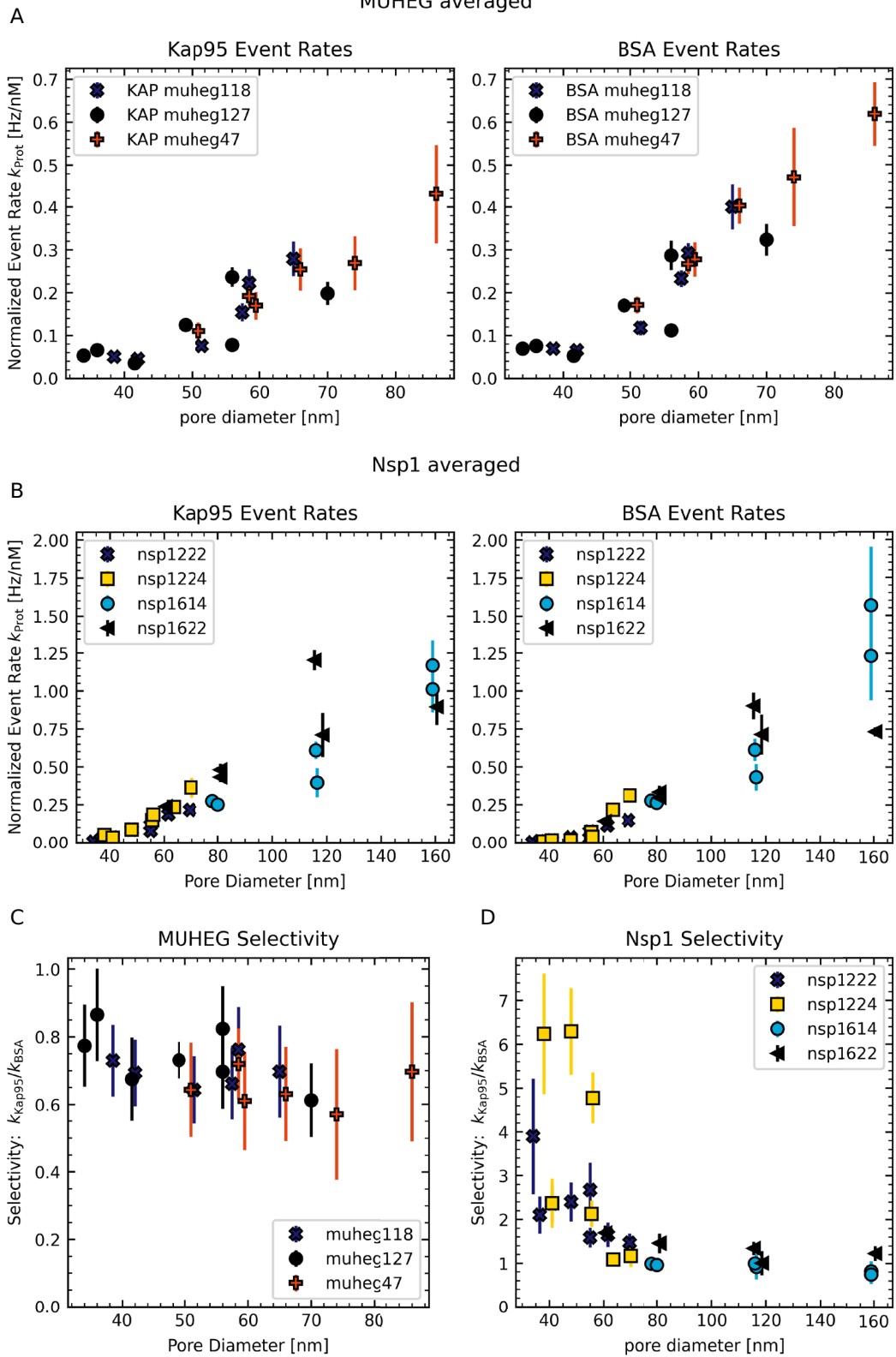

*Appendix 9—figure 1 continued on next page*

*Appendix 9—figure 1 continued*
**Appendix 9—figure 1.** Event rates and selectivity ratios are consistent between separate experiments. (**A, B**) Normalized event rates for different open pore and Nsp1 pore experiments, respectively. The different datasets are highlighted according to their experimental day. A label for each experimental realization is given in the legend. We do not observe any striking day-to-day variation. (**C, D**) Selectivity for different open pore and Nsp1 pore experiments. Also in the selectivity we do not observe a striking difference between experimental repetitions.

## Appendix 10

### Probe radius of BSA and Kap95

To determine the radius of a protein (used in the void analysis and for fitting the event rates), we used the procedure as described in *Winogradoff et al., 2022*. We started by computing the protein's moments of inertia, $I_X$, $I_Y$, and $I_Z$, from the all-atom crystal structure. We then matched the moments of inertia of the protein with those of a constant density ellipsoid using $I_X = \frac{1}{5}m\left(b^2 + c^2\right)$, $I_Y = \frac{1}{5}m\left(a^2 + c^2\right)$, and $I_Z = \frac{1}{5}m\left(a^2 + b^2\right)$, where $m$ is the total mass of the protein and $a$, $b$, and $c$ are the respective lengths of the three principal axes of the ellipsoid. To obtain the protein radius, we equated the volume of a sphere to the volume of the ellipsoid, that is, $r_p = (abc)^{1/3}$. Using this method, we find a probe radius of $r_p = 34$ Å for BSA (PDB ID: 4F5S) and $r_p = 40$ Å for Kap95 (PDB ID: 3ND2).

### Fitting calculated event rates for open pores

To obtain the scaling constant, $k_0$, of the Arrhenius relation, we fitted the calculated event rates for open pores, obtained from the PMF barriers using the Arrhenius relation, to the (concentration-normalized) experimental open pore event rates (see Appendix 6 for fitting procedure). We note that the calculated event rates for open pores follow *Equation 2* exactly. We therefore scaled the calculated event rates such that they align with the fits in *Figure 4A and B* (cyan lines) and used the same constant $k_{0,\mathrm{BSA}}$ to the Arrhenius relation for Nsp1-coated pores.

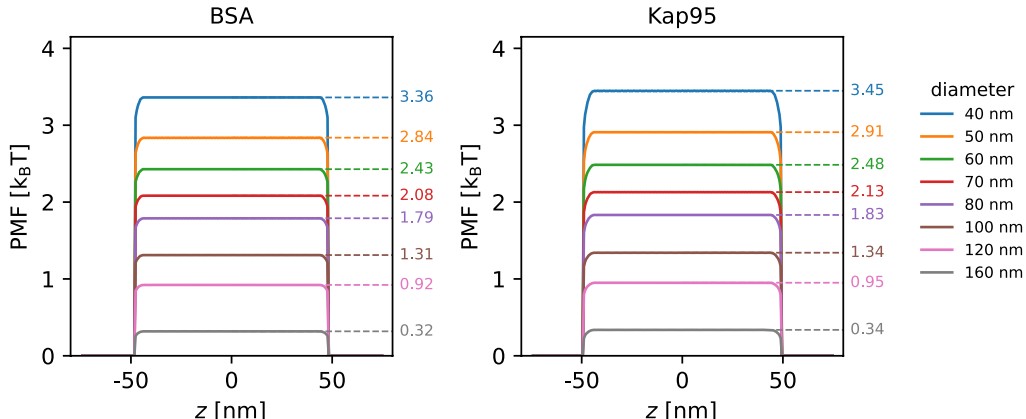

**Appendix 10—figure 1.** Potentials of mean force for open pores. Potentials of mean force along the pore axis of open pores with various diameters derived from void analysis using a probe radius of $r_{\mathrm{probe}}$ of 34 Å for BSA and 40 Å for Kap95. The numbers next to the dashed lines indicate the permeability barriers $\Delta E$.

### PMF for Nsp1-coated pores

The Nsp1 proteins in our simulations were anchored to the surface in a triangular fashion to achieve a homogeneous grafting density across the entire scaffold. We note that the location of the peaks in the PMF curves aligns with the $z$-coordinates of the Nsp1 anchor sites. Although the anchor sites clearly contribute to the translocation barrier, it would not be correct to use the maximum PMF as the energy barrier for translocation as the idealized anchoring of Nsp1 on the scaffold is not a good approximation of the experiments (where Nsp1 is not anchored at discrete $z$-positions). Instead, to obtain a correct estimate of the energy barrier, $\Delta E$, we used the average PMF for $-25$ nm$\leq z \leq 25$ nm.

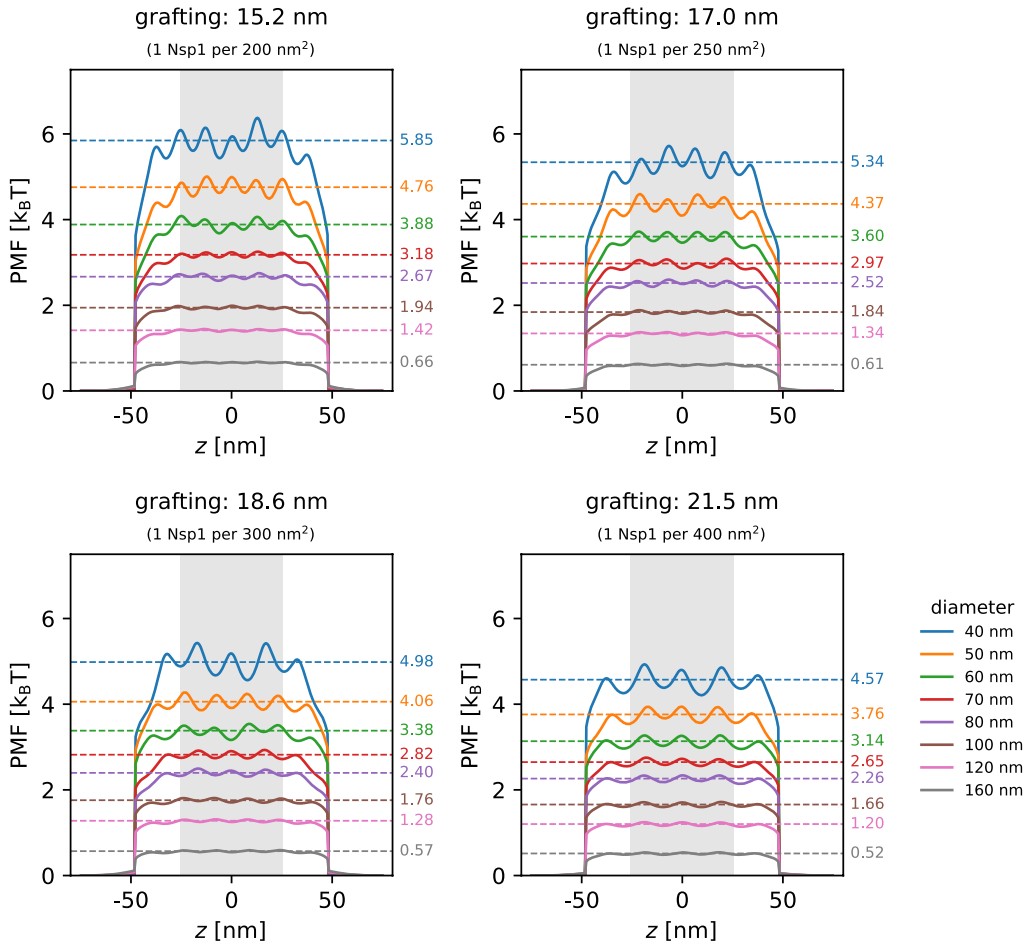

**Appendix 10—figure 2.** Potentials of mean force (PMFs) along the pore axis of Nsp1-coated pores with various diameters and grafting densities using a probe radius of $r_{probe}$ = 34 Å (BSA) derived from void analysis. The height of the PMF barrier, $\Delta E$ given on the right of the plots, is obtained by averaging the PMF between –25 nm ≤ $z$ ≤ 25 nm (gray region). The location of the peaks in the PMF curves is at the same $z$-coordinates as the anchoring points of the Nsp1 proteins.

## Dependence of selectivity on grafting density

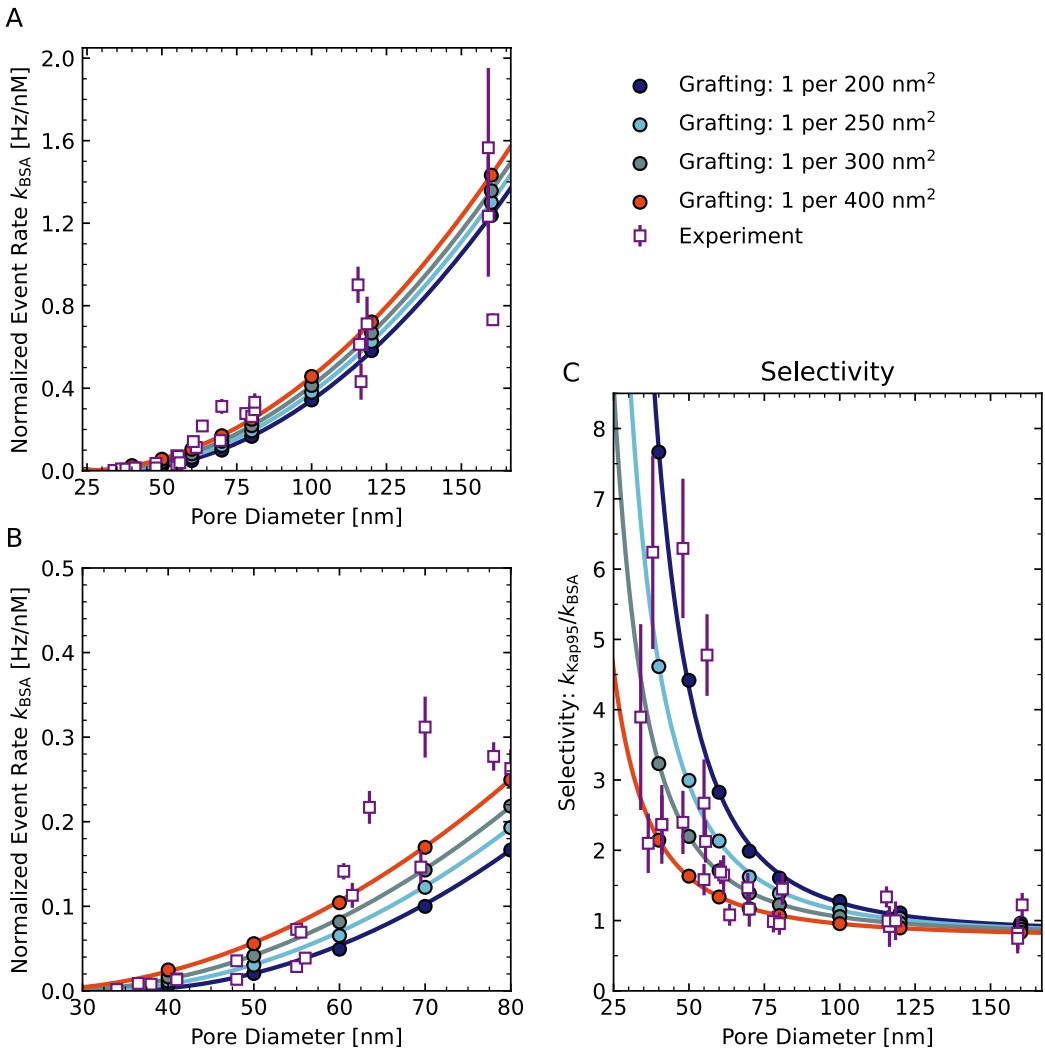

**Appendix 10—figure 3.** Effect of grafting density on the selectivity of Nsp1 pores. (**A**) Calculated BSA event rates for Nsp1 pores for various grafting densities. (**B**) Zoom-in of (**A**). (**C**) Apparent selectivity versus pore diameter, where the selectivity is calculated as the ratio of Kap95 to BSA event rates. For Kap95, we assumed that the event rate is the same for open pores and Nsp1 pores.

## Nsp1 density distributions

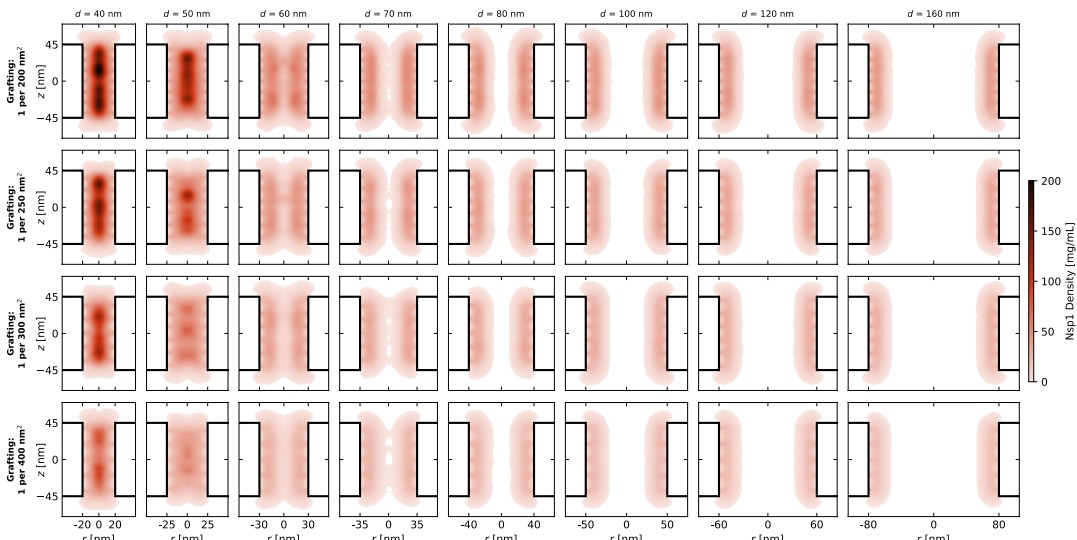

**Appendix 10—figure 4.** Time-averaged $r$–$z$ density distribution of Nsp1-coated nanopores for various diameters and grafting densities. Although there is a significant variation in the central channel densities for small diameter pores, the range of diameters at which the structural transition of the Nsp1 mesh takes place is largely independent of the Nsp1 grafting density in the range that was tested.

## Appendix 11

### Kap95 rate change

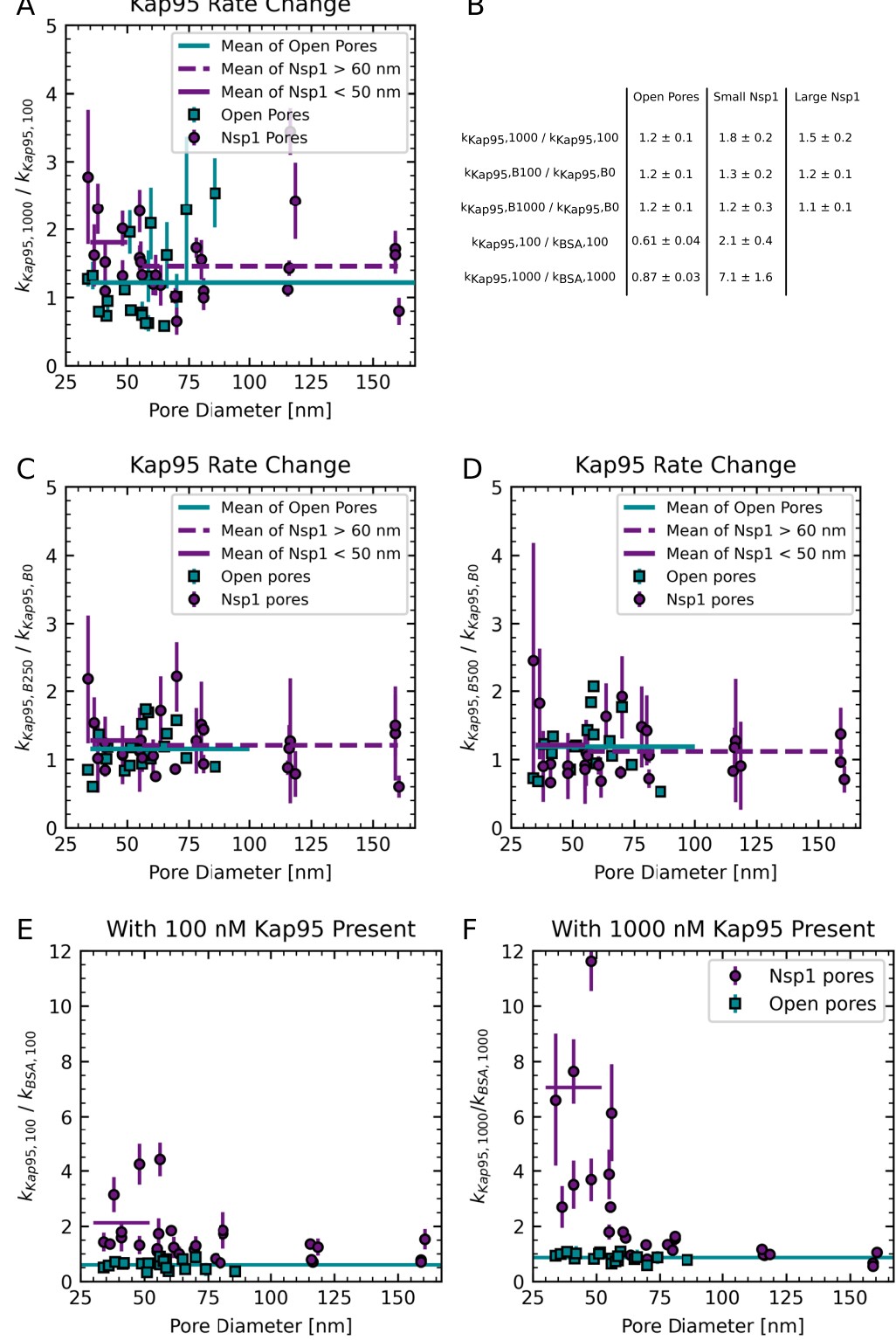

*Appendix 11—figure 1 continued on next page*

*Appendix 11—figure 1 continued*

**Appendix 11—figure 1.** Rate changes due to presence of Kap95. (**A**) Rate change of Kap95 vs. pore diameter. Event rate of Kap95 at 1000 nM ($k_{\text{Kap95,1000}}$) divided by the Kap95 event rate at 100 nM ($k_{\text{Kap95,1000}}$). At high Kap95 concentrations, there is an increase in normalized event rate for both open and Nsp1 pores. It is largest for small Nsp1 pores. (**B**) The averages indicated in the plots by horizontal lines given with standard deviation estimated from fitting horizontal lines. (**C, D**) Influence of BSA concentration on Kap95 event rate. When dividing the Kap95 event rate with 250 nM of BSA being present $k_{\text{Kap95,B250}}$ by the Kap95 event rate with 0 nM of BSA being present $k_{\text{Kap95,B0}}$ (**C**), this shows that the Kap95 event rate is barely influenced by the presence of BSA, as expected. The same holds for the Kap95 event rate change with 500 nM of BSA to 0 nM being present. (**E, F**) Selectivity vs. pore diameter for different Kap95 concentrations. When switching from 100 nM of Kap95 to 1000 nM of Kap95, the selectivity of small Nsp1 pores increases by a factor of 3, whereas the selectivity of large-pore decreases.

## Model for the residual selectivity of large pores

The model as presented in **Figure 6** is based on the assumption a pore is occupied by two separate cross-sectional areas, a selective area and an unselective area, whereas no assumption over their location (such as a ring) is made. The full cross-sectional area $A$ is thus divided as

$$A = A_{\text{open}} + A_{\text{Nsp1}}, \tag{18}$$

where $A_{\text{open}}$ is the open (unselective pore area) and $A_{\text{Nsp1}}$ is the selective (Nsp1-filled) area. Here, the selectivity $S$ of a pore is defined as the ratio of Kap95 and BSA rates

$$S = \frac{k_{\text{Kap95}}}{k_{\text{BSA}}}, \tag{19}$$

which can be expressed in terms of the contributions of the two phases as

$$S = \frac{A_{\text{Nsp1}} l_{\text{Kap95}}^{\text{Nsp1}} + A_{\text{open}} l_{\text{Kap95}}^{\text{open}}}{A_{\text{Nsp1}} l_{\text{BSA}}^{\text{Nsp1}} + A_{\text{open}} l_{\text{BSA}}^{\text{open}}}, \tag{20}$$

where $l_{\text{protein}}^{\text{phase}}$ is the area normalized event rate constant for the respective protein in the different phases, that is, open or Nsp1-filled.

Since we found experimentally the Kap95 event rates were unchanged by the presence of Nsp1, we approximate $l_{\text{Kap95}} \approx l_{\text{Kap95}}^{\text{open}} \approx l_{\text{Kap95}}^{\text{Nsp1}}$. The selectivity of a mixed phase pore is then given by

$$S = \frac{l_{\text{Kap95}}}{l_{\text{BSA}}^{\text{Nsp1}} \frac{A_{\text{Nsp1}}}{A} + l_{\text{BSA}}^{\text{open}} \frac{A_{\text{open}}}{A}} = \frac{1}{\frac{l_{\text{BSA}}^{\text{Nsp1}}}{l_{\text{Kap95}}} \frac{A_{\text{Nsp1}}}{A} + \frac{l_{\text{BSA}}^{\text{open}}}{l_{\text{Kap95}}} \frac{A_{\text{open}}}{A}} = \frac{1}{\frac{1}{S_{\text{Nsp1}}} \frac{A_{\text{Nsp1}}}{A} + \frac{1}{S_{\text{open}}} \frac{A_{\text{open}}}{A}} \tag{21}$$

Note that this corresponds to the harmonic mean of the selectivities.

When considering the structure of the pore, the number of Nsp1 molecules in the pore is determined by the grafting density $\sigma$, the pore's radius $r$, and length $L$ as

$$N = 2\pi r \sigma L. \tag{22}$$

If we assume that each Nsp1 molecule renders a certain volume $V$ selective, the effectively selective cross-sectional area can be calculated from the selective volume $V_{\text{Nsp1}}$ as

$$A_{\text{Nsp1}} = \frac{V_{\text{Nsp1}}}{L} = \frac{NV}{L} = 2\pi r \sigma V. \tag{23}$$

Thus, the selective area fraction depends on the pore radius as

$$\frac{A_{\text{Nsp1}}}{A} = \frac{2\pi r}{\pi r^2} \sigma V = \frac{2}{r} \sigma V, \tag{24}$$

Combining both expressions for the selective area fraction, we obtain

$$S = \cfrac{1}{\cfrac{1}{S_{\mathrm{Nsp1}}} \cfrac{2}{r}\sigma V + \cfrac{1}{S_{\mathrm{open}}}\left(1 - \cfrac{2}{r}\sigma V\right)} \tag{25}$$

where $\sigma V$ is the only fit parameter that we fit to the selectivities measured in *Figure 6*.

Note that, if each Nsp1 molecule occupies the same volume $V$, the rim thickness would depend on the pore radius and the ring thickness $h$ can be calculated using $A_{\mathrm{Nsp1}} = \pi r^2 - \pi (r - h)^2$ as

$$h = r - \sqrt{r^2 - 2r\sigma V}, \tag{26}$$

which requires $\sigma V < \cfrac{r}{2}$. For flat surfaces, where $r \to \infty$ the parameter $\sigma V \to h$ and $\sigma V$ represents the Nsp1 layer height in these cases. In the case of curved surfaces, the layer height needs to be calculated using *Equation 26*. Notably, the assumption of constant volume that is applied here remains valid for situations where the Nsp1 phase is not localized in the rim, for example, forming a central plug but is in contrast to previous models that assume a selective area (of whatever shape) similar in size to a ring of constant thickness, such as that proposed by *Kowalczyk et al., 2011*.

## Appendix 12

### Nsp1

The sequence of the Nsp1 used in this study was

MHHHHHHHHHHHGSGENLYFQGTSMGNFNTPQQNKTPFSFGTANNNSNTTNQNSSTGAGAF
GTGQSTFGFNNSAPNNTNNANSSITPAFGSNNTGNTAFGNSNPTSNVFGSNNSTTNTFGSNSAG
TSLFGSSSAQQTKSNGTAGGNTFGSSSLFNNSTNSNTTKPAFGGLNFGGGNNTTPSSTGNANTSNNLF
GATANANKPAFSFGATTNDDKKTEPDKPAFSFNSSVGNKTDAQAPTTGFSFGSQLGGNKTVNEAAKPS
LSFGSGSAGANPAGASQPEPTTNEPAKPALSFGTATSDNKTTNTTPSFSFGAKSDENKAGATSKPAFS
FGAKPEEKKDDNSSKPAFSFGAKSNEDKQDGTAKPAFSFGAKPAEKNNNETSKPAFSFGAKSDEKKDG
DASKPAFSFGAKPDENKASATSKPAFSFGAKPEEKKDDNSSKPAFSFGAKSNEDKQDGTAKPAFSFGA
KPAEKNNNETSKPAFSFGAKSDEKKDGDASKPAFSFGAKSDEKKDSDSSKPAFSFGTKSNEKKDSGSS
KPAFSFGAKPDEKKNDEVSKPAFSFGAKANEKKESDESKSAFSFGSKPTGKEEGDGAKAAISFGAKPE
EQKSSDTSKPAFTFGAQKDNEKKTEC

### Kap95

The sequence of the GST-3C-Kap95 before removal of the GST tag and before labeling was:

MSPILGYWKIKGLVQPTRLLLEYLEEKYEEHLYERDEGDKWRNKKFELGLEFPNLPYYIDGDVKLTQS
MAIIRYIADKHNMLGGCPKERAEISMLEGAVLDIRYGVSRIAYSKDFETLKVDFLSKLPEMLKMFEDR
LCHKTYLNGDHVTHPDFMLYDALDVVLYMDPMCLDAFPKLVCFKKRIEAIPQIDKYLKSSKYIAWPLQ
GWQATFGGGDHPPKSDLEVLFQGPASVGSMSTAEFAQLLENSILSPDQNIRLTSETQLKKLSNDNFLQ
FAGLSSQVLIDENTKLEGRILAALTLKNELVSKDSVKTQQFAQRWITQVSPEAKNQIKTNALTALVSIEPRI
ANAAAQLIAAIADIELPHGAWPELMKIMVDNTGAEQPENVKRASLLALGYMCESADPQSQALVS
SSNNILIAIVQGAQSTETSKAVRLAALNALADSLIFIKNNMEREGERNYLMQVVCEATQAEDIEVQAA
AFGCLCKIMSLYYTFMKPYMEQALYALTIATMKSPNDKVASMTVEFWSTICEEEIDIAYELAQFPQSP
LQSYNFALSSIKDVVPNLLNLLTRQNEDPEDDDWNVSMSAGACLQLFAQNCGNHILEPVLEFVEQNIT
ADNWRNREAAVMAFGSIMDGPDKVQRTYYVHQALPSILNLMNDQSLQVKETTAWCIGRIADSVA
ESIDPQQHLPGVVQACLIGLQDHPKVATNCSWTIINLVEQLAEATPSPIYNFYPALVDGLIGAANRID
NEFNARASAFSALTTMVEYATDTVAETSASISTFVMDKLGQTMSVDENQLTLEDAQSLQELQSNILTV
LAAVIRKSPSSVEPVADMLMGLFFRLLEKKDSAFIEDDVFYAISALAASLGKGFEKYLETFSPYLLKALNQV
DSPVSITAVGFIADISNSLEEDFRRYSDAMMNVLAQMISNPNARRELKPAVLSVFGDIASNIGADFIP
YLNDIMALCVAAQNTKPENGTLEALDYQIKVLEAVLDAYVGIVAGLHDKPEALFPYVGTIFQFIAQVA
EDPQLYSEDATSRAAVGLIGDIAAMFPDGSIKQFYGQDWVIDYIKRTRSGQLFSQATKDTARWAREQQ
KRQLSLLPETGG

After removal of the GST tag and sortase labeling, the sequence of Kap95 as used during the experiment was:

GPASVGSMSTAEFAQLLENSILSPDQNIRLTSETQLKKLSNDNFLQFAGLSSQVLIDENTKLEGRILA
ALTLKNELVSKDSVKTQQFAQRWITQVSPEAKNQIKTNALTALVSIEPRIANAAAQLIAAIADIELPHGAWP
ELMKIMVDNTGAEQPENVKRASLLALGYMCESADPQSQALVSSSNNILIAIVQGAQSTETSKAVRLAA
LNALADSLIFIKNNMEREGERNYLMQVVCEATQAEDIEVQAAAFGCLCKIMSLYYTFMKPYMEQALYA
LTIATMKSPNDKVASMTVEFWSTICEEEIDIAYELAQFPQSPLQSYNFALSSIKDVVPNLLNLLTRQNEDPE
DDDWNVSMSAGACLQLFAQNCGNHILEPVLEFVEQNITADNWRNREAAVMAFGSIMDGPDKVQR
TYYVHQALPSILNLMNDQSLQVKETTAWCIGRIADSVAESIDPQQHLPGVVQACLIGLQDHPKVATNC
SWTIINLVEQLAEATPSPIYNFYPALVDGLIGAANRIDNEFNARASAFSALTTMVEYATDTVAETSASISTF
VMDKLGQTMSVDENQLTLEDAQSLQELQSNILTVLAAVIRKSPSSVEPVADMLMGLFFRLLEKKDSAF
IEDDVFYAISALAASLGKGFEKYLETFSPYLLKALNQVDSPVSITAVGFIADISNSLEEDFRRYSDAMMNVL
AQMISNPNARRELKPAVLSVFGDIASNIGADFIPYLNDIMALCVAAQNTKPENGTLEALDYQIKVLEA

VLDAYVGIVAGLHDKPEALFPYVGTIFQFIAQVAEDPQLYSEDATSRAAVGLIGDIAAMFPDGSIKQF
YGQDWVIDYIKRTRSGQLFSQATKDTARWAREQQKRQLSLLPETGGG-Alexa647

## Appendix 13

### Data sanitation

As described in the 'Materials and methods' section, some datasets were discarded due to a lower molecular brightness. In a plot of the molecular brightness against the pore diameter, these datasets were clearly identified as outliers (yellow markers in *Appendix 13—figure 1A and B*). We only used the molecular brightness of the Kap95 channel for discrimination because the low event rates for BSA at small-pore diameters render the molecular brightness estimate unreliable. We additionally find a dependence of the molecular brightness on the pore diameter both for Nsp1-coated open pores, with a reduction of the molecular brightness at smaller pore diameters (*Appendix 13—figure 1C and D*). This effect was most evident for BSA in Nsp1-coated pores where event rates were low, where it most likely originates from a reduction of the correlation amplitude due to a larger contribution of background signal.

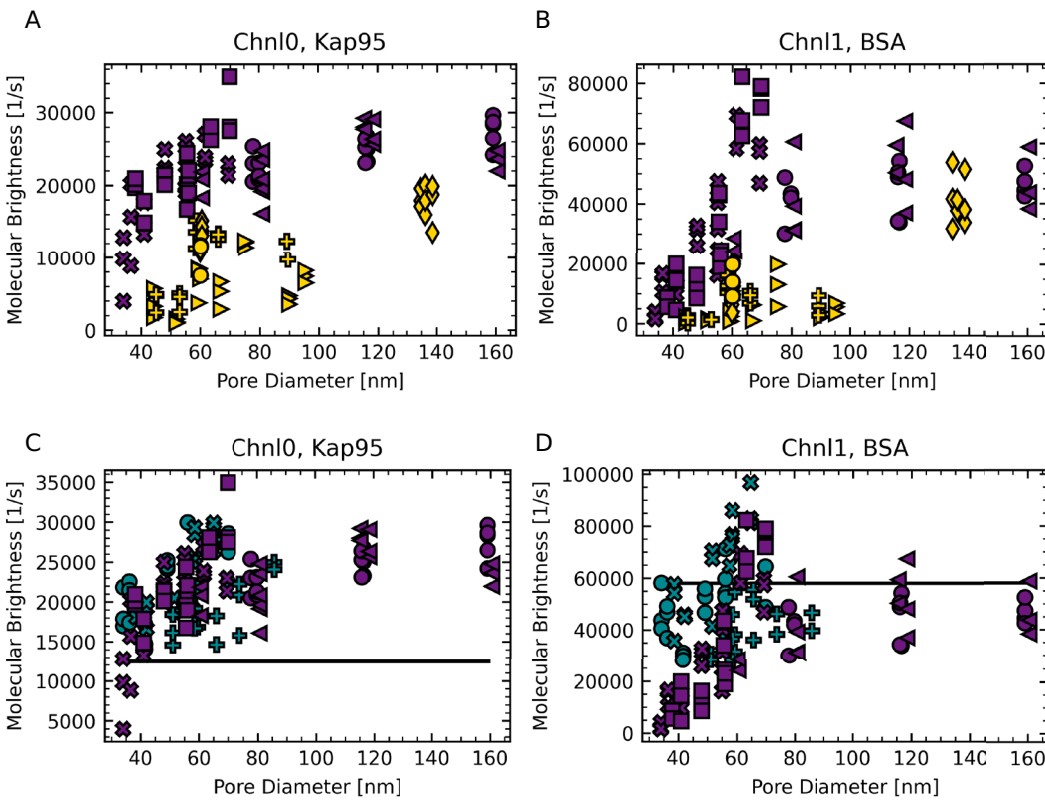

**Appendix 13—figure 1.** Molecular brightness for different time traces. (**A, B**) Molecular brightness per time trace versus pore diameter on Nsp1 pores. We see two populations for Kap95, labeled in purple and yellow. The yellow datasets show a decreased molecular brightness and therefore these datasets were removed from further analysis. (**C, D**) Comparison of molecular brightness between open pores (cyan), Nsp1 pores (purple), and free diffusion (black horizontal line). For Kap95, we found barely any difference of the data distribution. For BSA there, we observed a decrease in molecular brightness for small pores of Nsp1. This can be explained by the lower event rate in these pores. Different markers show different experiments.

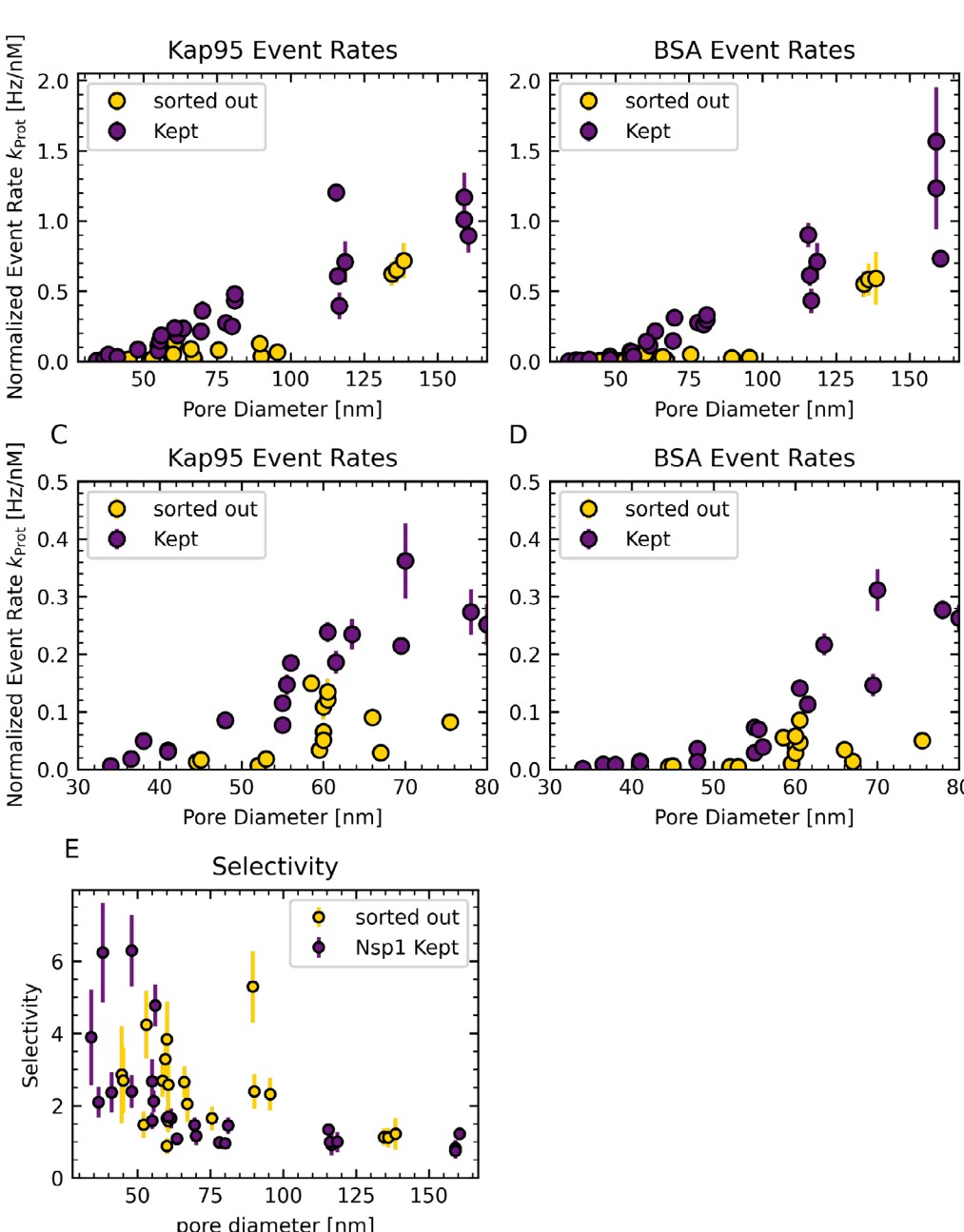

**Appendix 13—figure 2.** Effect of time trace removal. (**A, B**) Event rates of discarded pores of Nsp1. For completeness, we show the event rates of pores that were not further considered in the analysis (yellow) due to their lower molecular brightness, as shown in the previous figure. As expected, these pores show a much decreased normalized event rate both for Kap95 and BSA compared to the Nsp1 pores that were kept in the analysis (purple). (**C, D**) Zoom-in of (**A, B**). (**E**) Selectivity of discarded pores. The selectivity of discarded pores (yellow) deviates from the selectivity of the kept pores (purple). This can be explained when taking into account that a changed molecular brightness influences the event detection in each channel differently.

