## [Editor Report · eLife assessment]

This **important** study reports on a new method for the fabrication and the analysis of the transport through nuclear pore complexes mimic. Methods, data, and analyses are **convincing** and show a clear correlation between the size of the nuclear pore complex mimic and its transport selectivity. This work will be of high interest to biologists and biophysicists working on the mechanosensitivity of nucleocytoplasmic transport.

---

## [Referee Report · Reviewer #1 (Public Review)]

The contribution of Klughammer et al reports on the fabrication and functionalization of zero-mode waveguides of different diameters as a mimic system for nuclear pore complexes. Moreover, the researchers performed molecular transport measurements on these mimic systems (together with molecular dynamic simulations) to assess the contribution of pore diameter and Nsp functionalization on the translocation rates of BSA, the nuclear transport protein Kap95 and finally the impact of different Kap95 concentrations on BSA translocation and overall selectivity of the mimicked pores as function of their diameter. In order to assess the effect of the Nsp1 on the coated pores to the translocation rates and molecular selectivity they also conducted separated experiments on bare nano-pores, i.e., without coating, and of different diameters. One of the most novel aspects of this contribution is the detection scheme used to assess the translocation rates & selectivity, i.e., the use of an optical scheme based on single molecule fluorescence detection as compared to previous works that have mostly relied on conductance measurements. The results are convincing, the experiments carefully performed and the procedures explained in detail.

Importantly, this study provides new insights on the mechanisms of nuclear transport contributing to further our understanding on how real nuclear-pore complexes (i.e., in living cell) can regulate molecular transport. The recent findings that the nuclear pore complexes are sensitive to mechanical stimulation by modulating their effective diameters, adds an additional level of interest to the work reported here, since the authors thoroughly explored different nano-pore diameters and quantified their impact on translocation and selectivity. There are multiple avenues for future research based the system developed here, including higher throughput detection, extending to truly multicolor schemes or expanding the range of FG-Nups, nuclear transport proteins or cargos that need to be efficiently transported to the nucleus through the nuclear pore complexes. As a whole, this is an important contribution to the field.

---

## [Referee Report · Reviewer #2 (Public Review)]

In this study, a minimalist setup was used to investigate the selectivity of the nuclear pore complex as a function of its diameter. To this end, a series of solid-state pores in a free-standing palladium membrane were designed and attached to a PDMS-based fluid cell that could be mounted on a confocal microscope. In this way, the frequency of translocation events could be measured in an unbiased manner. Furthermore, the pores were designed to exhibit the key properties of the nuclear pore complex: (i) the size of the pore, (ii) disordered FG Nups specifically located in the central channel; (ii) transport receptors that can shuttle through the central channel by binding to the FG-Nups. Additionally, such system offered the advantage of monitoring the translocation of multiple fluorescently labeled molecules (e.g. Kap95 and BSA) simultaneously and under well-controlled conditions.

The authors were able to demonstrate convincingly that the pore selectivity depends on the pore diameter, the FG Nup layer organization within the pore and the transport receptors concentration that can specifically interact with FG Nups. It was shown that the pores coated with FG Nups (e.g. Nsp1 in this case) and smaller than 50-60 nm are highly selective and such selectivity is increasing with the decrease of the pore diameter. Also, it was shown that the pore selectivity moderately enhances at the high Kap95 concentration (1 µM). Importantly, it was also shown that the selectivity is becoming negligible for the pores, which are larger than 60-75 nm.

The experimental data are well supported by coarse-grained modelling of Nsp1-coated pores, and the theoretical prediction correlates qualitatively with the experimentally obtained data.

---

## [Author Response]

The following is the authors’ response to the original reviews.

**Reviewer #1:**
1. The most important concern that I have refers to the FDTD simulations to characterize the ZMW, as shown in Appendix 2, Figure 4. So far, the explanations given in the caption of Figure 4 are confusing and misleading: the authors should provide more detailed explanations on how the simulations were performed and the actual definition of the parameters used. In particular:a. lines 1330-1332: it is not clear to me how the fluorescence lifetime can be calculated from the detected signal S (z), and why they are horizontal, i.e., no z dependence? Which lifetimes are the authors referring to?b. lines 1333-1335: Where do these values come from? And how do they relate to panels D & E? From what I can see in these panels the lifetimes are highly dependent on z and show the expected reduction of lifetime inside the nanostructures.c. lines 1336-1337: Why the quantum yield of the dyes outside the ZMW differs from those reported in the literature? In particular the changes of quantum yield and lifetime for Alexa 488 are very large (also mentioned in the corresponding part of Materials & Methods but not explained in any detail).

We thank the Reviewer for his detailed questions on the FDTD simulations. We have now added the missing equation related to the computation of signal-averaged fluorescence lifetimes from the FDTD simulations. Specifically to the three points raised:

a) The fluorescence lifetime is indeed not calculated from the detected signal S(z), but from the radiative and non-radiative rates in the presence of the ZMW as given in eq. 9-10. However, we use the detected signal S(z) to compute the average fluorescence lifetime over the whole z-profile of the simulation box, which we relate to the experimentally measured fluorescence lifetimes as given in Appendix 7, Figure 1. We have now added the equation to compute the signal-weighted fluorescence lifetimes, which we denote as <τ>S , in eq. 13 in the methods. To clarify this point, we have added the symbol <τ>S to the plots in Appendix 2, Figure 4 D-E and Appendix 7, Figure 1 C-D.

b) The estimated lifetimes were obtained as the signal-weighted average over the lifetime profiles, (<τ>S) as given in the new eq. 13. All plotted quantities, i.e., the detection efficiency η, quantum yield ϕ, detected signal S(z), and fluorescence lifetime, are computed from the radiative and loss rates obtained from the FDTD simulation according to eqs. 8-11. To make this clearer, we have now added the new Appendix 2 – Figure 5 which shows the z-profiles of the quantities (radiative and loss rates) used to derive the experimental observables.

c) There are multiple reasons for the differences of the quantum yields of the two analytes used in this study compared to the literature values. For cyanine dyes such as Alexa647, it is well known that steric restriction (as e.g. caused by conjugation to a biomolecule) can lead to an increase of the quantum yield and fluorescence lifetime. We observe a minor increase of the fluorescence lifetime for Alexa647 from the literature value of 1.17 ns to a value of 1.37 ns when attached to Kap95, which is indicative of this effect. In the submitted manuscript, this was discussed in the methods in lines 936-938 (lines 938-945 in the revised manuscript). For the dye Alexa488, which is used to label the BSA protein, this effect is absent. Instead, we observe (as the Reviewer correctly notes) a quite drastic reduction of the fluorescence lifetime compared to the unconjugated dye from 4 ns to 2.3 ns. In cases where a single cysteine is labeled on a protein, such a drastic reduction of the quantum yield usually indicates the presence of a quenching moiety in proximity of the labeling site, such as tryptophane, which acts via the photo-induced electron transfer mechanism. Indeed, BSA contains two tryptophanes that could be responsible for the low quantum yield of the conjugated dyes. The situation is complicated by the fact that BSA contains 35 cysteines that can potentially be labeled (although 34 are involved in disulfide bridges). The labeled BSA was obtained commercially and the manufacturer lists the degree of labeling as ~6 dye molecules per protein, with a relative quantum yield of 0.2 compared to the standard fluorescein. This corresponds to an absolute quantum yield of ~0.16, which is low compared to the literature value for Alexa488 of ~0.8.

Based on the measured fluorescence lifetime, we estimate a quantum yield of 0.46, which is higher than the photometrically obtained value of 0.16 reported by the manufacturer. Fully quenched, nonfluorescent dyes will not contribute to the lifetime measurement but are detected in the photometric quantum yield estimates. The difference between the lifetime and photometric based quantum yield estimates thus suggest that part of the fluorophores are almost fully quenched. While it is unknown where the dyes are attached to the protein, the low quantum yield could be indicative of dye-dye interactions via pi-pi stacking, which can often lead to non-fluorescent dimers. This is supported by the fact that the manufacturer reports color differences between batches of labeled protein, which indicate spectral shifts of the absorption spectrum when dye-dye adducts are formed by π-π stacking. We have now added a short discussion of this effect in lines 938-941. We note that the conclusions drawn on the quenching effect of the metal nanostructure remain valid despite the drastic reduction of the quantum yield for Alexa488, which leads to a further quantum yield reduction of the partly quenched reference state.

1. A second important concern refers to Figure 3: Why is there so much variability on the burst intensities reported on panels C, D? They should correspond to single molecule translocation events and thus all having comparable intensity values. In particular, the data shown for BSA in panel D is highly puzzling, since it not only reflects a reduced number of bursts (which is the main finding) but also very low intensity values, suggesting a high degree of quenching of the fluorophore being proximal to the metal on the exit side of the pore. In fact, the count rates for BSA on the uncoated pore range form 50-100kcounts/s, while on the coated pores thy barely reach 30 kcounts/s, a clear indication of quenching. Importantly, and in direct relation to this, could the authors exclude the possibility that the low event rates measured on BSA are largely due to quenching of the dye by getting entangled in the Nsp mesh just underneath the pore but in close contact to the metal?

The Reviewer raises a valid concern, but further analysis shows that this is unproblematic. Notably, the burst intensities are in fact not reduced, in contrast to the visual impression obtained from the time traces shown in the figure. The time trace of the BSA intensity is visually dominated by high-intensity bursts which mask the low-intensity bursts in the plot. In contrast, in Figure 3 the reduced number of BSA events results in a sparser distribution of the intensity spikes, which allows low-intensity events to be seen. Different to the visual inspection, the spike-detection algorithm does not exhibit any bias in terms of the duration or the number of photons of the detected events between the different conditions for both BSA and Kap95, as shown in the new Appendix 7 – Figure 1. Using FCS analysis it can be tested whether the event duration varies between the different conditions shown in Figure 3 C-D. This did not show a significant difference in the estimated diffusion time for BSA (Appendix 7 – Figure 1 C,D). Contrary to the suggestion of the Reviewer, we also do not observe any indication of quenching by the metal between uncoated and Nsp1-coated pores for BSA. Such quenching should result in differences of the fluorescence lifetimes, which however is not evident in our experimental data (Appendix 7 – Figure 1 F).

1. Line 91: I suggest the authors remove the word "multiplexed" detection since it is misleading. Essentially the authors report on a two-color excitation/detection scheme which is far from being really multiplexing.

We have changed the word to “simultaneous” now and hope this avoids further confusion.

1. Line 121: why are the ZMW fabricated with palladium? Aluminum is the gold-standard to reduce light transmissivity. An explanation for the choice of this material would be appreciated by the community.

In a previous study (Klughammer and Dekker, Nanotechnology, 2021), we established that palladium can have distinct advantages compared to other ZMW metals such as aluminum and gold, most prominently, an increased chemical stability and reduced photoluminescence. For this study, we chose palladium over aluminum as it allowed the use of simple thiol chemistry for surface modification. In the beginning of the project, we experimented with aluminum pores as well. We consistently found that the pores got closed after measuring their ionic conductance in chlorine-containing solutions such as KCl or PBS. This problem was avoided by choosing palladium.

1. Lines 281-282: This statement is somewhat misleading, since it reads such that the molecules stay longer inside the pore. However, if I understand correctly, these results suggest that Kap95 stays closer to the metal on the exit side. This is because measurements are being performed on the exit side of the pore as the excitation field inside the pore is quite negligible.

We thank the Reviewer for this comment and have clarified the text in lines 290-292 as suggested to: “(…) this indicates that, on the exit side, Kap95 diffuses closer to the pore walls compared to BSA due to interactions with the Nsp1 mesh”

1. Lines 319-320: Although the MD simulations agree with the statement being written here, the variability could be also due to the fact that the proteins could interact in a rather heterogenous manner with the Nsp mesh on the exit side of the pore, transiently trapping molecules that then would stay longer and/or closer to the metal altering the emission rate of the fluorophores. Could the authors comment on this?

The variation mentioned in the text refers to a pore-to-pore variation and thus needs to be due to a structural difference between individual pores. This effect would also need to be stable for the full course of an experiment, typically hours. We did not find any structural changes in the fluorescence lifetimes measured on individual pores such as suggested by the Reviewer. We think that the suggested mechanism would show up as distinct clusters in Appendix 7 – Figure 1 E,F where we found no trace of such a change to happen. If we understand correctly, the Reviewer suggests a mechanism, not based on changes in the Nup layer density, that would lead to a varying amount of trapping of proteins close to the surface. Such a behavior should show up in the diffusion time of each pore ( Appendix 7 – figure 1 C,D), where we however find no trace of such an effect.

1. Lines 493-498: These claims are actually not supported by the experimental data shown in this contribution: (a) No direct comparison in terms of signal-to-noise ratio between fluorescence-based and conductance-based readouts has been provided in the ms. (b) I would change the word multiplexed by simultaneous since it is highly misleading. (c) The results shown are performed sequentially and thus low throughput. (d) Finally, the use of unlabeled components is dubious since the detection schemes relies on fluorescence and thus requiring labeling.

We thank the Reviewer for pointing this out.

a) We have now added a section in appendix 3 that discusses the signal-to-noise ratios. In brief, there are three observations that led us to conclude that ZMWs provide beneficial capabilities to resolve individual events from the background:

1. The signal-to-background ratio was determined to be 67±53 for our ZMW data of Kap95 which is an order of magnitude higher compared to the ~5.6 value for a conductance-based readout.

2. The detection efficiency for ZMWs is independent of the Kap95 occupancy within the pore. This is different from conductance based approaches that have reduced capability to resolve individual Kap95 translocations at high concentrations.

3. The fraction of detected translocations is much higher for ZMWs than for conductance-based data (where lots of translocations occur undetected) and matches closer to the theoretical predictions.

b) We have changed the wording accordingly.

c) We agree with the Reviewer that our method is still low throughput. However, the throughput is markedly increased compared to previous conductance-based nanopore measurements. This is because we can test many (here up to 8, but potentially many more) pores per chip in one experiment, whereas conductance-based readouts are limited to a single pore. We have now changed the wording to “increased throughput” in line 507 to avoid confusion.

d) We agree that only labeled components can be studied directly with our methods. However, the effect of unlabeled analytes can be assessed indirectly without any perturbation of the detection scheme due to the specificity of the fluorescent labeling. This is distinct from previous nanopore approaches using a conductance-based readout that lack specificity. In our study, we have for example used this advantage of our approach to access event rates at high concentrations (1000nM Kap95, 500nM BSA) and large pore diameters by reducing the fraction of labeled analyte in the sample. Finally, the dependence of the BSA leakage rate as a function of the concentration of Kap95 (Figure 6) relies on a specific readout of BSA events in the presence of large amounts of Kap95, which would be impossible in conductance-based experiments.

1. Line 769: specify the NA of the objective. Using a very long working distance would also affect the detection efficiency. Have the authors considered the NA of the objective on the simulations of the detection efficiency? This information should be included and it is important as the authors are detecting single molecule events.

We used an NA of 1.1 for the simulation of the Gaussian excitation field in the FDTD simulations, corresponding to the NA of the objective lens used in the experiments and as specified in the methods. The Reviewer is correct that the NA also affects the absolute detection efficiency of the fluorescence signal due to the finite opening angle of the collection cone of ~56°. In our evaluation of the simulations, we have neglected this effect for simplicity, because the finite collection efficiency of the objective lens represents only an additional constant factor that does not depend on the parameters of the simulated system, such as the pore diameter. Instead, we focused solely the effect of the ZMW and defined the detection efficiency purely based on the fraction of the signal that is emitted towards the detection side and can potentially be detected in the experiment, which also provides the benefit that the discussed numbers are independent of the experimental setup used.

To clarify this, we have now made this clearer in the method text on lines 917-920.

1. Line 831: I guess that 1160ps is a mistake, right?

This is not a mistake. We performed a tail fit of the fluorescence decay curves, meaning that the initial rise of the decay was excluded from the fit. The initial part of the fluorescence decay is dominated by the instrument response function (IRF) of the system, with an approximate width of ~500 ps. To minimize the influence of the IRF on the tail fit, we excluded the first ~1 ns of the fluorescence decay.

1. Lines 913-917: Why are the quantum yield of Alexa 488 and lifetime so much reduced as compared to the published values in literature?

See answer to point 1. We have added a short discussion at lines 938-941 where we speculate that the reduced quantum yield is most likely caused by dye-dye interactions due to the high degree of labeling of ~6 dyes per protein.

1. Lines 1503-1509: The predicted lifetimes with the Nsp-1 coating have not been shown in Appendix 2 - Figure 4. How have they been estimated?

We have not performed predictions of fluorescence lifetimes in the presence of an Nsp1 coating. Predictions of the fluorescence lifetime in the absence of the Nsp1 coating were obtained by assuming a uniform occupancy of the molecules over the simulation box. A prediction of the fluorescence lifetimes in the presence of the Nsp1 coating would require a precise knowledge of the spatial distribution of analytes, which depends, among other factors, on the extension of the Nsp1 brushes and the interaction strengths with the FG repeats. While simulations provide some insights on this, we consider a quantitative comparison of predicted and measured fluorescence lifetimes in the presence of the Nsp1 coating beyond the scope of the present study.

1. Lines 1534-1539: I disagree with this comment, since the measurements reported here have been performed outside the nano-holes, and thus the argument of Kap95 translocating along the edges of the pore and being responsible for the reduced lifetime does not make sense to me.

In accordance with our answer to point 5 above, we have now changed the interpretation to the proximity of Kap95 to the metal surface on the exit side, rather than speculating on the path that the protein takes through the pore (lines 1662-1664), as follows:

“This indicates that, in the presence of Nsp1, Kap95 molecules diffuse closer to or spend more time in proximity of the metal nanoaperture on the exit side.”

**Reviewer #2:**
(Numbers indicate the line number.)48: should cite more recent work: Timney et al. 2016 Popken et al 201559: should cite Zilman et al 2007, Zilman et al 201062: should cite Zilman et al 2010

We thank the Reviewer for the suggestions and have added them to the manuscript now.

65: one should be careful in making statements that the "slow" phase is immobile, as it likely rapidly exchanging NTRs with the "fast" phase.

We have removed this description and replaced it by “This 'slow phase' exhibits a reduced mobility due to the high affinity of NTRs to the FG-Nup mesh.” to avoid misunderstanding.

67: Schleicher 2014 does not provide evidence of dedicated channels

We agree with the Reviewer and therefore moved the reference to an earlier position in the sentence.

74-75: must cite work by Lusk & Lin et al on origami nanochannels

We thank the Reviewer for this suggestion. We have now added a reference to the nanotraps of Shen et al. 2021, JACS, in line 75. In addition, we now also refer to Shen et al. 2023, NSMB, in the discussion where viral transport is discussed.

77: Probably Jovanovic- Talisman (2009)?

We thank the Reviewer for pointing out this typo.

93; should cite Auger&Montel et al, PRL 2014

We thank the Reviewer for pointing out this reference. To give proper credit to previous ZMW, we have now incorporated a sentence in lines 100-102 citing this reference.

111-112: there appears to be some internal inconsistency between this interpretation and the BSA transport mostly taking place through the "central hole" as seems to be implied by Equation (3). Probably it should be specified explicitly that the "central hole" in large channels is a "void".

We thank the Reviewer for this suggestion and have added a clarifying sentence.

115-177: This competition was studied in Jovanovic-Talisman 2009 and theoretically analysed in Zilman et al Plos Comp Biol 2010. The differences in the results and the interpretation should be discussed.

We agree, therefore it is discussed in the discussion section (around line 594) and now added the reference to Zilman et al.

Figure 2 Caption: "A constant flow..." - is it clear that is flow does not generate hydrodynamic flow through the pore?

The Reviewer raises an important point. Indeed, the pressure difference over the membrane generates a hydrodynamic flow through the pore that leads to a reduction of the event rate compared to when no pressure is applied. However, as all experiments were performed under identical pressures, one can expect a proportional reduction of the absolute event rates due to the hydrodynamic flow against the concentration gradient. In other words, this will not affect the conclusions drawn on the selectivity, as it is defined as a ratio of event rates.

We have now added additional data on the influence of the hydrodynamic flow on the translocation rate in Appendix 3 – Figure 2, where we have measured the signal of free fluorophores at high concentration on the exit side of the pore as a function of the applied pressure. The data show a linear dependence of the signal reduction on the applied pressure. At the pressure values used for the experiments of 50 mbar, we see a ~5% reduction compared to the absence of pressure, implying that the reported absolute event rates are underestimated only by ~5%. Additionally we have added such data for Kap95 translocations that shows a similar effect (however less consistent). Measuring the event rate at zero flow is difficult, since this leads to an accumulation of fluorophores on the detection side.

Figure 3: it would help to add how long is each translocation, and what is the lower detection limit. A short explanation of why the method detects actual translocations would be good

With our method, unfortunately, we can not assess the duration of a translocation event since we only see the particle as it exists the pore. Instead, the measured event duration is determined by the time it takes for the particle to diffuse out of the laser focus. This is confirmed by FCS analysis of translocation events that show the same order of magnitude of diffusion times as for free diffusion (Appendix 7 – Figure 1 C,D) in contrast to a massively reduced diffusion time within a nanopore. In Figure 2D we show the detection efficiency at different locations around the ZMW as obtained from FDTD simulations and discuss the light blocking. This clearly shows that the big majority of the fluorescence signal comes from the laser illuminated side and therefore only particles that translocated through the ZMW are detected as presented between lines 170-190. In Yang et al. 2023, bioRxiv (https://doi.org/10.1101/2023.06.26.546504) a more detailed discussion about the optical properties of Pd nanopores is given.

This point also explains why we see actual translocations: since the light is blocked by the ZMW, fluorophores can only be detected after they have translocated. On parts of the membrane without pores and upstream the amount of spikes found in a timetrace was found to be negligibly small. Additionally, if a significant part of the signal would be contributed by leaking fluorescence from the dark top side, there should no difference in BSA event rate found between small open and Nsp1 pores which we did not observe.

With respect to the lower detection limit for events: In the burst search algorithm we require a false positive level rate of lower than 1 event in 100. Additionally, as described in Klughammer and Dekker, Nanotechnology (2021), we apply an empirical filtering to remove low signal to noise ratio events that contain less than 5 detected photons per event or a too low event rate. From the event detection algorithm there is no lower limit set on the duration of an event. Such a limit is then set by the instrument and the maximum frequency it which it can detect photons. This time is below 1μs. Practically we don’t find events shorter than 10μs as can be seen in the distribution of events where also the detection limits can be estimated (Appendix 7 – figure 1 A and B.)

Equation (1): this is true only for passive diffusion without interactions (see eg Hoogenboom et al Physics Reports 2021 for review). Using it for pores with interactions would predict, for instance, that the inhibition of the BSA translocation comes from the decrease in D which is not correct.

We agree with the Reviewer that this equation would not reproduce the measured data in a numerically correct way. We included it to justify why we subsequently fit a quadratic function to the data. As we write in line 260 we only used the quadratic equation “as a guide to the eye and for numerical comparison” and specifically don’t claim that this fully describes the translocation process. In this quadratic function, we introduced a scaling factor α that can be fitted to the data and thus incorporates deviations from the model. In appendix 5 we added a more elaborate way to fit the data including a confinement-based reduction of the diffusion coefficient (although not incorporating interactions). Given the variations of the measured translocation rates, the data is equally well described by both the simple and the more complex model function.

Equation (1): This is not entirely exact, because the concentration at the entrance to the pore is lower than the bulk concentration, which might introduce corrections

We agree with the Reviewer and have added that the concentration difference Δc is measured at the pore entrance and exit, and this may be lower than the bulk concentration. As described in our reaction to the Reviewer’s previous comment, equation (1) only serves as a justification to use the quadratic dependence and any deviations in Δc are absorbed into the prefactor α in equation (2).

Equation (3): I don't understand how this is consistent with the further discussion of BSA translocation. Clearly BSA can translocate through the pore even if the crossection is covered by the FG nups (through the "voids" presumably?).

The Reviewer raises an important point here. Equation 3 can only be used for a pore radius r > rprot + b. b was determined to be 11.5 nm and rprot is 3.4 nm for BSA, thus it needs to be that r > 15 nm. We would like to stress, however, that b does not directly give a height of a rigid Nsp1 ring but is related to the configuration of the Nsp1 inside the pore. Equation (3) (and equation (2)) were chosen because even these simple equations could fit the experimentally measured translocation rates well, and not because they would accurately model the setup in the pore. As we found from the simulations, the BSA translocations at low pore diameters presumably happen through transient openings of the mesh. The dynamics leading to the stochastic opening of voids on average leads to the observed translocation rate.

296-297: is it also consistent with the simulations?

We compare the experimentally and simulated b values in lines 387-388 and obtained b=9.9 ± 0.1 nm from the simulations (as obtained from fitting the translocation rates and not from measuring the extension of the Nsp1 molecules) and 11.5 ± 0.4 nm from the experiments – which we find in good agreement.

331: has it been established that the FG nups equilibrate on the microsecond scale?

As an example, we have analyzed the simulation trajectory of the most dense nanopore (diameter = 40 nm, grafting = 1/200 nm2). In Author response image 1 we show for each of the Nsp1-proteins how the radius of gyration (Rg) changes in time over the full trajectory (2 μs + 5 μs). As expected, the Rg values reached the average equilibrium values very well within 2 μs simulation time, showing that the FG-Nups indeed equilibrate on the (sub)microsecond scale.

334-347: the details of the method should be explained explicitly in the supplementary (how exactly voids distributions are estimated and the PMF are calculated etc)

The void analysis was performed with the software obtained from the paper of Winogradoff et al. In our Methods we provide an overview of how this software calculates the void probability maps and how these are converted into PMFs. For a more detailed description of how exactly the analysis algorithm is implemented in the software, we refer the reader to the original work. The analysis codes with the input files that were used in this manuscript have been made public ( https://doi.org/10.4121/22059227.v1 ) along with the manuscript.

Equation (4) is only an approximation (which works fine for high barriers but not the low ones). Please provide citations/derivation.

To our knowledge, the Arrhenius relation is a valid approximation for our nanopore simulations. We are unaware of the fact that it should not work for low barriers and cannot find mention of this in the literature. It would be helpful if the Reviewer can point us to relevant literature.

Figure 4: how was transport rate for Kaps calculated?

As mentioned in lines 388-391, we assumed that the Kap95 translocation rate through Nsp1-coated pores is equal to that for open pores, as we did not observe any significant hindrance of Kap95 translocation by the Nsp1 mesh in the experiment (Figure 4 A,C).

378: It's a bit strange to present the selectivity ratio as prediction of the model when only BSA translocation rate was simulated (indirectly).

We agree with the Reviewer that ideally we should also simulate the Kap95 translocation rate to obtain an accurate selectivity measure of the simulated nanopores. However, as the experiments showed very similar Kap95 translocation rates for open pores and Nsp1-coated pores, we believe it is reasonable to take the Kap95 rates for open and Nsp1-pores to be equal.

Figure 5C and lines 397: I am a bit confused how is this consistent with Figure 4D?

Figure 5C and figure 4D both display the same experimental data, where 4D only focuses on a low diameter regime. In relation to line 397 (now 407), the Nsp1 mesh within the 60-nm pore dynamically switches between closed configurations and configurations with an open channel. When taking the temporal average of these configurations, we find that the translocation rate is higher than for a closed pore but lower than for a fully open pore. The stochastic opening and closing of the Nup mesh results in the continuous increase of the translocation rates with increasing diameter, which is in contrast to a step-wise increase that would be expected from an instantaneous collapse of the Nsp1 mesh at a certain pore diameter.

428-439: Please discuss the differences from Jovanovic-Talisman 2009.

How our results for a Kap95 induced change of the BSA translocation rate are related to previous literature is discussed extensively in the lines 598-620.

440: How many Kaps are in the pore at different concentrations?

This is a very interesting question that we were, unfortunately, not able to answer within the scope of this project. With our fluorescent based methods we could not determine this number because the excitation light does not reach well into the nanopore.

In our previous work on Nsp1-coated SiN nanopores using conductance measurements, we quantified the drop in conductance at increasing concentrations of Kap95 (Fragasso et al., 2023, NanoResearch, http://dx.doi.org/10.1007/s12274-022-4647-1). From this, we estimated that on average ~20 Kap95 molecules are present in a pore with a diameter of 55 nm at a bulk concentration of 2 µM. In these experiments, however, the height of the pore was only ~20 nm, which is much lower compared to 100 nm long channel used here, and the grafting density of 1 per 21 nm2 was high compared to the grafting density here of 1 per 300 nm2. Assuming that the Kap95 occupancy scales linearly with the number of binding sites (FG repeats) in the vicinity of the pore, and hence the amount of Nsp1 molecules bound to the pore, we would expect approximately ~7 Kap95 molecules in a pore of similar diameter under saturating (> 1 µM) concentrations.

On the other hand, the simulations showed that the density of Nsp1 within the pore is equal to the density within the 20-nm thick SiN pores (line 380). For the longer channel and lower grafting density used here, Nsp1 was also more constrained to the pore compared to thinner pores used in previous studies (Fragasso et al., 2023, NanoResearch), where the grafted protein spilled out from the nanopores. Thus assuming that the Kap95 occupancy depends on the protein density in the pore volume rather than the total protein amount grafted to the pore walls, we would estimate a number of 100 Kap95 molecules per pore.

These varying numbers already show that we cannot accurately provide an estimate of the Kap95 occupancy within the pore from our data due to limitations of the ZMW approach.

445: how is this related to the BSA translocation increase?

For the calculation of the selectivity ratio, we assumed the normalized Kap95 translocation rate to be independent of the Kap95 concentration. Hence, the observed trends of the selectivity ratios at different concentrations of Kap95, as shown in Figure 6 D, are solely due to a change in the BSA translocation rate at different concentrations of Kap95, as given in Figure 6 B,C.

462-481: it's a bit confusing how this interfaces with the "void" analysis ( see my previous comments)

We agree that the phenomenological descriptions in terms of transient openings (small, dynamic voids) that for larger pores become a constantly opened channel (a single large, static void) might cause some confusion to the reader. In the last part of the results, we aimed to relate the loss of the BSA rate to a change of the Nsp1 mesh. We acknowledge that the model of a rim of Nsp1 and an open center described in Figure 5F is highly simplifying . We now explain this in the revised paper at lines 483-486 by referring to an effective layer thickness which holds true under the simplifying assumption of a central transport channel.

Figure 6D: I think the illustration of the effect of kaps on the brush is somewhat misleading: at low pore diameters, it is possible that the opposite happens: the kaps concentrate the polymers towards the center of the pore. It should be also made clear that there are no kaps in simulations (if I understand correctly?)

Indeed, at small pore diameters we think it would be possible to observe what the Reviewer describes. The illustration should only indicate what we think is happening for large pore diameters where we observed the opening of a central channel. To avoid confusion, we now shifted the sketches to panel G where the effective layer thickness is discussed.

Indeed, as stated in lines 331-340 no Kap95 or BSA molecules were present in the simulations. We have now clarified this point in lines 872-876.

518: Please provide more explanation on the role of hydrodynamics pressure.

We have now performed additional experiments and quantified the effect of the pressure to be a ~5% reduction of the event rates, as described in the answer to a previous question above.

**Reviewer #3 (Recommendations For The Authors):**
No experiments have been performed with the Ran-Mix regeneration system. It would be beneficial to add Ran-Mix to the trans compartment and see how this would affect Kap95 translocation events frequency and passive cargo diffusion. As the authors note in their outlook, this setup offers an advantage in using Ran-Mix and thus could also be considered here or in a future follow-up study.

We thank the Reviewer for this suggestion. We think, however, that it is beyond the scope of this paper and an interesting subject for a follow-up study.